# On the Complexity Theory of Masked Discrete Diffusion: From $\mathrm{poly}(1/\epsilon)$ to Nearly $\epsilon$-Free

## Abstract

We study *masked discrete diffusion*—a flexible paradigm for text generation in which tokens are progressively corrupted by special mask symbols before being denoised. Although this approach has demonstrated strong empirical performance, its theoretical complexity in high-dimensional settings remains insufficiently understood. Existing analyses largely focus on *uniform* discrete diffusion, and more recent attempts addressing masked diffusion either (1) overlook widely used Euler samplers, (2) impose restrictive bounded-score assumptions, or (3) fail to showcase the advantages of masked discrete diffusion over its uniform counterpart. To address this gap, we show that Euler samplers can achieve $\epsilon$-accuracy in total variation (TV) with $\tilde{O}(d^2\epsilon^{-3/2})$ discrete score evaluations, thereby providing the first rigorous analysis of typical Euler sampler in masked discrete diffusion. We then propose a *Mask-Aware Truncated Uniformization* (MATU) approach that both removes bounded-score assumptions and preserves unbiased discrete score approximation. By exploiting the property that each token can be unmasked at most once, MATU attains a nearly $\epsilon$-free complexity of $O(d \ln d \cdot (1 - \epsilon^2))$. This result surpasses existing uniformization methods under uniform discrete diffusion, eliminating the $\ln(1/\epsilon)$ factor and substantially speeding up convergence. Our findings not only provide a rigorous theoretical foundation for masked discrete diffusion, showcasing its practical advantages over uniform diffusion for text generation, but also pave the way for future efforts to analyze diffusion-based language models developed under masking paradigm.

## 1 Introduction

Diffusion language models (Sohl-Dickstein et al., 2015; Hoogeboom et al.; Austin et al., 2021; Lou et al., 2024; Ou et al., 2024) have recently emerged as a powerful class of generative paradigms, frequently regarded as both complements and competitors to the auto-regressive based language models (Achiam et al., 2023; Touvron et al., 2023; Zhao et al., 2023). Whereas auto-regressive models learn the conditional distribution of the next token given a prefix, diffusion language models approximate the joint distribution of an entire token sequence through a noising–denoising process. This process transforms a potentially complex data distribution into a simpler prior distribution and then iteratively reconstructs it. In the forward (*noising*) direction, tokens are progressively replaced by special mask symbols, thereby mapping the data distribution to a one-hot stationary distribution. The reverse (*denoising*) direction then recovers the original text step by step by estimating discrete scores (i.e., density ratios) over the corrupted samples.

Although masked discrete diffusion has empirically outperformed uniform discrete diffusion (where the forward process admits a uniform stationary distribution) (Lou et al., 2024), analyzing and mitigating its computational overhead in high-dimensional settings remains challenging. As summarized in Table 3, most existing theoretical results focus on *uniform discrete diffusion*. In these analyses, Euler-type samplers approximate continuous-time scores by holding them constant over short intervals, leading to polynomial complexity in the total variation (TV) distance $\epsilon$. Specifically, exponential-integrator methods (Zhang et al., 2024) require $\tilde{O}(\epsilon^{-2})$ steps, while $\tau$-leaping methods (Campbell et al., 2022; Lou et al., 2024) and their higher-order variants (Ren et al., 2025) need $\tilde{O}(\epsilon^{-1})$ steps. Notably, uniformization-based techniques offer a promising approach, achieving $O(\ln(1/\epsilon))$ complexity by unbiasedly simulating the reverse Markov chain. In the context of

*masked discrete diffusion*, Liang et al. (2025a) rigorously examined $\epsilon$-TV convergence, showing that $\tau$-leaping can take $\tilde{O}(\epsilon^{-2})$ steps to converge and also improves upon the dimensional dependence found in uniform discrete diffusion. However, their stronger bounded-score assumptions make direct comparisons of algorithmic complexity with existing works (Chen & Ying, 2024; Huang et al., 2025) uncertain. Although uniformization can theoretically reach a complexity of $O(\ln(1/\epsilon))$ in their framework, it retains the same $\epsilon$-dependence as uniform discrete diffusion and has yet to exhibit clear empirical benefits in masked diffusion. Finally, the analysis of the typical Euler sampler used in most empirical studies (Lou et al., 2024; Ou et al., 2024) is still not fully understood.

To address the theoretical challenges of masked discrete diffusion, we first analyze a typical Euler sampler that parallels the inference procedures used in many empirical studies (Lou et al., 2024; Ou et al., 2024). Our findings reveal that reaching $\epsilon$-TV convergence in masked discrete diffusion with the typical Euler sampler requires $\tilde{O}(d^2\epsilon^{-3/2})$ discrete score evaluations. This result stands as the first rigorous analysis of the typical Euler method in masked discrete diffusion and demonstrates faster convergence than the $\tau$-leaping approach (Liang et al., 2025a) under stringent accuracy demands. We then examine uniformization-based approaches for masked discrete diffusion, where uniformization converts a continuous-time Markov chain (CTMC) into a discrete-time Markov chain (DTMC) by sampling random Poisson jump times. This technique preserves the exact transition structure of the original CTMC and provides an unbiased simulation without time-step discretization error. To eliminate the bounded-score assumption used in previous uniformization analyses (Chen & Ying, 2024; Liang et al., 2025a), we propose a *Mask-Aware Truncated Uniformization* (MATU) method inspired by Huang et al. (2025). Under MATU, we rescale the outgoing transition rates of the reverse process according to the number of masked tokens in preceding states, naturally tighting enforcing boundedness in the discrete score estimator while preserving the unbiasedness of uniformization-based score approximation. We prove that MATU can reach the same $\epsilon$-TV convergence at a nearly $\epsilon$-free complexity, offering a significant speedup from $O(\ln(1/\epsilon))$ to $O(1 - \epsilon^2)$. The key insight is that uniformization in the masked setting explicitly identifies which tokens remain masked and require denoising, thereby avoiding the redundant denoising attempts that slow convergence in uniform discrete diffusion. Our main contributions are summarized as follows.

- We present the first rigorous theoretical analysis of typical Euler samplers for masked discrete diffusion. Achieving $\epsilon$-TV convergence requires $\tilde{O}(d^2\epsilon^{-3/2})$ discrete score evaluations, surpassing $\tau$-leaping (Liang et al., 2025a) in high-accuracy settings.

- We propose a new method called *Mask-Aware Truncated Uniformization* (MATU). Unlike simply applying uniformization to masked discrete diffusion (Liang et al., 2025a), our approach leverages a truncation on the outgoing rate, thereby removing the need for a score-bounded assumption. Moreover, our truncation is adaptive to the number of masked tokens, in contrast to Huang et al. (2025) which relies on a uniform constant, thus making full use of masked discrete diffusion properties.

- By leveraging the property that tokens cannot be unmasked multiple times, MATU significantly accelerates convergence on the discrete space $\{1, 2, \ldots, K\}^d$. Specifically, to reach $\epsilon$-TV convergence, MATU uses an expected number of discrete score calls on the order of

$$O\big(d \cdot (1 - \epsilon^2/d) + d\ln d\big).$$

Compared to uniformization-based sampler in uniform discrete diffusion (Huang et al., 2025; Liang et al., 2025a), this result improves upon the $O\big(\ln(1/\epsilon)\big)$ rate and surpasses the linear convergence limitation. Moreover, the dependence on both vocabulary size $K$ and dimension $d$ aligns with state-of-the-art performance (Zhang et al., 2024).

## 2 PRELIMINARIES

In this section, we establish the notation and setup for both forward and reverse Markov processes in general discrete diffusion models. We discuss marginal and conditional distributions, the transition rate function, neural-network-parameterized discrete scores (density ratios), and a standard training objective. We also present the commonly adopted assumption on score estimation error, which underlies many theoretical and empirical works (Zhang et al., 2024; Lou et al., 2024; Chen & Ying, 2024; Huang et al., 2025; Liang et al., 2025a). A comprehensive summary of the notation can be found in Table 2 of Appendix A.

**The forward process notations.** In this paper, we consider discrete distributions over $\mathcal{Y} = \{1, 2, \ldots, \mathrm{K}\}^d$. For any functions $f, g : \mathcal{Y} \to \mathbb{R}$, we define their inner product as

$$\langle f, g \rangle_{\mathcal{Y}} = \sum_{\boldsymbol{y} \in \mathcal{Y}} f(\boldsymbol{y}) \cdot g(\boldsymbol{y}).$$

Given a target distribution $q_*$, we define a forward Markov process $\{\mathbf{y}_t^{\rightarrow}\}_{t=0}^T$ with $q_0^{\rightarrow} = q_*$, which converges to a stationary distribution $q_{\infty}^{\rightarrow}$ as $T \to \infty$. We denote by $q_t^{\rightarrow}$ its marginal at time $t$, and use $q_{t',t}^{\rightarrow}(\boldsymbol{y}', \boldsymbol{y})$ and $q_{t'|t}^{\rightarrow}(\boldsymbol{y}'|\boldsymbol{y})$ to represent the joint and conditional distributions over times $t'$ and $t$, respectively:

$$(\mathbf{y}_{t'}^{\rightarrow}, \mathbf{y}_t^{\rightarrow}) \sim q_{t',t}^{\rightarrow}, \quad q_{t'|t}^{\rightarrow}(\boldsymbol{y}'|\boldsymbol{y}) = q_{t',t}^{\rightarrow}(\boldsymbol{y}', \boldsymbol{y})/q_t^{\rightarrow}(\boldsymbol{y}) \quad \text{for } t' > t.$$

Both masked and uniform discrete diffusion models treat this forward process as a time-homogeneous CTMC with transition rate function $R^{\rightarrow} : \mathcal{Y} \times \mathcal{Y} \to \mathbb{R}$ which denotes the instantaneous transition rate from $\boldsymbol{y}'$ to $\boldsymbol{y}$. Formally,

$$R^{\rightarrow}(\boldsymbol{y}, \boldsymbol{y}') := \lim_{\Delta t \to 0} \left[ (q_{\Delta t|0}^{\rightarrow}(\boldsymbol{y}|\boldsymbol{y}') - \delta_{\boldsymbol{y}'}(\boldsymbol{y}))/\Delta t \right] \tag{1}$$

where $\delta_{\boldsymbol{y}'}(\boldsymbol{y}) = 1$ if $\boldsymbol{y} = \boldsymbol{y}'$ and 0 otherwise. We further define $R^{\rightarrow}(\boldsymbol{y}') := \sum_{\boldsymbol{y} \neq \boldsymbol{y}'} R^{\rightarrow}(\boldsymbol{y}, \boldsymbol{y}')$ as the outgoing rate, which denotes the instantaneous transition rate from $\boldsymbol{y}'$ to all other feasible states. Under this condition, the discrete forward process follows

$$\frac{dq_{t|s}^{\rightarrow}}{dt}(\boldsymbol{y}|\boldsymbol{y}_0) = \left\langle R^{\rightarrow}(\boldsymbol{y}, \cdot), q_{t|s}^{\rightarrow}(\cdot|\boldsymbol{y}_0) \right\rangle_{\mathcal{Y}}, \quad \frac{dq_t^{\rightarrow}}{dt}(\boldsymbol{y}) = \langle R^{\rightarrow}(\boldsymbol{y}, \cdot), q_t^{\rightarrow}(\cdot) \rangle_{\mathcal{Y}}. \tag{2}$$

More details and derivation can be found in Appendix B.

**The reverse process notations.** To sample from $q_* = q_0^{\rightarrow}$, discrete diffusion models define a reverse process $\{\mathbf{y}_t^{\leftarrow}\}_{t=0}^T$ such that $\mathbf{y}_t^{\leftarrow} \sim q_t^{\leftarrow} = q_{T-t}^{\rightarrow}$ and $(\mathbf{y}_{t'}^{\leftarrow}, \mathbf{y}_t^{\leftarrow}) \sim q_{t',t}^{\leftarrow}$. By Lemma 1 (proof in Appendix B.2), this time-inhomogeneous Markov chain satisfies:

**Lemma 1** (Adapted from Eqs. (3) and (4) of Huang et al. (2025))**.** *The probability mass function $q_t^{\leftarrow}$ in the reverse process follows*

$$\frac{d q_t^{\leftarrow}}{d t}(\boldsymbol{y}) = \langle R_t^{\leftarrow}(\boldsymbol{y}, \cdot), q_t^{\leftarrow}(\cdot) \rangle_{\mathcal{Y}} \quad \text{where} \quad R_t^{\leftarrow}(\boldsymbol{y}, \boldsymbol{y}') := R^{\rightarrow}(\boldsymbol{y}', \boldsymbol{y}) \frac{q_t^{\leftarrow}(\boldsymbol{y})}{q_t^{\leftarrow}(\boldsymbol{y}')}, \tag{3}$$

*and the reverse transition function $\mathbf{R}_t^{\leftarrow}$ arises as the infinitesimal operator of the reverse process:*

$$R_t^{\leftarrow}(\boldsymbol{y}, \boldsymbol{y}') := \lim_{\Delta t \to 0} \left[ (q_{t+\Delta t|t}^{\leftarrow}(\boldsymbol{y} \mid \boldsymbol{y}') - \delta_{\boldsymbol{y}'}(\boldsymbol{y}))/\Delta t \right], \tag{4}$$

*while the outgoing rate is $R_t^{\leftarrow}(\boldsymbol{y}') = \sum_{\boldsymbol{y} \neq \boldsymbol{y}'} R_t^{\leftarrow}(\boldsymbol{y}, \boldsymbol{y}')$.*

Under this formulation, the reverse transition rate $R_t^{\leftarrow}$ depends on the forward transition rate $R^{\rightarrow}$ as well as the *discrete score*, defined as the density ratio $q_t^{\leftarrow}(\boldsymbol{y})/q_t^{\leftarrow}(\boldsymbol{y}')$. Since this ratio is generally intractable, it is approximated in practice by a neural network $\tilde{v}$:

$$\tilde{v}_{t,\boldsymbol{y}'}(\cdot) \approx v_{t,\boldsymbol{y}'}(\cdot) = q_t^{\leftarrow}(\cdot)/q_t^{\leftarrow}(\boldsymbol{y}'), \tag{5}$$

yielding an approximate reverse transition rate $\tilde{R}_t^{\leftarrow}$ via Eq. (3). To train $\tilde{v}$, one typically uses the *score entropy* loss (Lou et al., 2024; Benton et al., 2024),

$$L_{\text{SE}}(\tilde{v}) = \frac{1}{T} \int_0^T \mathbb{E}_{\mathbf{y}_t \sim q_t^{\rightarrow}} \left[ \sum_{\boldsymbol{y} \neq \mathbf{y}_t} R^{\rightarrow}(\mathbf{y}_t, \boldsymbol{y}) \, D_\phi \left( v_{T-t,\mathbf{y}_t}(\boldsymbol{y}) \big\| \tilde{v}_{T-t,\mathbf{y}_t}(\boldsymbol{y}) \right) \right] dt, \tag{6}$$

where $D_\phi \left( \cdot \| \cdot \right)$ is the Bregman divergence associated with $\phi(c) = c \ln c$. As in continuous diffusion (Chen et al., 2023), practitioners often replace $L_{\text{SE}}$ by *implicit* or *denoising score entropy* (Lou et al., 2024; Benton et al., 2024) for more tractable optimization but invariant minimum.

**General Assumptions.** To analyze both convergence properties and the computational effort required for achieving TV distance convergence in practical settings, we assume the score entropy loss will be upper-bounded. Formally:

**[A1] Score approximation error.** The discrete score $\tilde{v}_t$ obtained from Eq. (6) is well-trained, and its estimation error is small enough so that $L_{\text{SE}}(\tilde{v}) \leq \epsilon_{\text{score}}^2$.

This assumption is standard in theoretical inference research (Chen & Ying, 2024; Zhang et al., 2024; Lou et al., 2024), where it is commonly presumed that the score can be trained arbitrarily well such that $\epsilon_{\text{score}} \leq \epsilon$ for any desired $\epsilon > 0$.

## 3 THE FORWARD PROCESS OF MASKED DISCRETE DIFFUSION

In this section, we instantiate the masked discrete diffusion from the framework outlined in Section 2. We then construct a family of auxiliary distributions that approach the ideal forward marginal distribution exponentially quickly as time progresses. This construction leverages the forward transition kernel of masked discrete diffusion for any $0 < s < t < T$, and can be used as an alternative to the reverse initialization proposed by Liang et al. (2025a).

**Additional settings.** Following Ou et al. (2024), we adopt a diffusion-based language modeling framework. Our vocabulary is $\{1, 2, \ldots, \text{K}\}$, where K denotes the mask token. We aim to generate a length-$d$ sequence (sentence) $\boldsymbol{y} \in \mathcal{Y} = \{1, 2, \ldots, \text{K}\}^d$. The number of mask tokens in specific sentence $\boldsymbol{y}$ and the Hamming distance between two sentences ($\boldsymbol{y}$ and $\boldsymbol{y}'$) are denoted as

$$\text{numK}\,(\boldsymbol{y}) \coloneqq \sum_{i=1}^{d} \delta_{\text{K}}(\boldsymbol{y}_i) \quad \text{and} \quad \text{Ham}(\boldsymbol{y}, \boldsymbol{y}') = d - \sum_{i=1}^{d} \delta_{\boldsymbol{y}_i}(\boldsymbol{y}_i')$$

respectively. Generally, we suppose the mask token is never observed in target distribution:

**[A2] No mask in the target distribution.** The target distribution $q_0^{\rightarrow} = q_*: \mathcal{Y} \to \mathbb{R}$ assigns positive probability only to those sequences without any mask tokens, i.e. $q_*(\boldsymbol{y}) > 0$ if and only if $\text{numK}\,(\boldsymbol{y}) = 0$.

**Masked discrete diffusion instantiation and approximation.** We begin by specifying the absorbing forward transition rate function for masked discrete diffusion:

$$R^{\rightarrow}(\boldsymbol{y}, \boldsymbol{y}') = \begin{cases} 1 & \text{if } \text{Ham}(\boldsymbol{y}, \boldsymbol{y}') = 1 \text{ and } \boldsymbol{y}_{\text{DiffIdx}(\boldsymbol{y}, \boldsymbol{y}')} = \text{K} \\ -\sum_{i=1}^{d}\big[1 - \delta_K(\boldsymbol{y}_i)\big] & \text{if } \boldsymbol{y} = \boldsymbol{y}' \\ 0 & \text{otherwise} \end{cases}. \quad (7)$$

Here, $\text{DiffIdx}\,(\boldsymbol{y}, \boldsymbol{y}')$ denotes the single coordinate where $\boldsymbol{y}$ and $\boldsymbol{y}'$ differ. Under this transition rule, each non-masked coordinate tends to become masked at an exponential rate. Concretely, for any $0 < s < t < T$, the forward transition kernel satisfies

$$q_{t|s}^{\rightarrow}(\boldsymbol{y}|\boldsymbol{y}') = \prod_{i=1}^{d} \Big[\delta_{(\text{K},\text{K})}(\boldsymbol{y}_i, \boldsymbol{y}_i') + \big(1 - \delta_{(\text{K},\text{K})}(\boldsymbol{y}_i, \boldsymbol{y}_i')\big) \cdot \delta_0(\boldsymbol{y}_i - \boldsymbol{y}_i') \cdot e^{-(t-s)} \\ + \big(1 - \delta_{(\text{K},\text{K})}(\boldsymbol{y}_i, \boldsymbol{y}_i')\big) \cdot \delta_{\text{K}}(\boldsymbol{y}_i) \cdot (1 - e^{-(t-s)})\Big], \quad (8)$$

as shown in Lemma 8. To approximate the forward marginal distribution $q_t^{\rightarrow}$ at time $t$, we exploit this exponential decay by modeling each non-mask coordinate under a uniform distribution and masking coordinates at a constant rate. Specifically, we define

$$\tilde{q}_t(\boldsymbol{y}) \propto \prod_{i=1}^{d} \exp\Big(-t \cdot \big[1 - \delta_{\text{K}}(\boldsymbol{y}_i)\big]\Big) = \exp\Big(-t \cdot \big[d - \text{numK}\,(\boldsymbol{y})\big]\Big). \quad (9)$$

so that $\tilde{q}_t$ factorizes over coordinates and is straightforward to sample from. Moreover, as established in Lemma 2, the KL divergence between $q_t^{\rightarrow}$ and $\tilde{q}_t$ decreases exponentially with $t$.

**Lemma 2** (Exponentially decreasing KL divergence between $q_t^{\rightarrow}$ and $\tilde{q}_t$). *Suppose the CTMC* $\{\mathbf{y}_t^{\rightarrow}\}_{t=0}^T$ *has transition rates* $R^{\rightarrow}$ *from Eq.* (7)*, with* $\mathbf{y}_t^{\rightarrow} \sim q_t^{\rightarrow}$. *Let* $\tilde{q}_t$ *be the approximation of* $q_t^{\rightarrow}$ *defined by Eq.* (9)*. Then,*

$$\mathrm{KL}\left(q_t^{\rightarrow}\,\big\|\,\tilde{q}_t\right) \;\leq\; (1 + e^{-t})^d - 1.$$

*Consequently, to ensure* $\mathrm{KL}\left(q_t^{\rightarrow}\,\big\|\,\tilde{q}_t\right) \leq \epsilon$, *it suffices to choose* $t \geq \ln\!\big(4d/\epsilon\big)$.

From Lemma 2, the running time $T$ required for $\tilde{q}_T$ to approximate $q_T^{\rightarrow}$ falls on the order of $\mathcal{O}(\ln(d/\epsilon))$. It precisely matches the forward mixing time for uniform discrete diffusion (Chen & Ying, 2024; Zhang et al., 2024; Huang et al., 2025) and continuous diffusion (Chen et al., 2023) converging to their stationary distributions. Although the final results exhibit a similar convergence rate, the underlying analytical techniques differ substantially because the one-hot stationary distribution of masked discrete diffusion does not satisfy the modified log-Sobolev condition. Further technical details are deferred to Appendix B.3.

# 4 EULER SAMPLER IN MASKED DISCRETE DIFFUSION

This section first introduces the Euler sampler in masked discrete diffusion, widely used for its parallel coordinate updates when reverse transition can be factorized coordinate-wise. We then extend it to handle more general reverse marginals with unknown correlations, and show how to control accumulative errors by introducing the exponential integrator as the auxiliary process. Finally, we provide convergence and complexity guarantees for achieving $\epsilon$–TV convergence.

**Typical Euler samplers and their extensions.** Euler-type samplers have become increasingly popular in empirical studies (Lou et al., 2024; Ou et al., 2024) because their parallel-friendly updates often run faster than traditional auto-regressive models. Let $\{\hat{\mathbf{y}}_t\}_{t=0}^T$ denote the practical reverse process, whose marginal, joint, and conditional distributions satisfy:

$$\hat{\mathbf{y}} \sim \hat{q}_t, \quad (\hat{\mathbf{y}}_{t'}, \hat{\mathbf{y}}_t) \sim \hat{q}_{t',t}, \quad \text{and} \quad \hat{q}_{t'|t}(\boldsymbol{y}'|\boldsymbol{y}) = \hat{q}_{t',t}(\boldsymbol{y}',\boldsymbol{y})/\hat{q}_t(\boldsymbol{y}) \quad \text{where} \quad t' \geq t.$$

A key assumption is that the reverse transition for each coordinate is conditionally independent:

$$\hat{q}_{t+\Delta t|t}(\boldsymbol{y}'|\boldsymbol{y}) \propto \prod_{i=1}^d \hat{q}_{t+\Delta t|t}^{(i)}(\boldsymbol{y}[\{i\} \rightarrow \{\boldsymbol{y}'_i\}]|\boldsymbol{y}), \tag{10}$$

where the token revision function

$$\boldsymbol{y}[S\colon \rightarrow Y' \subseteq \mathcal{Y}^{|S|}] = \sum_{i=1}^d \boldsymbol{e}_i \cdot \mathbf{1}[i \notin S] \cdot \boldsymbol{y}_i + \sum_{j=1}^{|S|} \boldsymbol{e}_{s_j} \cdot Y'_j$$

indicates that the coordinates of $\boldsymbol{y}$ indexed by the set $S$ are replaced by the corresponding values in $Y'$. Then, each non-masked token can be updated independently in the reverse-time direction. Specifically, by discretizing Eq. (4) from Lemma 1, the update for the $i$th coordinate takes the form:

$$\hat{q}_{t+h|t}^{(i)}(\boldsymbol{y}[\{i\} \rightarrow \{\boldsymbol{y}'_i\}]|\boldsymbol{y}) = \delta_{\boldsymbol{y}_i}(\boldsymbol{y}'_i) + h \cdot R^{\rightarrow}(\boldsymbol{y}, \boldsymbol{y}[\{i\} \rightarrow \{\boldsymbol{y}'_i\}]) \cdot \tilde{v}_{t,\boldsymbol{y}}(\boldsymbol{y}[\{i\} \rightarrow \{\boldsymbol{y}'_i\}]).$$

Since $\mathrm{Ham}(\boldsymbol{y}, \boldsymbol{y}[\{i\} \rightarrow \boldsymbol{y}'_i]) = 1$, the definition of $R^{\rightarrow}$ in Eq. (7) ensures that $R^{\rightarrow}\big(\boldsymbol{y}, \boldsymbol{y}[\{i\} \rightarrow \boldsymbol{y}'_i]\big) \neq 0$. Hence, $\hat{q}_{t+h|t}^{(i)}(\boldsymbol{y}[\{i\} \rightarrow k]|\boldsymbol{y})$ for any non-mask token $k \neq \mathrm{K}$, enabling all coordinates to be updated in parallel.

However, if the assumption in Eq. (10) does not hold, parallel updates become invalid. A practical alternative is to discretize Eq. (4) jointly, leading to the sequential update:

$$\hat{q}_{t+h|t}(\boldsymbol{y}'|\boldsymbol{y}) \propto \delta_{\boldsymbol{y}}(\boldsymbol{y}') + h \cdot \tilde{R}_t(\boldsymbol{y}',\boldsymbol{y}) = \delta_{\boldsymbol{y}}(\boldsymbol{y}') + h \cdot R^{\rightarrow}(\boldsymbol{y},\boldsymbol{y}') \cdot \tilde{v}_{t,\boldsymbol{y}}(\boldsymbol{y}') \tag{11}$$

where $\hat{q}_{t+\Delta t|t}(\boldsymbol{y}' \mid \boldsymbol{y}) \neq 0$ only if $R^{\rightarrow}(\boldsymbol{y},\boldsymbol{y}') \neq 0$, which implies $\mathrm{Ham}(\boldsymbol{y},\boldsymbol{y}') = 1$ (see Eq. (7)). Consequently, at most one masked token could be denoised per update. In the subsequent analysis, we consider the Euler sampler using Eq. (11) in this more general setting.

**Theoretical results.** For the Euler sampler, the construction of the training loss, e.g., *denoising score entropy*, will be related to the step size $h$ and share the same minimum with

$$L_{\text{DisSE}}(\tilde{v}) := \frac{1}{T-\delta} \sum_{k=0}^{n-1} \int_{kh}^{(k+1)h} \mathbb{E}_{\mathbf{y}_t \sim q_t^{\leftarrow}} \left[ \sum_{\boldsymbol{y} \neq \mathbf{y}_t} R^{\rightarrow}(\mathbf{y}_t, \boldsymbol{y}) D_\phi \left( v_{kh,\mathbf{y}_t}(\boldsymbol{y}) || \tilde{v}_{kh,\mathbf{y}_t}(\boldsymbol{y}) \right) \right] \mathrm{d}t.$$

Correspondingly, to suppose the neural score estimator well approximates the discrete score only requires the following score estimation assumption, milder than Assumption **[A1]**, i.e.,

**[A1]- Score approximation error.** The discrete score $\tilde{v}_t$ obtained from Eq. (6) is well-trained, and its estimation error is small enough so that $L_{\text{DisSE}}(\tilde{v}) \leq \epsilon_{\text{score}}^2$.

Then, we summarize the convergence and complexity of Euler sampler (with proof in Section C.1).

**Theorem 1.** *Suppose Assumption **[A1]-**, **[A2]** and Assumption 2 of Liang et al. (2025a) hold, implement Euler sampler with Eq.* (11)*, if we require*

$$T = \ln(4d/\epsilon^2), \quad h \lesssim \min \left\{ \frac{\varepsilon}{K^2 d^2 \log(d/\varepsilon)}, \frac{\varepsilon^{\frac{3}{2}}}{d\sqrt{\log(d/\varepsilon)}} \right\}, \quad and \quad \epsilon_{score} \leq \tilde{o}(\epsilon^2/d),$$

*the Euler sampler will achieve* $\text{TV}\,(p_*, \hat{p}) \leq 2\epsilon$ *by requiring iterations to at an* $\tilde{O}(d^2 \epsilon^{-3/2})$ *level.*

Compared to the $\tau$-leaping method analyzed in Liang et al. (2025a), Euler-based approaches can be more effective in high-accuracy settings (e.g., $\epsilon \leq d^{-2}$). However, establishing a clear advantage over uniform discrete diffusion remains challenging. Due to time-discretization errors in discrete score estimation, Euler-based inference incurs polynomial complexity in both the dimensionality $d$ and the error tolerance $\epsilon$, which is still be worse than that in uniformization-based samplers.

# 5 Truncated Uniformization in Masked Discrete Diffusion

This section extends the truncated uniformization sampler of Huang et al. (2025) to masked discrete diffusion. We first revisit the core principle of unbiased reverse process simulation via uniformization. Next, we show that the expected complexity of uniformization-based inference depends critically on the outgoing rates of the reverse transition, and that masked discrete diffusion naturally offers smaller outgoing rates than its uniform counterpart, leading to faster convergence. We then introduce *Mask-Aware Truncated Uniformization* (MATU), which rescales the outgoing rates to eliminate the bounded-score assumption while preserving unbiased reverse process simulation. Finally, we provide theoretical results on MATU's convergence and computational complexity, and compare these findings with existing approaches in the literature.

**Uniformization and the expected number of discrete score calls.** Consider a time-dependent reverse transition rate $R_t^{\leftarrow}$ defined over the interval $[a, b]$. The evolution of the ideal reverse process for any $\boldsymbol{y}, \boldsymbol{y}'$ can be described by

$$q_{t+\Delta t|t}^{\leftarrow}(\boldsymbol{y}' \mid \boldsymbol{y}) = \begin{cases} \Delta t \cdot R_t^{\leftarrow}(\boldsymbol{y}', \boldsymbol{y}), & \boldsymbol{y}' \neq \boldsymbol{y}, \\ 1 - \Delta t \cdot R_t^{\leftarrow}(\boldsymbol{y}), & \boldsymbol{y}' = \boldsymbol{y}, \end{cases} \quad \text{as } \Delta t \to 0, \tag{12}$$

following Eq. (4). If the total outgoing rate–denoting the instantaneous transition rate from $\boldsymbol{y}$ to all other feasible states–is uniformly bounded by some $\beta$, i.e.,

$$R_t^{\leftarrow}(\boldsymbol{y}) = \sum_{\boldsymbol{y}' \neq \boldsymbol{y}} R_t^{\leftarrow}(\boldsymbol{y}', \boldsymbol{y}) \leq \beta_t \leq \max_{t \in [a,b]} \beta_t = \beta, \tag{13}$$

then with probability $1 - \Delta t \cdot \beta$, the particle remains in the same state in each infinitesimal time step, thus requiring no additional score computation.

Based on this observation, the standard *uniformization* method (van Dijk, 1992; van Dijk et al., 2018; Chen & Ying, 2024) simulates the reverse dynamics over $[a, b]$ by iterating the following two-step procedure in the limit $\Delta t \to 0$:

1. Sample whether a transition occurs with probability $\Delta t \cdot \beta$.

2. If a transition occurs, move $\mathbf{y}_t^{\leftarrow}$ from $\boldsymbol{y}$ to $\boldsymbol{y}'$ with probability

$$M_t(\boldsymbol{y}' \mid \boldsymbol{y}) \;=\; \begin{cases} \beta^{-1} R_t^{\leftarrow}(\boldsymbol{y}', \boldsymbol{y}), & \boldsymbol{y}' \neq \boldsymbol{y}, \\ 1 - \beta^{-1} R_t^{\leftarrow}(\boldsymbol{y}), & \text{otherwise.} \end{cases} \tag{14}$$

Under this update scheme, the reverse transitions of uniformization will be equivalent to Eq. (12) exactly and introduce no time-discretization error (see Appendix D.2 for details). Moreover, since the number of transitions (and hence the number of discrete score computations) over $[a, b]$ follows a Poisson distribution with mean $\beta \cdot (b - a)$, any tighter bound on $R_t^{\leftarrow}(\boldsymbol{y})$ reduces $\beta$ and thereby lowers the expected inference complexity.

**The comparison of computational complexity and outgoing rate.** By the previous discussion of uniformization, the expected number of discrete score calls over the time interval $[0, T]$ can be approximated by

$$\sum_{w=1}^{W} \max_{t \in [t_{w-1}, t_w]} \beta_t \cdot (t_w - t_{w-1}) \overset{W \to \infty}{\approx} \int_{t=0}^{T} \beta_t \mathrm{d}t, \tag{15}$$

where $[t_0, t_1, \ldots, t_W]$ is a partition of $[0, T]$. In uniform discrete diffusion, Chen & Ying (2024); Huang et al. (2025) show that the ideal reverse process satisfies

$$\beta_t \coloneqq 2K \cdot d \cdot \max\{1, (T - t)^{-1}\} \leq \beta \coloneqq 2K \cdot d \cdot \max\{1, (T - b)^{-1}\} \quad \forall\, t \in [a, b], \tag{16}$$

providing a uniform upper bound on the total outgoing rate $R_t^{\leftarrow}(\boldsymbol{y})$.

For *masked* discrete diffusion, Lemma 3 (with proof in Appendix D.1) shows that the outgoing rate can be bounded instead by

**Lemma 3** (Bound of the outgoing rate). *Consider a CTMC whose transition rate function $R^{\rightarrow}$ is defined as Eq.* (7). *Then, for any $\boldsymbol{y}$, the reverse transition rate function satisfies*

$$\sum_{\boldsymbol{y}' \neq \boldsymbol{y}} R_t^{\leftarrow}(\boldsymbol{y}', \boldsymbol{y}) = R_t^{\leftarrow}(\boldsymbol{y}) \leq \beta_t(\boldsymbol{y}) \coloneqq \frac{\mathrm{numK}\,(\boldsymbol{y}) \cdot K}{e^{(T-t)} - 1}. \tag{17}$$

Compared to (16), this bound explicitly depends on $\mathrm{numK}\,(\boldsymbol{y})$, the number of mask tokens in $\boldsymbol{y}$. Since $\mathrm{numK}\,(\boldsymbol{y}) \leq d$, it is strictly smaller than the uniform bound in (16). Furthermore, $\mathrm{numK}\,(\boldsymbol{y})$ decreases monotonically as the reverse process proceeds, which progressively enlarges the gap in outgoing rate between masked and uniform discrete diffusion. Because a lower outgoing rate implies fewer expected discrete score evaluations for each time $t$, masked discrete diffusion can be significantly more computationally efficient.

From an empirical perspective, a central observation is: *during inference, masked discrete diffusion only updates (denoises) masked tokens, whereas uniform discrete diffusion attempts to re-denoise tokens that have already been denoised.* Hence, in masked discrete diffusion, particles are more likely to remain unchanged at each step, leading to a smaller outgoing rate (and thus smaller $\beta_t$) over $[0, T]$. Consequently, fewer discrete score evaluations are required, underscoring the computational advantages of masked compared to uniform discrete diffusion.

**Mask-aware truncation and algorithm proposal.** In practice, we approximate the reverse transition rate $R_t^{\leftarrow}(\boldsymbol{y}', \boldsymbol{y})$ by a learned neural score $\tilde{v}_{t,\boldsymbol{y}}(\boldsymbol{y}')$, yielding

$$\tilde{R}_t(\boldsymbol{y}', \boldsymbol{y}) \;=\; R^{\rightarrow}(\boldsymbol{y}, \boldsymbol{y}')\, \tilde{v}_{t,\boldsymbol{y}}(\boldsymbol{y}'),$$

as dictated by Lemma 1 and Eq. (5). Because $\tilde{v}$ is a learned estimator, the outgoing rate $\tilde{R}_t(\boldsymbol{y})$ may have no explicit upper bounds, complicating control over the expected number of discrete score evaluations. To mitigate unbounded transition rates, prior work typically imposes a bounded-score assumption on $\tilde{R}_t(\boldsymbol{y})$, restricting it to remain below a fixed constant (Liang et al., 2025a) or to grow as a function of the inference time (Chen & Ying, 2024). However, such assumptions can severely impact inference efficiency because the chosen upper bound $\beta$ directly governs Step 2 of uniformization, as described in Eq. (14). When $\beta$ is unknown, it can be treated as a hyperparameter.

---

**Algorithm 1** MASK-AWARE TRUNCATED UNIFORMIZATION (MATU)

---

1: **Input:** Total time $T$, a time partition $0 = t_0 < \ldots < t_W = T - \delta$, parameters $\beta_{t_1}, \ldots, \beta_{t_W}$ set as Eq. (17), a reverse transition rate function $\hat{R}_t^{\leftarrow}$ obtained by the learnt score function $\tilde{v}_{t,\boldsymbol{y}'}(\cdot)$.
2: Draw an initial sample $\hat{\mathbf{y}}_{t_0} = [\mathrm{K}, \mathrm{K}, \ldots, \mathrm{K}]$.
3: **for** $w = 1$ **to** $W$ **do**
4:      Choose $\beta_{t_w} = K \cdot \mathrm{numK}(\hat{\mathbf{y}}_{t_{w-1}})/(e^{T - t_w} - 1)$
5:      Draw $N \sim \mathrm{Poisson}(\beta_{t_w}(t_w - t_{w-1}))$;
6:      Sample $N$ points i.i.d. uniformly from $[t_{w-1}, t_w]$ and sort them as $\tau_1 < \tau_2 < \ldots < \tau_N$;
7:      Set $\mathbf{z}_0 = \hat{\mathbf{y}}_{t_{w-1}}$;
8:      **for** $n = 1$ **to** $N$ **do**
9:          Find the index set $\mathcal{M}$ of [MASK] token appeared in random vector $\mathbf{z}_{n-1}$
10:          For any $i \in \mathcal{M}$ and $k \in \{1, 2, \ldots, K - 1\}$, update $\mathbf{z}_{n-1}$ with

$$\mathbf{z}_n = \begin{cases} \mathbf{z}_{n-1}[\mathbf{z}_i: K \to k] & w.p. \; \beta_{t_w}^{-1} \cdot \hat{R}_{\tau_n, \mathbf{z}_0}(\mathbf{z}_{n-1}[\mathbf{z}_i: K \to k], \mathbf{z}_{n-1}), \\ \mathbf{z}_{n-1}, & w.p. \; 1 - \beta_{t_w}^{-1} \cdot \hat{R}_{\tau_n, \mathbf{z}_0}(\mathbf{z}_{n-1}). \end{cases}$$

11:      **end for**
12:      Set $\hat{\mathbf{y}}_{t_w} = \mathbf{z}_N$.
13: **end for**
14: **return** $\hat{\mathbf{y}}_{t_W}$.

---

Setting $\beta$ too small may yield an infeasible probability $1 - \beta^{-1}\tilde{R}_t(\boldsymbol{y}) < 0$, forcing the algorithm to fail; setting it too large preserves feasibility but inflates complexity in direct proportion to $\beta$. Thus, tightening this bounding scheme is crucial for balancing both correctness and computational efficiency in uniformization-based inference.

Motivated by Huang et al. (2025), we propose a *mask-aware truncation* scheme to rescale the practical outgoing rate $\tilde{R}_t(\boldsymbol{y}', \boldsymbol{y})$. This ensures that the non time-discretization property is preserved without additional cost, even when $\tilde{R}_t(\boldsymbol{y})$ becomes large. Specifically, consider simulating the reverse process over the ($w$-th) time segment $[t_{w-1}, t_w]$, assuming the state at time $t_{w-1}$ is $\hat{\mathbf{y}}_{t_{w-1}} = \boldsymbol{y}_{t_{w-1}}$. Following from the monotonicity of $(e^{T-t} - 1)^{-1}$ and $\mathrm{numK}(\hat{\mathbf{y}}_t)$ in Lemma 3, the mask-aware truncation is chosen as $\beta_{t_w}(\boldsymbol{y}_{t_{w-1}})$, then we set

$$\hat{R}_{t,\boldsymbol{y}_{t_{w-1}}}(\boldsymbol{y}, \boldsymbol{y}') = \begin{cases} \tilde{R}_t(\boldsymbol{y}, \boldsymbol{y}') \, \beta_{t_w}(\boldsymbol{y}_{t_{w-1}})/\tilde{R}_t(\boldsymbol{y}'), & \text{if } \tilde{R}_t(\boldsymbol{y}') > \beta_{t_w}(\boldsymbol{y}_{t_{w-1}}), \\ \tilde{R}_t(\boldsymbol{y}, \boldsymbol{y}'), & \text{otherwise,} \end{cases} \quad \forall \boldsymbol{y}' \neq \boldsymbol{y}, \quad (18)$$

and

$$\hat{R}_{t,\boldsymbol{y}_{t_{w-1}}}(\boldsymbol{y}', \boldsymbol{y}') = -\sum_{\boldsymbol{y} \neq \boldsymbol{y}'} \hat{R}_{t,\boldsymbol{y}_{t_{w-1}}}(\boldsymbol{y}, \boldsymbol{y}'). \quad (19)$$

With these truncations, the corrected outgoing rate will be definitely upper bounded by $\beta_{t_w}(\boldsymbol{y}_{t_{w-1}})$. Then, we obtain a practical and efficient inference algorithm, summarized in Alg. 1.

**Theoretical results.** We summarize the convergence and complexity of Algorithm 1 for approximating $q_*$ in Theorem 2 (proved in Appendices D.2 and D.3).

**Theorem 2** (Combination of Theorem 3 and Theorem 4). *Suppose Assumption [A1] and [A2] hold, for Alg. 1, if we require*

$$T = \ln(4d/\epsilon^2), \quad \delta \leq d^{-1}\epsilon, \quad \epsilon_{score} \leq T^{-1/2}\epsilon, \quad \epsilon < 1,$$

*and the partition of the reverse process satisfies*

$$\eta = \epsilon/2d, \quad W = (T - \delta)/\eta, \quad t_0 = 0, \quad t_W = T - \delta, \quad t_w - t_{w-1} = \eta \quad \forall w \in \{1, 2, \ldots W\}$$

*the expectation of iteration/score estimation complexity of Alg. 1 will be upper bounded by*

$$2K(d - \epsilon^2/4) + 12Kd\ln d \quad (20)$$

*to achieve* $\mathrm{TV}(p_*, \hat{p}) \leq 2\epsilon$ *where* $\hat{p}$ *denotes the underlying distribution of generated samples.*

Table 1: Comparison with prior works simulating reverse particle SDEs, where **[A3]** denotes the bounded-score assumption used in Chen & Ying (2024) and **[A3]+** denotes the bounded-score assumption used in Liang et al. (2025a) which is a little bit stronger than **[A3]** due to the time-invariant requirement. All complexities are on TV convergence (or TV convergence dedued from KL convergence via Pinsker's inequality, e.g., Ren et al. (2024)), which are achieved by assuming $\epsilon_{\text{score}} = \tilde{o}(\epsilon)$ and setting early-stopping parameters $\delta = \epsilon/d$. Besides, the complexity presented by $\tilde{O}(\cdot)$ means the ln dependencies are omitted.

| Results | Forward Type | Inference Sampler | Assumptions | Complexity |
|---|---|---|---|---|
| Zhang et al. (2024) | Uniformed | Exponential Integrator | **[A1]**, **[A3]** | $\tilde{\mathcal{O}}(d^{5/3}\epsilon^{-2})$ |
| Ren et al. (2024) | Uniformed | $\tau$-leaping | **[A1]**,**[A3]** | $\tilde{O}(d^2\epsilon^{-2})$ |
| Chen & Ying (2024) | Uniformed | Uniformization | **[A1]**,**[A3]** | $O(d\ln(d/\epsilon))$ |
| Huang et al. (2025) | Uniformed | Truncated Uniformization | **[A1]** | $O(d\ln(d/\epsilon))$ |
| Theorem 1 | Masked | Typical Euler | **[A1]**,**[A2]**,**[A3]+** | $\tilde{O}(d^2\epsilon^{-3/2})$ |
| Liang et al. (2025a) | Masked | $\tau$-leaping | **[A1]**,**[A2]**,**[A3]+** | $O(d\epsilon^{-2})$ |
| Liang et al. (2025a) | Masked | Uniformization | **[A1]**,**[A2]**,**[A3]** | $O(d\ln(d/\epsilon))$ |
| Theorem 2 | Masked | MATU | **[A1]**,**[A2]** | $O(d\ln d)$ |

From the above theorem, Eq. (20) might appear to enable exact inference by setting $\epsilon = 0$. However, this would require infinite mixing time $T$, perfect score estimates ($\epsilon_{\text{score}} = 0$), and infinitely many intervals $W$, which is infeasible. Meanwhile, although each interval has length $\eta = \epsilon/(2d)$—leading to $\text{poly}(d/\epsilon)$ intervals in the reverse process—the total discrete score calls remain nearly independent of $\epsilon$, since many intervals involve no state transitions (see Eq. (15)). Thus, small intervals are used primarily to match the accurate outgoing rate upper bound, without inflating complexity.

Then, We provide a complexity comparison in Table 3. MATU achieves a SOTA for both the $\epsilon$-free complexity and the assumption without bounded-score estimator. Compared with existing uniformization-based method, Alg 1 achieves an $O\big(\ln(1/\epsilon)\big)$ speedup, primarily because each token is denoised at most once in masked diffusion, whereas uniform diffusion renoises tokens multiple times. Formally, masked diffusion leverages the monotonic decrease of masked tokens, which cancels the growing outgoing rate:

$$\mathbb{E}\left[\sum_{w=1}^{W}\beta_{t_w}(\hat{\mathbf{y}}_{t_{w-1}})\cdot(t_w-t_{w-1})\right] \approx \sum_{w=1}^{W}\mathbb{E}[\text{numK}(\mathbf{y}_{T-t_{w-1}}^{\rightarrow})]\cdot K\cdot\frac{e^{-(T-t_w)}}{1-e^{-(T-t_w)}}\cdot\eta$$

$$= \sum_{w=1}^{W}d\cdot\underbrace{(1-e^{-(T-t_{w-1})})}_{\text{decreasing factor}}\cdot K\cdot\underbrace{(1-e^{-(T-t_w)})^{-1}}_{\text{increasing factor}}\cdot e^{-(T-t_w)}\cdot\eta \leq CKd\cdot\sum_{w=1}^{W}e^{-(T-t_w)}\cdot\eta,$$

where the factor $e^{-(T-t_w)}$ keeps complexity low. In uniform diffusion, the same factor remains but grows with $1/(T-t_w)$, leading to a higher order overall:

$$\mathbb{E}\left[\sum_{w=1}^{W}\beta_{t_w}\cdot(t_w-t_{w-1})\right] \lesssim CKd\cdot\sum_{w=1}^{W}\max\{1,(T-t_w)^{-1}\}\cdot\eta.$$

Since the integral $\int(1/t)\,dt$ diverges more quickly than $\int e^{-t}\,dt$, masked diffusion achieves lower inference complexity than uniform diffusion.

# 6 RELATED WORK

Most recently, the impressive empirical performance of discrete diffusion models (DDMs) has sparked a proliferation of theoretical investigations aiming to elucidate DDMs from various perspectives.

**The Sample Complexity.** For example, Srikanth et al. (2025) develops a theoretical framework for discrete-state diffusion models and presents the first rigorous sample-complexity bound of $\tilde{O}(\epsilon^{-2})$ under practical assumptions about neural network training. By pursuing a structured error decomposition, the authors illustrate how approximation, statistical, optimization, and clipping constraints

jointly contribute to the total complexity, furnishing dimension-free insights for training discrete-state diffusion models. Meanwhile, Wan et al. (2025) conducts the first non-asymptotic error analysis for discrete flow models on finite state spaces. By proposing a novel Girsanov-type theorem and bounding the KL divergence between two continuous-time Markov chains (CTMCs) with distinct transition rates, they rigorously decompose the transition-rate estimation error (including stochastic, approximation, and early-stopping components). Employing uniformization for sampling, the authors derive an upper bound on the distribution error that avoids any additional discretization error, thereby advancing the theory of discrete flow models beyond existing analyses of discrete diffusion.

**The Inference Complexity.** In addition to quantifying error tolerance and dimensional dependencies, Liang et al. (2025b) introduces a differential-inequality–based analysis for discrete diffusion models that eliminates the strong regularity assumptions required by Girsanov-based methods, reducing the convergence rates for $\tau$-leaping from quadratic to linear in vocabulary size. Furthermore, Zheng et al. (2024) proposes the first-hitting-sampler (FHS) as a way to exactly simulate the reverse process by analytically sampling both the transition time and position. However, when discrete scores are parameterized by a time-dependent neural network (see Eq. 5), the uniform procedure for selecting the next unmasking position can introduce inference errors beyond those stemming from score estimation alone.

The key issue is that, although each masked position may share the same unmasking probability under the ideal reverse transition $q_t^\leftarrow$, this property may fail once the reverse process is learned. In particular, there can exist $i \neq j$ such that

$$\sum_{y',s.t.\ \mathrm{Ham}(y',y)=1,\mathrm{DffIdx}(y',y)=i,y_i=K} s_{\theta,t,y}(y') \neq \sum_{y',s.t.\ \mathrm{Ham}(y',y)=1,\mathrm{DffIdx}(y',y)=j,y_j=K} s_{\theta,t,y}(y').$$

so that uniformly choosing the next position to unmask biases the simulation of the learned reverse process, causing additional inference errors. Although this bias vanishes for time-independent discrete parameterizations (Ou et al., 2024), such as in Devlin et al. (2019); Chang et al. (2022); Ghazvininejad et al. (2019), where

$$q_t^\leftarrow(\boldsymbol{y}')/q_t^\leftarrow(\boldsymbol{y}) = \frac{e^{-t}}{1-e^{-t}} \cdot q_0^\rightarrow(\boldsymbol{y}_i'||\boldsymbol{y}^{UM}) \approx \frac{e^{-t}}{1-e^{-t}} \cdot p_{\theta,\boldsymbol{y}}(\boldsymbol{y}'), \tag{21}$$

the strong constraints, i.e.,

$$\sum_{\boldsymbol{y}',s.t.\ \mathrm{Ham}(\boldsymbol{y}',y)=1,\mathrm{DffIdx}(\boldsymbol{y}',\boldsymbol{y})=i,\boldsymbol{y}_i=K} p_{\theta,\boldsymbol{y}}(\boldsymbol{y}') = 1 = \sum q_0(\boldsymbol{y}_i'||\boldsymbol{y}^{UM}),$$

ensure that every position has identical transition rates. Nevertheless, FHS Zheng et al. (2024) provides no detailed or rigorous proof of its unbiasedness in this setting. In Theorem 7, we close this theoretical gap by coupling the trajectories of FHS and MATU, thereby controlling their differences and formally establishing FHS's unbiasedness.

## 7 CONCLUSION

In this paper, we provide a rigorous analysis of masked discrete diffusion. Differ from the analysis of uniform discrete diffusion, we show how to manage the initial KL blow-up and control the reverse-process KL divergence without relying on Girsanov theory. Building on this framework, we prove that Euler-type samplers TV converge in $\tilde{O}(d^2\epsilon^{-3/2})$. We further introduce a mask-aware truncated uniformization sampler that removes the $\ln(1/\epsilon)$ factor, achieving nearly $\epsilon$-free complexity. This acceleration aligns with the practical observation that masked diffusion denoises each masked token only once, whereas uniform diffusion repeatedly re-denoises already denoised tokens. Our results not only establish the first rigorous foundations for masked discrete diffusion but also explain why masked diffusion significantly reduces overhead in practice, opening avenues for more efficient text generation and advanced masked sampling techniques.

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

CONTENTS

## A  NOTATION SUMMARY

We summarize all notations used in the main paper and appendix in Table 2.

Table 2: Summary of key notations used in the paper.

| Symbol | Description |
|---|---|
| $q_*$ | Discrete distribution on $\mathcal{Y} = \{1, 2, \ldots, K\}^d$ |
| $\mathbf{y}_t^{\rightarrow}$ | Forward-time CTMC on $\mathcal{Y}$ |
| $q_t^{\rightarrow}$ | Marginal distribution of forward process at time $t$, i.e., $\mathbf{y}_t^{\rightarrow} \sim q_t^{\rightarrow}$ |
| $q_{t',t}^{\rightarrow}$ | Joint distribution of $(\mathbf{y}_{t'}^{\rightarrow}, \mathbf{y}_t^{\rightarrow})$ |
| $\tilde{q}_t$ | Aapproximation of $q_t^{\rightarrow}$ constructing the reverse initialization, Eq. (9) |
| $q_{t'|t}^{\rightarrow}(\boldsymbol{y}'|\boldsymbol{y})$ | Conditional transition probability in forward process, Eq. (37) |
| $\mathbf{y}_t^{\leftarrow}$ | Reverse-time CTMC defined by $q_t^{\leftarrow} := q_{T-t}^{\rightarrow}$, $\mathbf{y}_t^{\leftarrow} \sim q_t^{\leftarrow}$ |
| $q_t^{\leftarrow}$ | Marginal distribution of reverse process at time $t$, $q_t^{\leftarrow} = q_{T-t}^{\rightarrow}$ |
| $q_{t',t}^{\leftarrow}$ | Joint distribution of $(\mathbf{y}_{t'}^{\leftarrow}, \mathbf{y}_t^{\leftarrow})$ |
| $q_{t'|t}^{\leftarrow}(\boldsymbol{y}'|\boldsymbol{y})$ | Conditional transition probability of the ideal reverse process |
| $\hat{q}_t$ | Marginal distribution of reverse process at time $t$ implemented by Alg. 1 |
| $\hat{q}_{t',t}$ | Joint distribution of $(\hat{\mathbf{y}}_{t'}, \hat{\mathbf{y}}_t)$ |
| $\hat{q}_{t'|t}(\boldsymbol{y}'|\boldsymbol{y})$ | Conditional transition probability of the ideal reverse process |
| $R^{\rightarrow}(\boldsymbol{y}, \boldsymbol{y}')$ | Forward transition rate, i.e., Eq. (7), from state $\boldsymbol{y}'$ to $\boldsymbol{y}$. This follows the ordering of the conditional distribution $p(\boldsymbol{y}\|\boldsymbol{y}')$, which is the *transpose* of the convention used in some other works. |
| $R_t^{\leftarrow}(\boldsymbol{y}, \boldsymbol{y}')$ | Reverse transition rate at time $t$ from state $\boldsymbol{y}'$ to $\boldsymbol{y}$, $R_t^{\leftarrow}(\boldsymbol{y}, \boldsymbol{y}') := R^{\rightarrow}(\boldsymbol{y}', \boldsymbol{y}) \cdot \frac{q_t^{\leftarrow}(\boldsymbol{y})}{q_t^{\leftarrow}(\boldsymbol{y}')}$, Eq. (3) |
| $\tilde{R}_t(\boldsymbol{y}, \boldsymbol{y}')$ | Estimated reverse transition rate using the learned density ratio, $\tilde{R}_t(\boldsymbol{y}, \boldsymbol{y}') = R^{\rightarrow}(\boldsymbol{y}', \boldsymbol{y}) \cdot \tilde{v}_{t,\boldsymbol{y}'}(\boldsymbol{y})$, Eq. (6) |
| $\hat{R}_t(\cdot, \cdot)$ | Truncated version of $\tilde{R}_t(\cdot, \cdot)$ with threshold $\beta_t$, Eq. (18) |
| $R_t^{\leftarrow}(\boldsymbol{y}),\ \tilde{R}_t(\boldsymbol{y}),\ \hat{R}_t(\boldsymbol{y})$ | Total reverse transition rate out of state $\boldsymbol{y}$ for each rate type, defined as $R(\boldsymbol{y}) := \sum_{\boldsymbol{y}' \neq \boldsymbol{y}} R(\boldsymbol{y}', \boldsymbol{y})$ with $R \in \{R_t^{\leftarrow},\ \tilde{R}_t,\ \hat{R}_t\}$ |
| $\beta_t$ | Upper bound on $R_t^{\leftarrow}(\boldsymbol{y})$, $\beta_t = \mathrm{numK}(\boldsymbol{y}) \cdot K/(T-t)$, Eq. (17) |
| $v_{t,\boldsymbol{y}'}(\boldsymbol{y})$ | Density ratio $q_t^{\leftarrow}(\boldsymbol{y})/q_t^{\leftarrow}(\boldsymbol{y}')$ |
| $\tilde{v}_{t,\boldsymbol{y}'}(\boldsymbol{y})$ | Learned approximation to $v_{t,\boldsymbol{y}'}(\boldsymbol{y}) = q_t^{\leftarrow}(\boldsymbol{y})/q_t^{\leftarrow}(\boldsymbol{y}')$ |
| $\mathrm{numK}(\cdot)$ | The number of [MASK] token (or token $K$) in a vector. |
| $L_{\mathrm{SE}}(\hat{v})$ | Score entropy loss used to train $\tilde{v}$, Eq. (6) |
| $\boldsymbol{e}_i$ | One-hot vector with a 1 at position $i$ and 0 elsewhere |
| $\delta_{\boldsymbol{y}}(\cdot)$ | Indicator function with $\delta_{\boldsymbol{y}}(\boldsymbol{y}) = 1$ and $\delta_{\boldsymbol{y}}(\boldsymbol{y}') = 0$ $(\boldsymbol{y}' \neq \boldsymbol{y})$ |

## B  THE MARKOV PROCESSES OF DISCRETE DIFFUSION MODELS

### B.1  THE FORMULATIONS OF THE FORWARD PROCESS

**Semigroup Formulation.**  In general, the time-homogeneous CTMC can be described by a Markov semigroup $\mathcal{Q}_t^{\rightarrow}$ defined as:

$$\mathcal{Q}_t^{\rightarrow}[f](\boldsymbol{y}) = \mathbb{E}\left[f(\mathbf{y}_t)|\mathbf{y}_0 = \boldsymbol{y}\right] = \left\langle f, q_{t|0}^{\rightarrow}(\cdot|\boldsymbol{y})\right\rangle_{\mathcal{Y}} \tag{22}$$

where the function $f: \mathcal{Y} \to \mathbb{R}$. Due to the definition, the infinitesimal operator $\mathcal{L}^{\rightarrow}$ of the time homogeneous $\mathcal{Q}_t^{\rightarrow}$ is denoted as

$$\mathcal{L}^{\rightarrow}[f](\boldsymbol{y}) = \lim_{t \to 0}\left[\frac{\mathcal{Q}_t^{\rightarrow}[f] - f}{t}\right](\boldsymbol{y}) = \left\langle f, \partial_t q_{t|0}^{\rightarrow}(\cdot|\boldsymbol{y})\Big|_{t=0}\right\rangle_{\mathcal{Y}} := \langle f, R^{\rightarrow}(\cdot, \boldsymbol{y})\rangle_{\mathcal{Y}} \tag{23}$$

where

$$R^{\rightarrow}(\boldsymbol{y}', \boldsymbol{y}) := \partial_t q_{t|0}^{\rightarrow}(\boldsymbol{y}'|\boldsymbol{y})\Big|_{t=0} = \lim_{t \to 0}\left[\frac{q_{t|0}^{\rightarrow}(\boldsymbol{y}'|\boldsymbol{y}) - \delta_{\boldsymbol{y}}(\boldsymbol{y}')}{t}\right]. \tag{24}$$

According to the time-homogeneous property, we have

$$q_{t+\Delta t|t}^{\rightarrow}(\boldsymbol{y}'|\boldsymbol{y}) = \delta_{\boldsymbol{y}}(\boldsymbol{y}') + \Delta t \cdot R^{\rightarrow}(\boldsymbol{y}', \boldsymbol{y}) + o(\Delta t)$$

for any $t$. Here, the transition rate function $R^{\rightarrow}$ must satisfy

$$R^{\rightarrow}(\boldsymbol{y}, \boldsymbol{y}') \geq 0 \text{ when } \boldsymbol{y}' \neq \boldsymbol{y} \quad \text{and} \quad R^{\rightarrow}(\boldsymbol{y}', \boldsymbol{y}') = -\sum_{\boldsymbol{y} \neq \boldsymbol{y}'} R^{\rightarrow}(\boldsymbol{y}, \boldsymbol{y}') \leq 0 \qquad (25)$$

due to the definition Eq. (24). Under this setting, we can provide the dynamic of $q_{t|0}$ for any $t$. Specifically, we have

$$\partial_t \mathcal{Q}_t^{\rightarrow}[f](\boldsymbol{y}) = \mathcal{Q}_t^{\rightarrow}[\mathcal{L}f](\boldsymbol{y}) = \left\langle \mathcal{L}^{\rightarrow}f, q_{t|0}^{\rightarrow}(\cdot|\boldsymbol{y}) \right\rangle_{\mathcal{Y}} = \sum_{\boldsymbol{y}' \in \mathcal{Y}} \mathcal{L}^{\rightarrow}[f](\boldsymbol{y}') \cdot q_{t|0}^{\rightarrow}(\boldsymbol{y}'|\boldsymbol{y})$$

$$= \sum_{\boldsymbol{y}' \in \mathcal{Y}} \left[ \sum_{\tilde{\boldsymbol{y}} \in \mathcal{Y}} f(\tilde{\boldsymbol{y}}) \cdot R^{\rightarrow}(\tilde{\boldsymbol{y}}, \boldsymbol{y}') \cdot q_{t|0}(\boldsymbol{y}'|\boldsymbol{y}) \right] = \sum_{\tilde{\boldsymbol{y}} \in \mathcal{Y}} \left[ f(\tilde{\boldsymbol{y}}) \cdot \sum_{\boldsymbol{y}' \in \mathcal{Y}} R^{\rightarrow}(\tilde{\boldsymbol{y}}, \boldsymbol{y}') \cdot q_{t|0}(\boldsymbol{y}'|\boldsymbol{y}) \right],$$

where the first inequality follows from the semigroup property. Combined with the fact

$$\partial_t \mathcal{Q}_t^{\rightarrow}[f](\boldsymbol{y}) = \left\langle f, \partial_t q_{t|0}^{\rightarrow}(\cdot|\boldsymbol{y}) \right\rangle_{\mathcal{Y}}$$

derived from Eq. (22), we have

$$\partial_t q_{t|0}^{\rightarrow}(\tilde{\boldsymbol{y}}|\boldsymbol{y}) = \sum_{\boldsymbol{y}' \in \mathcal{Y}} R(\tilde{\boldsymbol{y}}, \boldsymbol{y}') \cdot q_{t|0}^{\rightarrow}(\boldsymbol{y}'|\boldsymbol{y}) = \left\langle R(\tilde{\boldsymbol{y}}, \cdot), q_{t|0}^{\rightarrow}(\cdot|\boldsymbol{y}) \right\rangle_{\mathcal{Y}}.$$

According to the time-homogeneous property, the above equation can be easily extended to

$$\partial_t q_{t|s}^{\rightarrow}(\tilde{\boldsymbol{y}}|\boldsymbol{y}) = \sum_{\boldsymbol{y}' \in \mathcal{Y}} R(\tilde{\boldsymbol{y}}, \boldsymbol{y}') \cdot q_{t|s}^{\rightarrow}(\boldsymbol{y}'|\boldsymbol{y}) = \left\langle R(\tilde{\boldsymbol{y}}, \cdot), q_{t|s}^{\rightarrow}(\cdot|\boldsymbol{y}) \right\rangle_{\mathcal{Y}}. \qquad (26)$$

Combining with Bayes' Theorem, the transition of the marginal distribution is

$$\frac{\mathrm{d}q_t^{\rightarrow}}{\mathrm{d}t}(\boldsymbol{y}) = \langle R(\boldsymbol{y}, \cdot), q_t^{\rightarrow} \rangle_{\mathcal{Y}}. \qquad (27)$$

**Matrix Formulation.** Suppose the support set $\mathcal{Y}$ of $q_t^{\rightarrow}$ be written as $\mathcal{Y} = \{\boldsymbol{y}_1, \boldsymbol{y}_2, \ldots, \boldsymbol{y}_{|\mathcal{Y}|}\}$, we may consider the marginal distribution $q_s^{\rightarrow}$ to be a vector, i.e.,

$$\boldsymbol{q}_t^{\rightarrow} = \left[ q_t(\boldsymbol{y}_1), q_t(\boldsymbol{y}_2), \ldots, q_t(\boldsymbol{y}_{|\mathcal{Y}|}) \right],$$

conditional transition probability function $q_{t|s}^{\rightarrow}$ to be a matrix, i.e.,

$$\boldsymbol{Q}_{t|s}^{\rightarrow} = \begin{bmatrix} q_{t|s}^{\rightarrow}(\boldsymbol{y}_1|\boldsymbol{y}_1) & q_{t|s}^{\rightarrow}(\boldsymbol{y}_1|\boldsymbol{y}_2) & \cdots & q_{t|s}^{\rightarrow}(\boldsymbol{y}_1|\boldsymbol{y}_{|\mathcal{Y}|}) \\ q_{t|s}^{\rightarrow}(\boldsymbol{y}_2|\boldsymbol{y}_1) & q_{t|s}^{\rightarrow}(\boldsymbol{y}_2|\boldsymbol{y}_2) & \cdots & q_{t|s}^{\rightarrow}(\boldsymbol{y}_2|\boldsymbol{y}_{|\mathcal{Y}|}) \\ \cdots & \cdots & \cdots & \cdots \\ q_{t|s}^{\rightarrow}(\boldsymbol{y}_{|\mathcal{Y}|}|\boldsymbol{y}_1) & q_{t|s}^{\rightarrow}(\boldsymbol{y}_{|\mathcal{Y}|}|\boldsymbol{y}_2) & \cdots & q_{t|s}^{\rightarrow}(\boldsymbol{y}_{|\mathcal{Y}|}|\boldsymbol{y}_{|\mathcal{Y}|}) \end{bmatrix}.$$

Similarly, the function $R$ can also be presented as

$$\boldsymbol{R}^{\rightarrow} = \begin{bmatrix} R^{\rightarrow}(\boldsymbol{y}_1, \boldsymbol{y}_1) & R^{\rightarrow}(\boldsymbol{y}_1, \boldsymbol{y}_2) & \cdots & R^{\rightarrow}(\boldsymbol{y}_1, \boldsymbol{y}_{|\mathcal{Y}|}) \\ R^{\rightarrow}(\boldsymbol{y}_2, \boldsymbol{y}_1) & R^{\rightarrow}(\boldsymbol{y}_2, \boldsymbol{y}_2) & \cdots & R^{\rightarrow}(\boldsymbol{y}_2, \boldsymbol{y}_{|\mathcal{Y}|}) \\ \cdots & \cdots & \cdots & \cdots \\ R^{\rightarrow}(\boldsymbol{y}_{|\mathcal{Y}|}, \boldsymbol{y}_1) & R^{\rightarrow}(\boldsymbol{y}_{|\mathcal{Y}|}, \boldsymbol{y}_2) & \cdots & R^{\rightarrow}(\boldsymbol{y}_{|\mathcal{Y}|}, \boldsymbol{y}_{|\mathcal{Y}|}) \end{bmatrix}. \qquad (28)$$

Under this condition, Eq. (27) can be written as

$$\mathrm{d}\boldsymbol{q}_t^{\rightarrow}/\mathrm{d}t = \boldsymbol{R}^{\rightarrow} \cdot \boldsymbol{q}_t^{\rightarrow} \qquad (29)$$

matching the usual presentation shown in Chen & Ying (2024); Zhang et al. (2024).

## B.2 THE PROOF OF LEMMA 1

*The proof of Lemma 1.* For any $t \in [0, T]$, the marginal, joint, and conditional distribution w.r.t. $\{\mathbf{y}_t^{\leftarrow}\}$ are denoted as

$$\mathbf{y}_t^{\leftarrow} \sim q_t^{\leftarrow}, \quad (\mathbf{y}_t^{\leftarrow}, \mathbf{y}_{t'}^{\leftarrow}) \sim q_{t,t'}^{\leftarrow}, \quad \text{and} \quad q_{t'|t}^{\leftarrow} = q_{t',t}/q_t,$$

which have $q_t^{\leftarrow} = q_{T-t}^{\rightarrow}$. Then, we start to check the dynamic of $q_{t|s}^{\leftarrow}$, i.e.,

$$\partial_t q_{t|s}^{\leftarrow}(\mathbf{y}'|\mathbf{y}) = -1 \cdot \partial_{T-t} q_{T-t|T-s}^{\rightarrow}(\mathbf{y}'|\mathbf{y}) = -1 \cdot \partial_{T-t} \left[ \frac{q_{T-s|T-t}^{\rightarrow}(\mathbf{y}|\mathbf{y}') \cdot q_{T-t}^{\rightarrow}(\mathbf{y}')}{q_{T-s}^{\rightarrow}(\mathbf{y})} \right]$$

$$= \underbrace{-\partial_{T-t} q_{T-s|T-t}^{\rightarrow}(\mathbf{y}|\mathbf{y}') \cdot \frac{q_{T-t}^{\rightarrow}(\mathbf{y}')}{q_{T-s}^{\rightarrow}(\mathbf{y})}}_{\text{Term 1}} - \underbrace{\frac{q_{T-s|T-t}^{\rightarrow}(\mathbf{y}|\mathbf{y}')}{q_{T-s}^{\rightarrow}(\mathbf{y})} \cdot \partial_{T-t} q_{T-t}^{\rightarrow}(\mathbf{y}')}_{\text{Term 2}}. \quad (30)$$

For Term 1 of Eq. (30), we have

$$\text{Term 1} = -\sum_{\tilde{\mathbf{y}} \in \mathcal{Y}} R^{\rightarrow}(\tilde{\mathbf{y}}, \mathbf{y}') \cdot q_{T-s|T-t}^{\rightarrow}(\mathbf{y}|\tilde{\mathbf{y}}) \cdot \frac{q_{T-t}^{\rightarrow}(\tilde{\mathbf{y}})}{q_{T-s}^{\rightarrow}(\mathbf{y})} \cdot \frac{q_{T-t}^{\rightarrow}(\mathbf{y}')}{q_{T-t}^{\rightarrow}(\tilde{\mathbf{y}})}$$

$$= -\sum_{\tilde{\mathbf{y}} \in \mathcal{Y}} R^{\rightarrow}(\tilde{\mathbf{y}}, \mathbf{y}') \cdot \frac{q_{T-t}^{\rightarrow}(\mathbf{y}')}{q_{T-t}^{\rightarrow}(\tilde{\mathbf{y}})} \cdot q_{T-t|T-s}^{\rightarrow}(\tilde{\mathbf{y}}|\mathbf{y}),$$

where the first equation follows from the Kolmogorov backward theorem (Lemma 14) and Eq. (23):

$$\partial_{T-t} q_{T-s|T-t}^{\rightarrow}(\mathbf{y}|\mathbf{y}') = -\mathcal{L}^{\rightarrow}[q_{T-s|T-t}^{\rightarrow}(\mathbf{y}|\cdot)](\mathbf{y}') = -\left\langle q_{T-s|T-t}^{\rightarrow}(\mathbf{y}|\cdot), R^{\rightarrow}(\cdot, \mathbf{y}') \right\rangle_{\mathcal{Y}}.$$

For Term 2 of Eq. (30), we have

$$\text{Term 2} = \frac{q_{T-s|T-t}^{\rightarrow}(\mathbf{y}|\mathbf{y}')}{q_{T-s}^{\rightarrow}(\mathbf{y})} \cdot \sum_{\tilde{\mathbf{y}} \in \mathcal{Y}} R^{\rightarrow}(\mathbf{y}', \tilde{\mathbf{y}}) \cdot q_{T-t}^{\rightarrow}(\tilde{\mathbf{y}})$$

$$= \frac{q_{T-s|T-t}^{\rightarrow}(\mathbf{y}|\mathbf{y}') \cdot q_{T-t}^{\rightarrow}(\mathbf{y}')}{q_{T-s}^{\rightarrow}(\mathbf{y})} \cdot \sum_{\tilde{\mathbf{y}} \in \mathcal{Y}} R^{\rightarrow}(\mathbf{y}', \tilde{\mathbf{y}}) \cdot \frac{q_{T-t}^{\rightarrow}(\tilde{\mathbf{y}})}{q_{T-t}^{\rightarrow}(\mathbf{y}')} = 0,$$

where the first equation follows from Eq. (27) and the last equation follows from the fact

$$\sum_{\tilde{\mathbf{y}} \in \mathcal{Y}} R^{\rightarrow}(\mathbf{y}', \tilde{\mathbf{y}}) \cdot \frac{q_{T-t}^{\rightarrow}(\tilde{\mathbf{y}})}{q_{T-t}^{\rightarrow}(\mathbf{y}')} = \sum_{\tilde{\mathbf{y}} \in \mathcal{Y}} \lim_{t \to 0} \left[ \frac{q_{t|0}^{\rightarrow}(\mathbf{y}'|\tilde{\mathbf{y}}) - \delta_{\tilde{\mathbf{y}}}(\mathbf{y}')}{t} \right] \cdot \frac{q_{T-t}^{\rightarrow}(\tilde{\mathbf{y}})}{q_{T-t}^{\rightarrow}(\mathbf{y}')}$$

$$= \sum_{\tilde{\mathbf{y}} \in \mathcal{Y}} \lim_{t' \to T-t} \left[ \frac{q_{t'|T-t}^{\rightarrow}(\mathbf{y}'|\tilde{\mathbf{y}}) - \delta_{\tilde{\mathbf{y}}}(\mathbf{y}')}{t' - (T-t)} \right] \cdot \lim_{t' \to T-t} \frac{q_{T-t}^{\rightarrow}(\tilde{\mathbf{y}})}{q_{t'}^{\rightarrow}(\mathbf{y}')} = \sum_{\tilde{\mathbf{y}} \in \mathcal{Y}} \lim_{t' \to T-t} \left[ \frac{q_{T-t|t'}^{\rightarrow}(\tilde{\mathbf{y}}|\mathbf{y}') - \delta_{\mathbf{y}'}(\tilde{\mathbf{y}})}{t' - (T-t)} \right] = 0.$$

Under this condition, by setting

$$R_t^{\leftarrow}(\mathbf{y}', \tilde{\mathbf{y}}) := R(\tilde{\mathbf{y}}, \mathbf{y}') \cdot \frac{q_t^{\leftarrow}(\mathbf{y}')}{q_t^{\leftarrow}(\tilde{\mathbf{y}})},$$

then Eq. (30) can be summarized as

$$\partial_t q_{t|s}^{\leftarrow}(\mathbf{y}'|\mathbf{y}) = \left\langle R_t^{\leftarrow}(\mathbf{y}', \cdot), q_{t|s}^{\leftarrow}(\cdot|\mathbf{y}) \right\rangle_{\mathcal{Y}} = \sum_{\tilde{\mathbf{y}} \in \mathcal{Y}} R_t^{\leftarrow}(\mathbf{y}', \tilde{\mathbf{y}}) \cdot q_{t|s}^{\leftarrow}(\tilde{\mathbf{y}}|\mathbf{y}). \quad (31)$$

Combining with Bayes' Theorem, we have

$$\frac{\mathrm{d}q_t^{\leftarrow}}{\mathrm{d}t}(\mathbf{y}) = \langle R_t^{\leftarrow}(\mathbf{y}, \cdot), q_t^{\leftarrow} \rangle_{\mathcal{Y}}. \quad (32)$$

Hence, Eq. (3) establishes.

Moreover, since the RHS of Eq. (4) satisfies

$$\lim_{\Delta t \to 0} \left[ \frac{q_{t+\Delta t|t}^{\leftarrow}(\boldsymbol{y}|\boldsymbol{y}') - \delta_{\boldsymbol{y}'}(\boldsymbol{y})}{\Delta t} \right] = \lim_{s \to t} \partial_t q_{t|s}^{\leftarrow}(\boldsymbol{y}|\boldsymbol{y}').$$

Besides, we have

$$\lim_{s \to t} \partial_t q_{t|s}^{\leftarrow}(\boldsymbol{y}|\boldsymbol{y}') = \lim_{s \to t} \partial_t \left[ q_{T-s|T-t}^{\rightarrow}(\boldsymbol{y}'|\boldsymbol{y}) \cdot \frac{q_{T-t}^{\rightarrow}(\boldsymbol{y})}{q_{T-s}^{\rightarrow}(\boldsymbol{y}')} \right]$$

$$= \lim_{s \to t} \left[ \partial_t(q_{T-s|T-t}^{\rightarrow}(\boldsymbol{y}'|\boldsymbol{y})) \cdot \frac{q_{T-t}^{\rightarrow}(\boldsymbol{y})}{q_{T-s}^{\rightarrow}(\boldsymbol{y}')} + q_{T-s|T-t}^{\rightarrow}(\boldsymbol{y}'|\boldsymbol{y}) \cdot \frac{\partial_t q_{T-t}^{\rightarrow}(\boldsymbol{y})}{q_{T-s}^{\rightarrow}(\boldsymbol{y}')} \right].$$

When $\boldsymbol{y} \neq \boldsymbol{y}'$, we have

$$\lim_{s \to t} q_{T-s|T-t}^{\rightarrow}(\boldsymbol{y}'|\boldsymbol{y}) = 0,$$

which implies

$$\lim_{s \to t} \partial_t q_{t|s}^{\leftarrow}(\boldsymbol{y}|\boldsymbol{y}') = \lim_{s \to t} \partial_t(q_{T-s|T-t}^{\rightarrow}(\boldsymbol{y}'|\boldsymbol{y})) \cdot \frac{q_{T-t}^{\rightarrow}(\boldsymbol{y})}{q_{T-s}^{\rightarrow}(\boldsymbol{y}')} = R^{\rightarrow}(\boldsymbol{y}', \boldsymbol{y}) \cdot \frac{q_{T-t}^{\rightarrow}(\boldsymbol{y})}{q_{T-t}^{\rightarrow}(\boldsymbol{y}')}.$$

The last equation follows from the Kolmogorov backward theorem, i.e., Lemma 14 and Eq. (23)

$$\partial_{T-t} q_{T-s|T-t}^{\rightarrow}(\boldsymbol{y}'|\boldsymbol{y}) = -\mathcal{L}^{\rightarrow}[q_{T-s|T-t}^{\rightarrow}(\boldsymbol{y}'|\cdot)](\boldsymbol{y}) = -\left\langle q_{T-s|T-t}^{\rightarrow}(\boldsymbol{y}'|\cdot), R^{\rightarrow}(\cdot, \boldsymbol{y}) \right\rangle_{\mathcal{Y}} = R^{\rightarrow}(\boldsymbol{y}', \boldsymbol{y}).$$

Combining with Eq. (3), we have

$$\lim_{\Delta t \to 0} \left[ \frac{q_{t+\Delta t|t}^{\leftarrow}(\boldsymbol{y}|\boldsymbol{y}') - \delta_{\boldsymbol{y}'}(\boldsymbol{y})}{\Delta t} \right] = \lim_{s \to t} \partial_t q_{t|s}^{\leftarrow}(\boldsymbol{y}|\boldsymbol{y}') = R^{\rightarrow}(\boldsymbol{y}', \boldsymbol{y}) \cdot \frac{q_{T-t}^{\rightarrow}(\boldsymbol{y})}{q_{T-t}^{\rightarrow}(\boldsymbol{y}')} = R_t^{\leftarrow}(\boldsymbol{y}, \boldsymbol{y}') \quad (33)$$

when $\boldsymbol{y}' \neq \boldsymbol{y}$. Besides, we have

$$\sum_{\boldsymbol{y} \in \mathcal{Y}} R_t^{\leftarrow}(\boldsymbol{y}, \boldsymbol{y}') = \sum_{\boldsymbol{y} \in \mathcal{Y}} R^{\rightarrow}(\boldsymbol{y}', \boldsymbol{y}) \cdot \frac{q_{T-t}^{\rightarrow}(\boldsymbol{y})}{q_{T-t}^{\rightarrow}(\boldsymbol{y}')}$$

$$= \sum_{\boldsymbol{y} \in \mathcal{Y}} \lim_{\Delta t \to 0} \left[ \frac{q_{T-t+\Delta t|T-t}^{\rightarrow}(\boldsymbol{y}'|\boldsymbol{y}) - \delta_{\boldsymbol{y}}(\boldsymbol{y}')}{\Delta t} \right] \cdot \frac{q_{T-t}^{\rightarrow}(\boldsymbol{y})}{q_{T-t}^{\rightarrow}(\boldsymbol{y}')} = \sum_{\boldsymbol{y} \in \mathcal{Y}} \lim_{\Delta t \to 0} \left[ \frac{q_{T-t+\Delta t|T-t}^{\rightarrow}(\boldsymbol{y}|\boldsymbol{y}') - \delta_{\boldsymbol{y}'}(\boldsymbol{y})}{\Delta t} \right] = 0,$$

which means

$$R_t^{\leftarrow}(\boldsymbol{y}', \boldsymbol{y}') = -\sum_{\boldsymbol{y} \neq \boldsymbol{y}'} R_t^{\leftarrow}(\boldsymbol{y}, \boldsymbol{y}') = \lim_{\Delta t \to 0} -\left[ \frac{1 - \sum_{\boldsymbol{y} \neq \boldsymbol{y}'} q_{t+\Delta t|t}^{\leftarrow}(\boldsymbol{y}|\boldsymbol{y}')}{\Delta t} \right],$$

where the last inequality follows from Eq. (33). Hence, Eq. (3) establishes, and the proof is completed. $\square$

### B.3 THE PROOF OF LEMMA 2

**Lemma 4.** *The close solution of Eq.* (29) *is*

$$\boldsymbol{q}_t^{\rightarrow} = \exp(t\boldsymbol{R}^{\rightarrow}) \cdot \boldsymbol{q}_0^{\rightarrow} \quad \text{where} \quad \exp(t\boldsymbol{R}^{\rightarrow}) = \sum_{i=0}^{\infty} \frac{1}{i!}(t\boldsymbol{R}^{\rightarrow})^i = \boldsymbol{I} + t\boldsymbol{R}^{\rightarrow} + \frac{(t\boldsymbol{R}^{\rightarrow})^2}{2} + \dots.$$

*Proof.* We can easily verify that

$$\frac{\mathrm{d}\boldsymbol{q}_t^{\rightarrow}}{\mathrm{d}t} = \frac{\mathrm{d}}{\mathrm{d}t}\left[\exp(t\boldsymbol{R}^{\rightarrow})\boldsymbol{q}_0^{\rightarrow}\right] = \frac{\mathrm{d}}{\mathrm{d}t}\left[\exp(t\boldsymbol{R}^{\rightarrow})\right]\boldsymbol{q}_0^{\rightarrow}.$$

With the following equation,

$$\frac{\mathrm{d}}{\mathrm{d}t}\left[\exp(t\boldsymbol{R}^{\rightarrow})\right] = \frac{\mathrm{d}}{\mathrm{d}t}\left[\sum_{i=0}^{\infty}\frac{(t\boldsymbol{R}^{\rightarrow})^i}{i!}\right] = \sum_{i=1}^{\infty}\frac{t^{i-1}}{(i-1)!}\cdot(\boldsymbol{R}^{\rightarrow})^i = \boldsymbol{R}^{\rightarrow}\cdot\sum_{j=0}^{\infty}\frac{(t\boldsymbol{R}^{\rightarrow})^j}{j!} = \boldsymbol{R}^{\rightarrow}\cdot\exp(t\boldsymbol{R}^{\rightarrow}),$$

we have

$$\frac{\mathrm{d}\boldsymbol{q}_t^{\rightarrow}}{\mathrm{d}t} = \boldsymbol{R}^{\rightarrow}\cdot\exp(t\boldsymbol{R}^{\rightarrow})\cdot\boldsymbol{q}_0^{\rightarrow} = \boldsymbol{R}^{\rightarrow}\cdot\boldsymbol{q}_t^{\rightarrow}.$$

Hence, the proof is completed. $\square$

**Lemma 5.** *Suppose the transition rate matrix $\boldsymbol{R}^{\rightarrow}$ shown as Eq. (28) satisfies Eq. (7). It can be decomposed as*

$$\boldsymbol{R}^{\rightarrow} = \sum_{i=1}^{d} \boldsymbol{R}_i^{\rightarrow} \quad where \quad \boldsymbol{R}_i^{\rightarrow} = \underbrace{\boldsymbol{I} \otimes \cdots}_{i-1 \; terms} \otimes \boldsymbol{A} \otimes \cdots \otimes \boldsymbol{I},$$

*where $\otimes$ denotes the Kronecker product, $\boldsymbol{I}$ denotes the identity matrix on $\mathbb{R}^{K \times K}$, and $\boldsymbol{A}$ satisfies*

$$\boldsymbol{A} = \begin{bmatrix} -1 & 0 & \dots & 0 \\ 0 & -1 & \dots & 0 \\ \vdots & \vdots & \ddots & \vdots \\ 1 & 1 & \dots & 0 \end{bmatrix}. \tag{34}$$

*Proof.* According to the calculation of the Kronecker product, we have

$$\boldsymbol{R}_i^{\rightarrow}(\boldsymbol{y}, \boldsymbol{y}') = \boldsymbol{I}(\boldsymbol{y}_1, \boldsymbol{y}_1') \cdot \ldots \cdot \boldsymbol{A}(\boldsymbol{y}_i, \boldsymbol{y}_i') \cdot \ldots \cdot \boldsymbol{I}(\boldsymbol{y}_d, \boldsymbol{y}_d').$$

Under this condition, suppose $\text{Ham}(\boldsymbol{y}, \boldsymbol{y}') \geq 2$ and $\text{DiffIdx}(\boldsymbol{y}, \boldsymbol{y}') = \{j_1, j_2, \ldots\}$ without loss of generality, for any $j \notin \{j_1, j_2\}$, we have

$$\boldsymbol{R}_j^{\rightarrow}(\boldsymbol{y}, \boldsymbol{y}') = \boldsymbol{A}(\boldsymbol{y}_j, \boldsymbol{y}_j') \cdot \boldsymbol{I}(\boldsymbol{y}_1, \boldsymbol{y}_1') \cdot \ldots \cdot \underbrace{\boldsymbol{I}(\boldsymbol{y}_{j_1}, \boldsymbol{y}_{j_1}')}_{=0} \cdot \ldots \cdot \underbrace{\boldsymbol{I}(\boldsymbol{y}_{j_2}, \boldsymbol{y}_{j_2}')}_{=0} \cdot \ldots \cdot \boldsymbol{I}(\boldsymbol{y}_d, \boldsymbol{y}_d') = 0.$$

Besides, for $j = j_1$, we have

$$\boldsymbol{R}_{j_1}^{\rightarrow}(\boldsymbol{y}, \boldsymbol{y}') = \boldsymbol{A}(\boldsymbol{y}_{j_1}, \boldsymbol{y}_{j_1}') \cdot \boldsymbol{I}(\boldsymbol{y}_1, \boldsymbol{y}_1') \cdot \ldots \cdot \underbrace{\boldsymbol{I}(\boldsymbol{y}_{j_2}, \boldsymbol{y}_{j_2}')}_{=0} \cdot \ldots \cdot \boldsymbol{I}(\boldsymbol{y}_d, \boldsymbol{y}_d') = 0.$$

A similar result will be satisfied for $j = j_2$. Hence, it has

$$\boldsymbol{R}^{\rightarrow}(\boldsymbol{y}, \boldsymbol{y}') = \sum_{i=1}^{d} \boldsymbol{R}_i^{\rightarrow}(\boldsymbol{y}, \boldsymbol{y}') = 0 \quad \text{when} \quad \text{Ham}(\boldsymbol{y}, \boldsymbol{y}') \geq 2$$

Then, suppose $\text{Ham}(\boldsymbol{y}, \boldsymbol{y}') = 1$ and $\text{DiffIdx}(\boldsymbol{y}, \boldsymbol{y}') = j_1$, for any $j \neq j_1$, we have

$$\boldsymbol{R}_j^{\rightarrow}(\boldsymbol{y}, \boldsymbol{y}') = \boldsymbol{A}(\boldsymbol{y}_j, \boldsymbol{y}_j') \cdot \boldsymbol{I}(\boldsymbol{y}_0, \boldsymbol{y}_0') \cdot \ldots \cdot \underbrace{\boldsymbol{I}(\boldsymbol{y}_{j_1}, \boldsymbol{y}_{j_1}')}_{=0} \cdot \ldots \cdot \boldsymbol{I}(\boldsymbol{y}_d, \boldsymbol{y}_d') = 0.$$

Otherwise, when $j = j_1$, we have

$$\boldsymbol{R}_{j_1}^{\rightarrow}(\boldsymbol{y}, \boldsymbol{y}') = \boldsymbol{A}(\boldsymbol{y}_{j_1}, \boldsymbol{y}_{j_1}') \cdot \boldsymbol{I}(\boldsymbol{y}_1, \boldsymbol{y}_1') \cdot \ldots \cdot \boldsymbol{I}(\boldsymbol{y}_d, \boldsymbol{y}_d') = \boldsymbol{A}(\boldsymbol{y}_{j_1}, \boldsymbol{y}_{j_1}')$$

where the second equation establishes since $\text{Ham}(\boldsymbol{y}, \boldsymbol{y}') = 1$ and $\boldsymbol{y}_j = \boldsymbol{y}_j'$ when $j \neq j_1$. Then, only when $\boldsymbol{y}_{j_1} = K$, we will have $\boldsymbol{A}(\boldsymbol{y}_{j_1}, \boldsymbol{y}_{j_1}') = 1$ otherwise $\boldsymbol{A}(\boldsymbol{y}_{j_1}, \boldsymbol{y}_{j_1}') = 0$ due to the definition Eq. (34). That means

$$\boldsymbol{R}^{\rightarrow}(\boldsymbol{y}, \boldsymbol{y}') = \sum_{i=1}^{d} \boldsymbol{R}_i^{\rightarrow}(\boldsymbol{y}, \boldsymbol{y}') = 0 \quad \text{when} \quad \text{Ham}(\boldsymbol{y}, \boldsymbol{y}') = 1 \text{ and } \boldsymbol{y}_{\text{DiffIdx}(\boldsymbol{y}, \boldsymbol{y}')} \neq K$$

$$\boldsymbol{R}^{\rightarrow}(\boldsymbol{y}, \boldsymbol{y}') = \sum_{i=1}^{d} \boldsymbol{R}_i^{\rightarrow}(\boldsymbol{y}, \boldsymbol{y}') = 1 \quad \text{when} \quad \text{Ham}(\boldsymbol{y}, \boldsymbol{y}') = 1 \text{ and } \boldsymbol{y}_{\text{DiffIdx}(\boldsymbol{y}, \boldsymbol{y}')} = K.$$

Then, suppose $\text{Ham}(\boldsymbol{y}, \boldsymbol{y}') = 0$, i.e., $\boldsymbol{y} = \boldsymbol{y}'$, for any $j \in \{1, 2, \ldots, d\}$, we have

$$\boldsymbol{R}_j^{\rightarrow}(\boldsymbol{y}, \boldsymbol{y}') = \boldsymbol{A}(\boldsymbol{y}_j, \boldsymbol{y}_j') \cdot \boldsymbol{I}(\boldsymbol{y}_1, \boldsymbol{y}_1') \cdot \ldots \cdot \boldsymbol{I}(\boldsymbol{y}_d, \boldsymbol{y}_d') = \boldsymbol{A}(\boldsymbol{y}_j, \boldsymbol{y}_j'),$$

and

$$\sum_{i=1}^{d} \boldsymbol{R}_i^{\rightarrow}(\boldsymbol{y}, \boldsymbol{y}') = \sum_{j=1}^{d} \boldsymbol{A}(\boldsymbol{y}_j, \boldsymbol{y}_j) = -\sum_{i=1}^{d} (1 - \delta_K(\boldsymbol{y}_i)),$$

which implies we have $\boldsymbol{R}^{\rightarrow}(\boldsymbol{y}, \boldsymbol{y}') = \sum_{i=0}^{d-1} \boldsymbol{R}_i^{\rightarrow}(\boldsymbol{y}, \boldsymbol{y}')$ when $\boldsymbol{y} = \boldsymbol{y}'$. Hence, the proof is completed. $\qquad \square$

**Lemma 6.** *With the decomposition shown in Lemma 5, i.e.,*

$$\boldsymbol{R}^{\rightarrow} = \sum_{i=1}^{d} \boldsymbol{R}_i^{\rightarrow} \quad \text{where} \quad \boldsymbol{R}_i^{\rightarrow} = \underbrace{\boldsymbol{I} \otimes \ldots \otimes \boldsymbol{I}}_{i-1 \text{ terms}} \otimes \boldsymbol{A} \otimes \underbrace{\boldsymbol{I} \otimes \ldots \otimes \boldsymbol{I}}_{d-i \text{ terms}},$$

*for any $i, j \in \{1, 2, \ldots, d\}$, the matrices $\boldsymbol{R}_i^{\rightarrow}$ and $\boldsymbol{R}_j^{\rightarrow}$ satisfy*

$$\boldsymbol{R}_i^{\rightarrow} \cdot \boldsymbol{R}_j^{\rightarrow} = \boldsymbol{R}_j^{\rightarrow} \cdot \boldsymbol{R}_i^{\rightarrow},$$

*which implies*

$$\exp(t\boldsymbol{R}^{\rightarrow}) = \exp\left(t \sum_{i=1}^{d} \boldsymbol{R}_i^{\rightarrow}\right) = \prod_{i=1}^{d} \exp\left(t\boldsymbol{R}_i^{\rightarrow}\right) = \exp(t\boldsymbol{A})^{\otimes d}$$

*Proof.* According to Lemma 5, the matrix $\boldsymbol{R}^{\rightarrow}$ has the following decomposition, i.e.,

$$\boldsymbol{R}^{\rightarrow} = \sum_{i=1}^{d} \boldsymbol{R}_i^{\rightarrow} \quad \text{where} \quad \boldsymbol{R}_i^{\rightarrow} = \underbrace{\boldsymbol{I} \otimes \ldots \otimes \boldsymbol{I}}_{i-1 \text{ terms}} \otimes \boldsymbol{A} \otimes \underbrace{\boldsymbol{I} \otimes \ldots \otimes \boldsymbol{I}}_{d-i \text{ terms}},$$

where $\otimes$ denotes the Kronecker product, $\boldsymbol{I}$ denotes the identity matrix on $\mathbb{R}^{K \times K}$, and $\boldsymbol{A}$ satisfies

$$\boldsymbol{A} = \begin{bmatrix} -1 & 0 & \ldots & 0 \\ 0 & -1 & \ldots & 0 \\ \vdots & \vdots & \ddots & \vdots \\ 1 & 1 & \ldots & 0 \end{bmatrix}.$$

We can easily verify that the matrix $\boldsymbol{A}$ can be decomposed as

$$\begin{bmatrix} -\boldsymbol{I}_{K-1} & \boldsymbol{0} \\ \boldsymbol{1}_{1 \times (K-1)} & 0 \end{bmatrix} = \underbrace{\begin{bmatrix} \boldsymbol{I}_{K-1} & \boldsymbol{0} \\ -\boldsymbol{1}_{1 \times (K-1)} & 1 \end{bmatrix}}_{\boldsymbol{U}} \cdot \underbrace{\begin{bmatrix} -\boldsymbol{I}_{K-1} & 0 \\ \boldsymbol{0} & 0 \end{bmatrix}}_{\Lambda} \cdot \underbrace{\begin{bmatrix} \boldsymbol{I}_{K-1} & \boldsymbol{0} \\ \boldsymbol{1}_{1 \times (K-1)} & 1 \end{bmatrix}}_{\boldsymbol{U}^{-1}} \quad \text{where} \quad \boldsymbol{U}\boldsymbol{U}^{-1} = \boldsymbol{U}^{-1}\boldsymbol{U} = \boldsymbol{I}_K.$$

(35)

Under this condition, $\boldsymbol{R}_i^{\rightarrow}$ can be reformulated as

$$\boldsymbol{R}_i^{\rightarrow} = \underbrace{(\boldsymbol{U}\boldsymbol{U}^{-1}) \otimes \ldots \otimes (\boldsymbol{U}\boldsymbol{U}^{-1})}_{i-1 \text{ terms}} \otimes (\boldsymbol{U}\Lambda\boldsymbol{U}^{-1}) \otimes (\boldsymbol{U}\boldsymbol{U}^{-1}) \otimes \ldots (\boldsymbol{U}\boldsymbol{U}^{-1})$$

$$= (\boldsymbol{U} \otimes \ldots \otimes \boldsymbol{U}) \cdot \left(\underbrace{\boldsymbol{I} \otimes \ldots \otimes \boldsymbol{I}}_{i-1 \text{ terms}} \otimes \Lambda \otimes \boldsymbol{I} \ldots \otimes \boldsymbol{I}\right) \cdot \left(\boldsymbol{U}^{-1} \otimes \ldots \otimes \boldsymbol{U}^{-1}\right) := \boldsymbol{U}^{\otimes d} \cdot \Lambda_i \cdot (\boldsymbol{U}^{-1})^{\otimes d}$$

where the last inequality follows from Lemma 13. Under this condition, it has

$$\boldsymbol{R}_i^{\rightarrow} \cdot \boldsymbol{R}_j^{\rightarrow} = \boldsymbol{U}^{\otimes d} \cdot \Lambda_i \cdot (\boldsymbol{U}^{-1})^{\otimes d} \cdot \boldsymbol{U}^{\otimes d} \cdot \Lambda_j \cdot (\boldsymbol{U}^{-1})^{\otimes d} = \boldsymbol{U}^{\otimes d} \cdot \Lambda_i \cdot \Lambda_j \cdot (\boldsymbol{U}^{-1})^{\otimes d}$$

$$= \boldsymbol{U}^{\otimes d} \cdot \Lambda_j \cdot \Lambda_i \cdot (\boldsymbol{U}^{-1})^{\otimes d} = \boldsymbol{U}^{\otimes d} \cdot \Lambda_i \cdot (\boldsymbol{U}^{-1})^{\otimes d} \cdot \boldsymbol{U}^{\otimes d} \cdot \Lambda_j \cdot (\boldsymbol{U}^{-1})^{\otimes d} = \boldsymbol{R}_j^{\rightarrow} \cdot \boldsymbol{R}_i^{\rightarrow},$$

where the second and forth equations follows from Lemma 13 and Eq. (35).

For the property about the matrix exponential, we start from investigating the case of two commuting matrices, i.e., $\boldsymbol{R}_1^{\rightarrow}$ and $\boldsymbol{R}_2^{\rightarrow}$. By definition, we have

$$\exp(\boldsymbol{R}_1^{\rightarrow} + \boldsymbol{R}_2^{\rightarrow}) = \sum_{i=0}^{\infty} \frac{1}{i!} (\boldsymbol{R}_1^{\rightarrow} + \boldsymbol{R}_2^{\rightarrow})^i = \sum_{i=0}^{\infty} \frac{1}{i!} \sum_{j=0}^{i} C_i^j \cdot (\boldsymbol{R}_1^{\rightarrow})^j \cdot (\boldsymbol{R}_2^{\rightarrow})^{i-j}$$

where the last equation establishes since $\boldsymbol{R}_1^{\rightarrow}$ and $\boldsymbol{R}_2^{\rightarrow}$ are commute. Then, we have

$$\sum_{i=0}^{\infty} \frac{1}{i!} \sum_{j=0}^{i} C_i^j \cdot (\boldsymbol{R}_1^{\rightarrow})^j \cdot (\boldsymbol{R}_2^{\rightarrow})^{i-j} = \sum_{i=0}^{\infty} \sum_{j=0}^{i} \frac{1}{i!} \cdot \frac{i!}{j!(i-j)!} \cdot (\boldsymbol{R}_1^{\rightarrow})^j \cdot (\boldsymbol{R}_2^{\rightarrow})^{i-j}$$

$$= \sum_{i=0}^{\infty} \sum_{j=0}^{i} \frac{1}{j!(i-j)!} \cdot (\boldsymbol{R}_1^{\rightarrow})^j \cdot (\boldsymbol{R}_2^{\rightarrow})^{i-j} = \left(\sum_{j=0}^{\infty} \frac{(\boldsymbol{R}_1^{\rightarrow})^j}{j!}\right) \cdot \left(\sum_{i=0}^{\infty} \frac{(\boldsymbol{R}_2^{\rightarrow})^i}{i!}\right) = \exp(\boldsymbol{R}_1^{\rightarrow}) \cdot \exp(\boldsymbol{R}_2^{\rightarrow}).$$

According to the definition of the matrix exponential, we will have $\exp(\boldsymbol{A} \otimes \boldsymbol{B}) = \exp(\boldsymbol{A}) \otimes \exp(\boldsymbol{B})$ when one of the factors is the identity. When we multiply all these exponentials, it has

$$
\begin{aligned}
\exp(\boldsymbol{R}_1^{\rightarrow}) \cdot \exp(\boldsymbol{R}_2^{\rightarrow}) &= [\exp(\boldsymbol{A}) \otimes \boldsymbol{I} \otimes \ldots \otimes \boldsymbol{I}] \cdot [\boldsymbol{I} \otimes \exp(\boldsymbol{A}) \otimes \ldots \otimes \boldsymbol{I}] \\
&= [\exp(\boldsymbol{A}) \cdot \boldsymbol{I}] \otimes [\boldsymbol{I} \cdot \exp(\boldsymbol{A})] \otimes \boldsymbol{I} \ldots \otimes \boldsymbol{I}.
\end{aligned}
$$

Then, following a recursive manner, we have

$$
\exp\left(t \sum_{i=1}^{d} \boldsymbol{R}_i^{\rightarrow}\right) = \prod_{i=1}^{d} \exp\left(t \boldsymbol{R}_i^{\rightarrow}\right) = \exp(t\boldsymbol{A})^{\otimes d},
$$

hence the proof is completed. $\qquad\square$

**Lemma 7.** *Suppose matrix $\boldsymbol{A}$ is*

$$
\boldsymbol{A} = \begin{bmatrix} -1 & 0 & \ldots & 0 \\ 0 & -1 & \ldots & 0 \\ \vdots & \vdots & \ddots & \vdots \\ 1 & 1 & \ldots & 0 \end{bmatrix},
$$

*the matrix exponential $\exp(t\boldsymbol{A})$ becomes*

$$
\exp(t\boldsymbol{A}) = \begin{bmatrix} e^{-t} & 0 & \ldots & 0 & 0 \\ 0 & e^{-t} & \ldots & 0 & 0 \\ \vdots & \vdots & \ddots & \vdots & \vdots \\ 1 - e^{-t} & 1 - e^{-t} & \ldots & 1 - e^{-t} & 1 \end{bmatrix}.
$$

*Proof.* According to Lemma 4, $\bar{\boldsymbol{A}}(t) := \exp(t\boldsymbol{A})$ can be considered as the close solution of the following matrix ODE, i.e.,

$$
\frac{\mathrm{d}\bar{\boldsymbol{A}}(t)}{\mathrm{d}t} = \boldsymbol{A} \cdot \bar{\boldsymbol{A}}(t), \quad \text{where} \quad \bar{\boldsymbol{A}}(0) = \boldsymbol{I}. \tag{36}
$$

To provide a close form of $\bar{\boldsymbol{A}}_t$, we first decompose the matrix $\boldsymbol{A}$ as follows

$$
\boldsymbol{A} = \begin{bmatrix} \boldsymbol{B} & \boldsymbol{0} \\ \boldsymbol{C} & \boldsymbol{0} \end{bmatrix} \quad \text{where} \quad \boldsymbol{B} := -\boldsymbol{I}_{K-1} \in \mathbb{R}^{(K-1)\times(K-1)} \text{ and } \boldsymbol{C} := [1, 1, \ldots, 1] \in \mathbb{R}^{1\times(K-1)}.
$$

Then, the ODE. (36) can be equivalently think column-by-column, the $j$–th column of $\bar{\boldsymbol{A}}(t)$ solves

$$
\frac{\mathrm{d}}{\mathrm{d}t}\bar{\boldsymbol{a}}(t) = \boldsymbol{A}\bar{\boldsymbol{a}}(t) \quad \text{where} \quad \boldsymbol{a}(0) = \boldsymbol{e}_j.
$$

We use the block structure to split $\bar{\boldsymbol{a}}(t) \in \mathbb{R}^K$ into two parts, i.e., $\bar{\boldsymbol{a}}(t) = [\bar{\boldsymbol{a}}_1(t), \bar{\boldsymbol{a}}_K(t)]$ where $\mathbf{q}_1(t) \in \mathbb{R}^{K-1}$ and $\boldsymbol{a}_K(t) \in \mathbb{R}$ denotes the last coordinate. Under this condition, we have

$$
\frac{\mathrm{d}}{\mathrm{d}t}\bar{\boldsymbol{a}}_1(t) = \boldsymbol{B}\bar{\boldsymbol{a}}_1(t) + \boldsymbol{0} \cdot \bar{\boldsymbol{a}}_K(t) = \boldsymbol{B}\bar{\boldsymbol{a}}_1(t).
$$

According to the definition of $\boldsymbol{B} = -\boldsymbol{I}_{K-1}$, we have

$$
\frac{\mathrm{d}}{\mathrm{d}t}\bar{\boldsymbol{a}}_1(t) = -\bar{\boldsymbol{a}}_1(t) \quad \Rightarrow \bar{\boldsymbol{a}}_1(t) = e^{-t}\bar{\boldsymbol{a}}_1(0).
$$

If we consider the solution of $\bar{\boldsymbol{a}}_K(t)$, it has

$$
\frac{\mathrm{d}}{\mathrm{d}t}\bar{\boldsymbol{a}}_K(t) = \boldsymbol{C} \cdot \bar{\boldsymbol{a}}_1(t) + \boldsymbol{0} \cdot \bar{\boldsymbol{a}}_K(t) = \boldsymbol{C} \cdot e^{-t} \cdot \bar{\boldsymbol{a}}_1(0).
$$

For the initial condition, i.e., $\bar{\boldsymbol{a}}(0) = \boldsymbol{e}_j$, where $j \in \{1, 2, \ldots, K-1\}$ and $\boldsymbol{C} \cdot \bar{\boldsymbol{a}}_1(0) = 1$, then it has

$$
\frac{\mathrm{d}}{\mathrm{d}t}\bar{\boldsymbol{a}}_K(t) = \boldsymbol{C} \cdot \bar{\boldsymbol{a}}_1(t) + \boldsymbol{0} \cdot \bar{\boldsymbol{a}}_K(t) = e^{-t},
$$

which implies

$$\bar{a}_K(t) = \bar{a}_K(0) + 1 - e^{-t} = 1 - e^{-t}.$$

For the initial condition, $\bar{a}(0) = e_K$, we have $C \cdot \bar{a}_1(0) = 0$ and

$$\bar{a}_K(t) = \bar{a}_K(0) + 0 = 1.$$

Therefore, we have

$$\exp(t\boldsymbol{A}) = \begin{bmatrix} e^{-t} & 0 & \ldots & 0 & 0 \\ 0 & e^{-t} & \ldots & 0 & 0 \\ \vdots & \vdots & \ddots & \vdots & \vdots \\ 1 - e^{-t} & 1 - e^{-t} & \ldots & 1 - e^{-t} & 1 \end{bmatrix}.$$

$\square$

**Lemma 8** (Forward transition kernel). *Consider the forward CTMC, i.e., $\{\mathbf{y}_t\}_{t=0}^T$ with the infinitesimal operator $R^{\rightarrow}$ given in Eq. (7). Then, for any two timestamps $s \leq t$, the forward transition probability satisfies, for any $\boldsymbol{y}, \boldsymbol{y}' \in \mathcal{Y}$,*

$$\vec{q}_{t|s}(\boldsymbol{y}|\boldsymbol{y}') = \prod_{i=1}^d \Big[ \delta_{(K,K)}(\boldsymbol{y}_i, \boldsymbol{y}'_i) + \big(1 - \delta_{(K,K)}(\boldsymbol{y}_i, \boldsymbol{y}'_i)\big) \cdot \delta_0(\boldsymbol{y}_i - \boldsymbol{y}'_i) \cdot e^{-(t-s)}$$

$$+ \big(1 - \delta_{(K,K)}(\boldsymbol{y}_i, \boldsymbol{y}'_i)\big) \cdot \delta_K(\boldsymbol{y}_i) \cdot (1 - e^{-(t-s)}) \Big]. \tag{37}$$

*Proof.* Under the matrix presentation, Eq. (26) implies the transition matrix $\boldsymbol{Q}_{t|s}^{\rightarrow}$ can be considered as the solution of the ODE

$$\mathrm{d}\boldsymbol{Q}_{t|s}^{\rightarrow}/\mathrm{d}t = \boldsymbol{R}^{\rightarrow} \cdot \boldsymbol{Q}_{t|s}^{\rightarrow} \quad \text{where} \quad \boldsymbol{Q}_{s|s}^{\rightarrow} = \boldsymbol{I}.$$

Combining Lemma 4 and 6, we have

$$\boldsymbol{Q}_{t|s}^{\rightarrow} = \exp\big((t-s)\boldsymbol{R}^{\rightarrow}\big) = \exp\big((t-s)\boldsymbol{A}\big)^{\otimes d}, \tag{38}$$

which implies

$$\boldsymbol{Q}_{t|s}^{\rightarrow} = \begin{bmatrix} e^{-(t-s)} & 0 & \ldots & 0 & 0 \\ 0 & e^{-(t-s)} & \ldots & 0 & 0 \\ \vdots & \vdots & \ddots & \vdots & \vdots \\ 1 - e^{-(t-s)} & 1 - e^{-(t-s)} & \ldots & 1 - e^{-(t-s)} & 1 \end{bmatrix}^{\otimes d}$$

due to the close solution of $\exp((t-s)\boldsymbol{A})$ shown in Lemma 7. Combining this result with the calculation of the Kronecker product Lemma 12, we have

$$\vec{q}_{t|s}(\boldsymbol{y}|\boldsymbol{y}') = \prod_{i=1}^d \Big[ \delta_{(K,K)}(\boldsymbol{y}_i, \boldsymbol{y}'_i) + \big(1 - \delta_{(K,K)}(\boldsymbol{y}_i, \boldsymbol{y}'_i)\big) \cdot \delta_0(\boldsymbol{y}_i - \boldsymbol{y}'_i) \cdot e^{-(t-s)}$$

$$+ \big(1 - \delta_{(K,K)}(\boldsymbol{y}_i, \boldsymbol{y}'_i)\big) \cdot \delta_K(\boldsymbol{y}_i) \cdot (1 - e^{-(t-s)}) \Big].$$

where $\boldsymbol{y}, \boldsymbol{y}' \in \mathcal{Y}$. Hence, the proof is completed. $\square$

*The proof of Lemma 2.* According to Eq. (29), the solution of $\vec{q}_t$ can be calculated as

$$\vec{q}_t = \exp(t\boldsymbol{R}^{\rightarrow}) \cdot \vec{q}_0 = \exp(t\boldsymbol{A})^{\otimes d} \cdot \vec{q}_0 = \begin{bmatrix} e^{-t} & 0 & \ldots & 0 & 0 \\ 0 & e^{-t} & \ldots & 0 & 0 \\ \vdots & \vdots & \ddots & \vdots & \vdots \\ 0 & 0 & \ldots & e^{-t} & 0 \\ 1 - e^{-t} & 1 - e^{-t} & \ldots & 1 - e^{-t} & 1 \end{bmatrix}^{\otimes d} \cdot \vec{q}_0$$

where the first equation follows from Lemma 4, the second equation follows from Lemma 6, and the last equation follows from Lemma 7. With the calculation of the Kronecker product Lemma 12, we have

$$\boldsymbol{q}_t^{\rightarrow}(\boldsymbol{y}) = \sum_{\boldsymbol{y}' \in \mathcal{Y}} \exp(t\boldsymbol{A})^{\otimes d}(\boldsymbol{y}, \boldsymbol{y}') \cdot \boldsymbol{q}_0^{\rightarrow}(\boldsymbol{y}') = \sum_{\boldsymbol{y}'} \left[ \prod_{i=1}^d \exp(t\boldsymbol{A})(\boldsymbol{y}_i, \boldsymbol{y}_i') \right] \cdot \boldsymbol{q}_0^{\rightarrow}(\boldsymbol{y}'). \tag{39}$$

Under this condition, for any $\boldsymbol{y}$, we denote the coordinate set of token $K$ as $\mathcal{K}$ satisfying $\boldsymbol{y}_i = K \quad \forall\, i \in \mathcal{K}(\boldsymbol{y})$, and

$$\boldsymbol{y}_{\mathcal{K}^c(\boldsymbol{y})} = \boldsymbol{y}'_{\mathcal{K}^c(\boldsymbol{y})} \quad \Leftrightarrow \quad \boldsymbol{y}_i = \boldsymbol{y}_i' \,\forall\, i \notin \mathcal{K}(\boldsymbol{y}).$$

Then, Eq. (39) can be rewritten as

$$\boldsymbol{q}_t^{\rightarrow}(\boldsymbol{y}) = \sum_{\boldsymbol{y}'_{\mathcal{K}^c(\boldsymbol{y})} = \boldsymbol{y}_{\mathcal{K}^c(\boldsymbol{y})}} \left[ \prod_{j \notin \mathcal{K}} \exp(t\boldsymbol{A})(\boldsymbol{y}_j, \boldsymbol{y}_j') \cdot \prod_{j \neq i}^d \exp(t\boldsymbol{A})(K, \boldsymbol{y}_j') \right] \cdot \boldsymbol{q}_0^{\rightarrow}(\boldsymbol{y}')$$

$$+ \sum_{\boldsymbol{y}'_{\mathcal{K}^c(\boldsymbol{y})} \neq \boldsymbol{y}_{\mathcal{K}^c(\boldsymbol{y})}} \left[ \prod_{j=1}^d \exp(t\boldsymbol{A})(\boldsymbol{y}_j, \boldsymbol{y}_j') \right] \cdot \boldsymbol{q}_0^{\rightarrow}(\boldsymbol{y}')$$

$$= \sum_{\boldsymbol{y}'_{\mathcal{K}^c(\boldsymbol{y})} = \boldsymbol{y}_{\mathcal{K}^c(\boldsymbol{y})}} \left[ e^{-t \cdot |\mathcal{K}^c(\boldsymbol{y})|} \cdot (1 - e^{-t})^{|\mathcal{K}(\boldsymbol{y})|} \right] \cdot \boldsymbol{q}_0^{\rightarrow}(\boldsymbol{y}')$$

$$\leq e^{-t \cdot (d - \mathrm{numK}(\boldsymbol{y}))} \cdot \sum_{\boldsymbol{y}'_{\mathcal{K}^c(\boldsymbol{y})} = \boldsymbol{y}_{\mathcal{K}^c(\boldsymbol{y})}} \boldsymbol{q}_0^{\rightarrow}(\boldsymbol{y}') \leq \exp(-t \cdot (d - \mathrm{numK}(\boldsymbol{y}))),$$

where the second equation establishes since we have

$$\exp(t\boldsymbol{A})(\boldsymbol{y}_j, \boldsymbol{y}_j') = \begin{cases} e^{-t} & \boldsymbol{y}_j = \boldsymbol{y}_j' \quad \text{and} \quad \boldsymbol{y}_j \neq K \\ \mathbf{1}_K(\boldsymbol{y}_j') \cdot (1 - e^{-t}) + (1 - \mathbf{1}_K(\boldsymbol{y}_j')) & \boldsymbol{y}_j = K \\ 0 & \text{otherwise} \end{cases}.$$

According to the definition of $\tilde{q}(\boldsymbol{y})$, we can calculate the normalizing constant of $\tilde{q}$ as

$$\tilde{Z}_t = \sum_{\boldsymbol{y}} \exp(-t \cdot (d - \mathrm{numK}(\boldsymbol{y}))) = \sum_{i=0}^d \sum_{\mathrm{numK}(\boldsymbol{y}) = i} \exp(-t \cdot (d - i)) = \sum_{i=1}^d C_d^i \cdot e^{-t \cdot i} = (1 + e^{-t})^d.$$

Therefore, the KL divergence between $q_t^{\rightarrow}$ and $\tilde{q}_t$ can be written as

$$\mathrm{KL}\left(\boldsymbol{q}_t^{\rightarrow} \| \tilde{q}_t\right) = \sum_{\boldsymbol{y} \in \mathcal{Y}} q_t^{\rightarrow}(\boldsymbol{y}) \cdot \ln \frac{q_t^{\rightarrow}(\boldsymbol{y})}{\tilde{q}_t(\boldsymbol{y})} = q_t^{\rightarrow}([K, \ldots, K]) \cdot \ln \frac{q_t^{\rightarrow}([K, \ldots, K])}{\tilde{q}_t([K, \ldots, K])} + \sum_{\boldsymbol{y} \neq [K, \ldots, K]} q_t^{\rightarrow}(\boldsymbol{y}) \cdot \ln \frac{q_t^{\rightarrow}(\boldsymbol{y})}{\tilde{q}_t(\boldsymbol{y})}$$

$$\leq \ln \tilde{Z}_t + \sum_{\boldsymbol{y} \neq [K, \ldots, K]} q_t^{\rightarrow}(\boldsymbol{y}) \ln \frac{q_t^{\rightarrow}(\boldsymbol{y})}{\exp(-t \cdot (d - \mathrm{numK}(\boldsymbol{y}))) / \tilde{Z}_t} = \ln \tilde{Z}_t + \sum_{\boldsymbol{y} \neq [K, \ldots, K]} q_t^{\rightarrow}(\boldsymbol{y}) \ln \tilde{Z}_t$$

$$\leq 2 \ln \tilde{Z}_t = 2 \ln \left[ 1 + (1 + e^{-t})^d - 1 \right] \leq 2 \cdot (1 + e^{-t})^d - 2.$$

Suppose we require the TV distance to be small enough, e.g.,

$$\mathrm{KL}\left(\boldsymbol{q}_t^{\rightarrow} \| \tilde{q}_t\right) \leq \epsilon \quad \Leftrightarrow \quad (1 + e^{-t})^d - 1 \leq \epsilon/2 \quad \Leftrightarrow \quad d \ln(1 + e^{-t}) \leq \ln(1 + \epsilon/2),$$

then, since $\ln(1 + c) \leq c$ when $c > 0$, the sufficient condition for the establishment of the above equation is to require

$$d \cdot e^{-t} \leq \ln(1 + \epsilon/2) \quad \Leftrightarrow \quad t \geq \ln(d / \ln(1 + \epsilon/2)) \quad \Leftarrow \quad t \geq \ln(4d/\epsilon),$$

where the last derivation establishes since $\epsilon/4 \leq \ln(1 + \epsilon/2)$ when $\epsilon \leq 1$ without loss of generality. Hence, the proof is completed. $\qquad\square$

## C    Euler Discretization Analysis

By Assumption 2 of Liang et al. (2025a), $\tilde{v}_{t,\boldsymbol{y}}(\boldsymbol{y}') \leq M$.

**[A1]- Score approximation error assumption** The discrete score $\tilde{v}_t$ obtained from Eq. (6) is well-trained, and its estimation error satisfies for the chosen discretization step size $h$, and $T = nh + \delta$:

$$\frac{1}{T-\delta} \sum_{k=0}^{n-1} \int_{kh}^{(k+1)h} \mathbb{E}_{\mathbf{y}_t \sim q_t^{\leftarrow}} \left[ \sum_{\boldsymbol{y} \neq \mathbf{y}_t} R^{\rightarrow}(\mathbf{y}_t, \boldsymbol{y}) D_\phi \left( v_{kh,\mathbf{y}_t}(\boldsymbol{y}) || \tilde{v}_{kh,\mathbf{y}_t}(\boldsymbol{y}) \right) \right] \mathrm{d}t \leq \epsilon_{score}^2.$$

### C.1    Proof of Theorem 1

Consider the Euler-discretization update in Eq. (11):

$$q_{t+\Delta t|t}^{Eu}(\boldsymbol{y}'|\boldsymbol{y}) \propto \delta_{\boldsymbol{y}}(\boldsymbol{y}') + \Delta t \cdot \tilde{R}_t(\boldsymbol{y}', \boldsymbol{y}) = \delta_{\boldsymbol{y}}(\boldsymbol{y}') + \Delta t \cdot R^{\rightarrow}(\boldsymbol{y}, \boldsymbol{y}') \cdot \tilde{v}_{t,\boldsymbol{y}}(\boldsymbol{y}')$$

Without loss of generality, assume that $\tilde{R}_t(\boldsymbol{y}', \boldsymbol{y})^{\top}$ satisfies the two sufficient conditions of the transition rate matrix: its off-diagonal entries are non-negative, and each row sums to zero [1]. In this way, both $e^{h\tilde{R}_t}$ and $I + h\tilde{R}_t$ are the transpose of valid transition matrices. The probability transition matrix of the Euler discretization can then be written as $Q_{t,t+h}^{Eu} = I + h\tilde{R}_t^{\top}$, where each element can be written as

$$Q_{t,t+h}^{Eu}(\boldsymbol{y}, \boldsymbol{y}') = q_{t+h|t}^{Eu}(\boldsymbol{y}'|\boldsymbol{y}) = \delta_{\boldsymbol{y}}(\boldsymbol{y}') + h \cdot \tilde{R}_t(\boldsymbol{y}', \boldsymbol{y}) \tag{40}$$

To prove the convergence bound for TV $\left(q_\delta^{\leftarrow}, q_{T-\delta}^{Eu}\right)$, we introduce an auxiliary process $q^{EI}$ using the exponential integrator update $Q_{t,t+h}^{Eu} = e^{h\tilde{R}_t^{\top}}$ (Zhang et al., 2024). We first prove the bound for TV $\left(q_{T-\delta}^{Eu}, q_{T-\delta}^{EI}\right)$ and TV $\left(q_\delta^{\leftarrow}, q_{T-\delta}^{EI}\right)$ separately, and use the triangle inequality to conclude the proof. Take $T = nh + \delta$.

**Bound for** TV $\left(q_{T-\delta}^{Eu}, q_{T-\delta}^{EI}\right)$.    For time interval $[kh, (k+1)h]$, by the chain rule of TV distance (Lemma 16), we have

$$\text{TV} \left(q_{(k+1)h}^{Eu}, q_{(k+1)h}^{EI}\right) \leq \text{TV} \left(q_{kh}^{Eu}, q_{kh}^{EI}\right) + \mathbb{E}_{\boldsymbol{y} \sim q_{kh}^{Eu}} \text{TV} \left(q_{(k+1)h|kh}^{Eu}(\cdot \mid \boldsymbol{y}), q_{(k+1)h|kh}^{EI}(\cdot \mid \boldsymbol{y})\right) \tag{41}$$

By the definition of total variation distance, we have

$$\text{TV} \left(q_{(k+1)h|kh}^{Eu}(\cdot \mid \boldsymbol{y}), q_{(k+1)h|kh}^{EI}(\cdot \mid \boldsymbol{y})\right) = \sum_{\boldsymbol{y}'} \left| q_{(k+1)h|kh}^{Eu}(\boldsymbol{y}' \mid \boldsymbol{y}) - q_{(k+1)h|kh}^{EI}(\boldsymbol{y}' \mid \boldsymbol{y}) \right|$$

$$= \sum_{\boldsymbol{y}'} \left| Q_{kh,(k+1)h}^{Eu}(\boldsymbol{y}, \boldsymbol{y}') - Q_{kh,(k+1)h}^{EI}(\boldsymbol{y}, \boldsymbol{y}') \right| \tag{42}$$

Writing out the difference between $Q_{kh,(k+1)h}^{Eu} = I + h\tilde{R}_{kh}^{\top}$ and $Q_{kh,(k+1)h}^{EI} = e^{h\tilde{R}_{kh}^{\top}}$ using the Taylor series expansion for the matrix exponential:

$$Q_{kh,(k+1)h}^{EI} = e^{h\tilde{R}_{kh}^{\top}} = \sum_{i=0}^{\infty} \frac{1}{i!}(h\tilde{R}_{kh}^{\top})^i = I + h\tilde{R}_{kh}^{\top} + \frac{1}{2!}h^2(\tilde{R}_{kh}^{\top})^2 + \frac{1}{3!}h^3(\tilde{R}_{kh}^{\top})^3 + \ldots,$$

we have

$$Q_{kh,(k+1)h}^{EI} - Q_{kh,(k+1)h}^{Eu} = e^{h\tilde{R}_{kh}^{\top}} - \left(I + h\tilde{R}_{kh}^{\top}\right) = \sum_{i=2}^{\infty} \frac{1}{i!}(h\tilde{R}_{kh}^{\top})^i.$$

---

[1]Notice that our notation of $R$ is the *transpose* of the convention used in some other works.

Thus, by the triangle inequality, we have

$$\sum_{\boldsymbol{y}'\in\mathcal{Y}}\left|Q^{EI}_{kh,(k+1)h}(\boldsymbol{y},\boldsymbol{y}') - Q^{Eu}_{kh,(k+1)h}(\boldsymbol{y},\boldsymbol{y}')\right| = \sum_{\boldsymbol{y}'\in\mathcal{Y}}\left|\sum_{i=2}^{\infty}\frac{1}{i!}\left((h\tilde{R}^{\top}_{kh})^i\right)(\boldsymbol{y},\boldsymbol{y}')\right|$$

$$\leq \sum_{\boldsymbol{y}'\in\mathcal{Y}}\sum_{i=2}^{\infty}\frac{h^i}{i!}\left|\left((\tilde{R}^{\top}_{kh})^i\right)(\boldsymbol{y},\boldsymbol{y}')\right|$$

$$= \sum_{i=2}^{\infty}\frac{h^i}{i!}\sum_{\boldsymbol{y}'\in\mathcal{Y}}\left|\left((\tilde{R}^{\top}_{kh})^i\right)(\boldsymbol{y},\boldsymbol{y}')\right| \qquad\text{(Tonelli's theorem for series)}$$

$$= \sum_{i=2}^{\infty}\frac{h^i}{i!}\sum_{\boldsymbol{y}'\in\mathcal{Y}}\left|\left((\tilde{R}_{kh})^i\right)(\boldsymbol{y}',\boldsymbol{y})\right| \leq \sum_{i=2}^{\infty}\frac{h^i}{i!}\left\|(\tilde{R}_{kh})^i\right\|_1 \leq \sum_{i=2}^{\infty}\frac{h^i}{i!}\left\|\tilde{R}_{kh}\right\|_1^i,$$

where $\|A\|_1 = \max_{1\leq j\leq n}\sum_{i=1}^{m}|a_{i,j}| = \max_{x\neq\boldsymbol{0}}\|Ax\|_1/\|x\|_1$ denotes the 1-norm of the matrix. And the last inequality is due to the multiplicative property of this matrix norm.

Therefore,

$$\sum_{\boldsymbol{y}'\in\mathcal{Y}}\left|Q^{EI}_{kh,(k+1)h}(\boldsymbol{y},\boldsymbol{y}') - Q^{Eu}_{kh,(k+1)h}(\boldsymbol{y},\boldsymbol{y}')\right| \leq \sum_{i=2}^{\infty}\frac{h^i}{i!}\left\|\tilde{R}_{kh}\right\|_1^i = e^{h\left\|\tilde{R}_{kh}\right\|_1} - 1 - h\left\|\tilde{R}_{kh}\right\|_1$$

$$\leq \left(h\left\|\tilde{R}_{kh}\right\|_1\right)^2,$$

when $h\left\|\tilde{R}_{kh}\right\|_1 \leq 1$. Plugging this into Eq. (41) and (42), we have

$$\text{TV}\left(q^{Eu}_{(k+1)h}, q^{EI}_{(k+1)h}\right) \leq \text{TV}\left(q^{Eu}_{kh}, q^{EI}_{kh}\right) + \left(h\left\|\tilde{R}_{kh}\right\|_1\right)^2, \qquad (43)$$

when $h\left\|\tilde{R}_{kh}\right\|_1 \leq 1$.

By Assumption 2 of Liang et al. (2025a), $\tilde{v}_{t,\boldsymbol{y}}(\boldsymbol{y}') \leq M$. We have

$$\left\|\tilde{R}_t\right\|_1 = \max_{\boldsymbol{y}}\sum_{\boldsymbol{y}'\in\mathcal{Y}}\left|\tilde{R}_t(\boldsymbol{y}',\boldsymbol{y})\right| = \max_{\boldsymbol{y}}\sum_{\boldsymbol{y}'\in\mathcal{Y}}\left|R^{\rightarrow}(\boldsymbol{y},\boldsymbol{y}')\cdot\tilde{v}_{t,\boldsymbol{y}}(\boldsymbol{y}')\right|$$

$$= \max_{\boldsymbol{y}}\left((d - \text{numK}(\boldsymbol{y})) + \sum_{\text{Ham}(\boldsymbol{y},\boldsymbol{y}')=1 \text{ and } \boldsymbol{y}_{\text{DiffIdx}(\boldsymbol{y},\boldsymbol{y}')}=K}|\tilde{v}_{t,\boldsymbol{y}}(\boldsymbol{y}')|\right) \quad\text{(By Eq. 7)}$$

$$\leq \max_{\boldsymbol{y}}\left(d - \text{numK}(\boldsymbol{y}) + KdM\right)$$

$$\leq 2KdM.$$

Thus $\left\|\tilde{R}_{kh}\right\|_1 \leq 2KdM$. By (43) we have

$$\text{TV}\left(q^{Eu}_{nh}, q^{EI}_{nh}\right) \leq \text{TV}\left(q^{Eu}_0, q^{EI}_0\right) + \sum_{k=1}^{n}\left(h\left\|\tilde{R}_{kh}\right\|_1\right)^2$$

$$\leq K^2d^2\sum_{k=1}^{n}h^2M^2 \leq K^2d^2nh^2M^2 \leq K^2(T-\delta)hd^2M^2. \qquad (44)$$

By taking $h \leq \frac{\varepsilon}{K^2d^2M^2\log(d/\varepsilon)}$, then $\text{TV}\left(q^{Eu}_{nh}, q^{EI}_{nh}\right) \leq \varepsilon$.

**Bound for** $\text{TV}\left(q^{EI}_{T-\delta}, q^{\leftarrow}_{T-\delta}\right)$**.** We first prove $\text{KL}\left(q^{\leftarrow}_{T-\delta}\|q^{EI}_{T-\delta}\right)$, then use Pinsker's inequality to derive the bound for $\text{TV}\left(q^{EI}_{T-\delta}, q^{\leftarrow}_{T-\delta}\right)$.

For time interval $[kh, (k+1)h]$, we have

$$\mathrm{KL}\left(q^{\leftarrow}_{(k+1)h}\big\|q^{EI}_{(k+1)h}\right) = \mathrm{KL}\left(q^{\leftarrow}_{kh}\big\|q^{EI}_{kh}\right) + \int_{kh}^{(k+1)h} \frac{\mathrm{dKL}\left(q^{\leftarrow}_t\big\|q^{EI}_t\right)}{\mathrm{d}t}\mathrm{d}t. \tag{45}$$

By the chain rule of KL divergence (Lemma 15)

$$\begin{aligned}
\frac{\mathrm{d}}{\mathrm{d}t}\mathrm{KL}\left(q^{\leftarrow}_t\big\|q^{EI}_t\right) &= \lim_{\Delta t\to 0}\frac{\mathrm{KL}\left(q^{\leftarrow}_{t+\Delta t}\big\|q^{EI}_{t+\Delta t}\right) - \mathrm{KL}\left(q^{\leftarrow}_t\big\|q^{EI}_t\right)}{\Delta t}\\
&\leq \lim_{\Delta t\to 0}\mathbb{E}_{\boldsymbol{y}\sim q^{\leftarrow}_t}\frac{\mathrm{KL}\left(q^{\leftarrow}_{t+\Delta t|t}(\cdot\mid\boldsymbol{y})\big\|q^{EI}_{t+\Delta t|t}(\cdot\mid\boldsymbol{y})\right)}{\Delta t}\\
&= \mathbb{E}_{\boldsymbol{y}\sim q^{\leftarrow}_t}\underbrace{\lim_{\Delta t\to 0}\frac{\mathrm{KL}\left(q^{\leftarrow}_{t+\Delta t|t}(\cdot\mid\boldsymbol{y})\big\|q^{EI}_{t+\Delta t|t}(\cdot\mid\boldsymbol{y})\right)}{\Delta t}}_{\text{Term 1}}
\end{aligned} \tag{46}$$

For each $\boldsymbol{y}\in\mathcal{Y}$, we focus on Term 1 of Eq. (46), and have

$$\begin{aligned}
\text{Term 1} &= \lim_{\Delta t\to 0}\left[\Delta t^{-1}\cdot\sum_{\boldsymbol{y}'\in\mathcal{Y}}q^{\leftarrow}_{t+\Delta t|t}(\boldsymbol{y}'|\boldsymbol{y})\cdot\ln\frac{q^{\leftarrow}_{t+\Delta t|t}(\boldsymbol{y}'|\boldsymbol{y})}{q^{EI}_{t+\Delta t|t}(\boldsymbol{y}'|\boldsymbol{y})}\right]\\
&= \underbrace{\lim_{\Delta t\to 0}\left[\sum_{\boldsymbol{y}'\neq\boldsymbol{y}}\frac{q^{\leftarrow}_{t+\Delta t|t}(\boldsymbol{y}'|\boldsymbol{y})}{\Delta t}\cdot\ln\frac{q^{\leftarrow}_{t+\Delta t|t}(\boldsymbol{y}'|\boldsymbol{y})}{q^{EI}_{t+\Delta t|t}(\boldsymbol{y}'|\boldsymbol{y})}\right]}_{\text{Term 1.1}} +\\
&\quad\underbrace{\lim_{\Delta t\to 0}\left[\Delta t^{-1}\cdot\left(1-\sum_{\boldsymbol{y}'\neq\boldsymbol{y}}q^{\leftarrow}_{t+\Delta t|t}(\boldsymbol{y}'|\boldsymbol{y})\right)\cdot\ln\frac{1-\sum_{\boldsymbol{y}'\neq\boldsymbol{y}}q^{\leftarrow}_{t+\Delta t|t}(\boldsymbol{y}'|\boldsymbol{y})}{1-\sum_{\boldsymbol{y}'\neq\boldsymbol{y}}q^{EI}_{t+\Delta t|t}(\boldsymbol{y}'|\boldsymbol{y})}\right]}_{\text{Term 1.2}}.
\end{aligned} \tag{47}$$

For Term 1.1, we have

$$\begin{aligned}
\text{Term 1.1} &= \sum_{\boldsymbol{y}'\neq\boldsymbol{y}}\lim_{\Delta t\to 0}\left[\frac{q^{\leftarrow}_{t+\Delta t|t}(\boldsymbol{y}'|\boldsymbol{y})}{\Delta t}\right]\cdot\lim_{\Delta t\to 0}\left[\ln\frac{q^{\leftarrow}_{t+\Delta t|t}(\boldsymbol{y}'|\boldsymbol{y})}{q^{EI}_{t+\Delta t|t}(\boldsymbol{y}'|\boldsymbol{y})}\right]\\
&= \sum_{\boldsymbol{y}'\neq\boldsymbol{y}}R^{\leftarrow}_t(\boldsymbol{y}',\boldsymbol{y})\cdot\ln\left[\lim_{\Delta t\to 0}\left(\frac{q^{\leftarrow}_{t+\Delta t|t}(\boldsymbol{y}'|\boldsymbol{y})}{\Delta t}\cdot\frac{\Delta t}{q^{EI}_{t+\Delta t|t}(\boldsymbol{y}'|\boldsymbol{y})}\right)\right]\\
&= \sum_{\boldsymbol{y}'\neq\boldsymbol{y}}R^{\leftarrow}_t(\boldsymbol{y}',\boldsymbol{y})\cdot\ln\frac{R^{\leftarrow}_t(\boldsymbol{y}',\boldsymbol{y})}{\tilde{R}_{kh}(\boldsymbol{y}',\boldsymbol{y})},
\end{aligned} \tag{48}$$

where the second equation follows from the composition rule of the limit calculation. For Term 1.2, we have

$$\begin{aligned}
\text{Term 1.2} &= \lim_{\Delta t\to 0}\left[1-\sum_{\boldsymbol{y}'\neq\boldsymbol{y}}q^{\leftarrow}_{t+\Delta t|t}(\boldsymbol{y}'|\boldsymbol{y})\right]\cdot\lim_{\Delta t\to 0}\left[\Delta t^{-1}\cdot\ln\frac{1-\sum_{\boldsymbol{y}'\neq\boldsymbol{y}}q^{\leftarrow}_{t+\Delta t|t}(\boldsymbol{y}'|\boldsymbol{y})}{1-\sum_{\boldsymbol{y}'\neq\boldsymbol{y}}q^{EI}_{t+\Delta t|t}(\boldsymbol{y}'|\boldsymbol{y})}\right]\\
&= \sum_{\boldsymbol{y}'\neq\boldsymbol{y}}\left(\tilde{R}_{kh}(\boldsymbol{y}',\boldsymbol{y}) - R^{\leftarrow}_t(\boldsymbol{y}',\boldsymbol{y})\right) = \tilde{R}_{kh}(\boldsymbol{y}) - R^{\leftarrow}_t(\boldsymbol{y})
\end{aligned} \tag{49}$$

where the first inequality follows from Lemma 9. Plugging Eq. (48), Eq. (49) and Eq. (47), into Eq. (46) we have

$$
\frac{\mathrm{dKL}\left(q_t^{\leftarrow}\big\|q_t^{EI}\right)}{\mathrm{d}t} \leq \sum_{\boldsymbol{y}\in\mathcal{Y}} q_t^{\leftarrow}(\boldsymbol{y}) \cdot \left( \sum_{\boldsymbol{y}'\neq\boldsymbol{y}} R_t^{\leftarrow}(\boldsymbol{y}',\boldsymbol{y}) \cdot \ln \frac{R_t^{\leftarrow}(\boldsymbol{y}',\boldsymbol{y})}{\tilde{R}_{kh}(\boldsymbol{y}',\boldsymbol{y})} + \tilde{R}_{kh}(\boldsymbol{y}) - R_t^{\leftarrow}(\boldsymbol{y}) \right)
$$

$$
= \sum_{\boldsymbol{y}\in\mathcal{Y}} q_t^{\leftarrow}(\boldsymbol{y}) \cdot \left( \sum_{\boldsymbol{y}'\neq\boldsymbol{y}} R_t^{\leftarrow}(\boldsymbol{y}',\boldsymbol{y}) \cdot \ln \frac{R_t^{\leftarrow}(\boldsymbol{y}',\boldsymbol{y})}{\tilde{R}_{kh}(\boldsymbol{y}',\boldsymbol{y})} + \sum_{\boldsymbol{y}'\neq\boldsymbol{y}} \tilde{R}_{kh}(\boldsymbol{y}',\boldsymbol{y}) - \sum_{\boldsymbol{y}'\neq\boldsymbol{y}} R_t^{\leftarrow}(\boldsymbol{y}',\boldsymbol{y}) \right)
$$

$$
= \sum_{\boldsymbol{y}\in\mathcal{Y}} q_t^{\leftarrow}(\boldsymbol{y}) \cdot \sum_{\boldsymbol{y}'\neq\boldsymbol{y}} R^{\rightarrow}(\boldsymbol{y},\boldsymbol{y}') \cdot \left[ -\frac{q_t^{\leftarrow}(\boldsymbol{y}')}{q_t^{\leftarrow}(\boldsymbol{y})} + \tilde{v}_{kh,\boldsymbol{y}}(\boldsymbol{y}') + \frac{q_t^{\leftarrow}(\boldsymbol{y}')}{q_t^{\leftarrow}(\boldsymbol{y})} \ln \frac{q_t^{\leftarrow}(\boldsymbol{y}')}{q_t^{\leftarrow}(\boldsymbol{y})\tilde{v}_{kh,\boldsymbol{y}}(\boldsymbol{y}')} \right]
$$

$$
= \sum_{\boldsymbol{y}\in\mathcal{Y}} q_t^{\leftarrow}(\boldsymbol{y}) \cdot \sum_{\boldsymbol{y}'\neq\boldsymbol{y}} R^{\rightarrow}(\boldsymbol{y},\boldsymbol{y}') \cdot \left[ -v_{t,\boldsymbol{y}}(\boldsymbol{y}') + \tilde{v}_{kh,\boldsymbol{y}}(\boldsymbol{y}') + v_{t,\boldsymbol{y}}(\boldsymbol{y}') \ln \frac{v_{t,\boldsymbol{y}}(\boldsymbol{y}')}{\tilde{v}_{kh,\boldsymbol{y}}(\boldsymbol{y}')} \right]
$$

$$
= \sum_{\boldsymbol{y}\in\mathcal{Y}} q_t^{\leftarrow}(\boldsymbol{y}) \cdot \underbrace{\sum_{\mathrm{Ham}(\boldsymbol{y},\boldsymbol{y}')=1 \text{ and } \boldsymbol{y}_{\mathrm{DiffIdx}(\boldsymbol{y},\boldsymbol{y}')}=K} \left[ -v_{t,\boldsymbol{y}}(\boldsymbol{y}') + \tilde{v}_{kh,\boldsymbol{y}}(\boldsymbol{y}') + v_{t,\boldsymbol{y}}(\boldsymbol{y}') \ln \frac{v_{t,\boldsymbol{y}}(\boldsymbol{y}')}{\tilde{v}_{kh,\boldsymbol{y}}(\boldsymbol{y}')} \right]}_{\text{Term 2}}
$$

(50)

For $\boldsymbol{y}' = \boldsymbol{y}[\boldsymbol{y}_i \to k]$, by Eq. (55) we have $v_{t,\boldsymbol{y}}(\boldsymbol{y}') = \frac{q_t^{\leftarrow}(\boldsymbol{y}[\boldsymbol{y}_i\to k])}{q_t^{\leftarrow}(\boldsymbol{y})} \leq \frac{1}{e^{(T-t)}-1}$. By (Liang et al., 2025a, Lemma 2), there exist $c > 0$ such that $v_{t,\boldsymbol{y}}(\boldsymbol{y}') \geq \frac{1}{c}e^{-(T-t)}$. Therefore, by (Zhang et al., 2024, Proposition 3), letting $C = \max\{M, ce^T\}$, Term 2 satisfies

$$
\text{Term 2} \leq \sum_{\mathrm{Ham}(\boldsymbol{y},\boldsymbol{y}')=1 \text{ and } \boldsymbol{y}_{\mathrm{DiffIdx}(\boldsymbol{y},\boldsymbol{y}')}=K} \left( C \left\| v_{t,\boldsymbol{y}}(\boldsymbol{y}') - v_{kh,\boldsymbol{y}}(\boldsymbol{y}') \right\|^2 + 2C^2 D_\phi(v_{kh,\boldsymbol{y}}(\boldsymbol{y}')\|\tilde{v}_{kh,\boldsymbol{y}}(\boldsymbol{y}')) \right)
$$

where $D_\phi$ is the Bregman divergence with $\phi(x) = x\ln x$ (as Eq. (6)), i.e.,

$$
D_\phi(u\|v) = \phi(u) - \phi(v) - \langle \nabla\phi(v), u - v \rangle = u\ln\frac{u}{v} - u + v.
$$

By (Liang et al., 2025a, Lemma 7), we have $\|v_{t,\boldsymbol{y}}(\boldsymbol{y}') - v_{kh,\boldsymbol{y}}(\boldsymbol{y}')\| \lesssim \gamma^{-1}(t - kh) \lesssim h$, where $\gamma$ is defined in (Liang et al., 2025a, Assumption 4). We therefore have

$$
\text{Term 2} \lesssim CK\mathrm{numK}(\boldsymbol{y})h^2 + C^2 \sum_{\mathrm{Ham}(\boldsymbol{y},\boldsymbol{y}')=1 \text{ and } \boldsymbol{y}_{\mathrm{DiffIdx}(\boldsymbol{y},\boldsymbol{y}')}=K} (D_\phi(v_{kh,\boldsymbol{y}}(\boldsymbol{y}')\|\tilde{v}_{kh,\boldsymbol{y}}(\boldsymbol{y}')))
$$

$$
\lesssim CKdh^2 + C^2 \sum_{\mathrm{Ham}(\boldsymbol{y},\boldsymbol{y}')=1 \text{ and } \boldsymbol{y}_{\mathrm{DiffIdx}(\boldsymbol{y},\boldsymbol{y}')}=K} (D_\phi(v_{kh,\boldsymbol{y}}(\boldsymbol{y}')\|\tilde{v}_{kh,\boldsymbol{y}}(\boldsymbol{y}')))
$$

(51)

Since by Eq. (45), we have

$$
\mathrm{KL}\left(q_{nh}^{\leftarrow}\big\|q_{nh}^{EI}\right) = \mathrm{KL}\left(q_0^{\leftarrow}\big\|q_0^{EI}\right) + \sum_{k=0}^{n-1} \int_{kh}^{(k+1)h} \frac{\mathrm{dKL}\left(q_t^{\leftarrow}\big\|q_t^{EI}\right)}{\mathrm{d}t}\mathrm{d}t.
$$

Then, by Eq. (50), Eq. (51), Eq. (6) and Assumption **[A1]-**, we have

$$
\mathrm{KL}\left(q_{T-\delta}^{\leftarrow}\big\|q_{T-\delta}^{EI}\right) \lesssim (T-\delta)C^2\epsilon_{\text{score}}^2 + C(T-\delta)Kdh^2.
$$

By Pinsker's inequality, we have

$$
\mathrm{TV}\left(q_{T-\delta}^{\leftarrow}, q_{T-\delta}^{EI}\right) \leq \sqrt{\frac{1}{2}\mathrm{KL}\left(q_{T-\delta}^{\leftarrow}\big\|q_{T-\delta}^{EI}\right)} \lesssim \sqrt{\frac{1}{2}}\sqrt{(T-\delta)C^2\epsilon_{\text{score}}^2 + C(T-\delta)Kdh^2}
$$

By taking $\epsilon_{\text{score}} \lesssim \varepsilon/(\sqrt{T}C)$, and $h \lesssim \varepsilon/\sqrt{CdT}$, we have TV $\left(q_{T-\delta}^{\leftarrow}, q_{T-\delta}^{EI}\right) \leq \varepsilon$.

Therefore, taking $h \lesssim \min\{\frac{\varepsilon}{K^2 d^2 M^2 \log(d/\varepsilon)}, \frac{\varepsilon}{\sqrt{Cd\log(d/\varepsilon)}}\}$, by the triangle inequality, we have

$$\text{TV}\left(q_\delta^{\leftarrow}, q_{T-\delta}^{Eu}\right) \leq \text{TV}\left(q_{T-\delta}^{Eu}, q_{T-\delta}^{EI}\right) + \text{TV}\left(q_\delta^{\leftarrow}, q_{T-\delta}^{EI}\right) \lesssim \varepsilon.$$

Plugging in $C = \Theta(d/\varepsilon)$, we have for $h \lesssim \min\{\frac{\varepsilon}{K^2 d^2 M^2 \log(d/\varepsilon)}, \frac{\varepsilon^{\frac{3}{2}}}{d\sqrt{\log(d/\varepsilon)}}\}$, we have TV $\left(q_\delta^{\leftarrow}, q_{T-\delta}^{Eu}\right) \lesssim \varepsilon$.

Hence, the proof is completed.

**Lemma 9.** *Following the notations shown in Section 2, for $t \in [kh, (k+1)h]$, we have*

$$\lim_{\Delta t \to 0}\left[\Delta t^{-1} \cdot \ln \frac{1 - \sum_{\boldsymbol{y}' \neq \boldsymbol{y}} q_{t+\Delta t|t}^{\leftarrow}(\boldsymbol{y}'|\boldsymbol{y})}{1 - \sum_{\boldsymbol{y}' \neq \boldsymbol{y}} q_{t+\Delta t|t}^{EI}(\boldsymbol{y}'|\boldsymbol{y})}\right] = \tilde{R}_{kh}(\boldsymbol{y}) - R_t^{\leftarrow}(\boldsymbol{y}).$$

*Proof.* Since we have required $\Delta t \to 0$, that is to say

$$q_{t+\Delta t|t}^{EI}(\boldsymbol{y}'|\boldsymbol{y}) \to q_{t|t}^{EI}(\boldsymbol{y}'|\boldsymbol{y}) = 0 \quad \text{and} \quad q_{t+\Delta t|t}^{\leftarrow}(\boldsymbol{y}'|\boldsymbol{y}) \to q_{t|t}^{\leftarrow}(\boldsymbol{y}'|\boldsymbol{y}) = 0 \quad \forall \boldsymbol{y}' \neq \boldsymbol{y},$$

which automatically makes

$$\left|\frac{\sum_{\boldsymbol{y}' \neq \boldsymbol{y}} \left(q_{t+\Delta t|t}^{EI}(\boldsymbol{y}'|\boldsymbol{y}) - q_{t+\Delta t|t}^{\leftarrow}(\boldsymbol{y}'|\boldsymbol{y})\right)}{1 - \sum_{\boldsymbol{y}' \neq \boldsymbol{y}} q_{t+\Delta t|t}^{EI}(\boldsymbol{y}'|\boldsymbol{y})}\right| \leq \frac{1}{2} < 1.$$

Under this condition, we have

$$\ln \frac{1 - \sum_{\boldsymbol{y}' \neq \boldsymbol{y}} q_{t+\Delta t|t}^{\leftarrow}(\boldsymbol{y}'|\boldsymbol{y})}{1 - \sum_{\boldsymbol{y}' \neq \boldsymbol{y}} q_{t+\Delta t|t}^{EI}(\boldsymbol{y}'|\boldsymbol{y})} = \ln\left[1 + \frac{\sum_{\boldsymbol{y}' \neq \boldsymbol{y}} \left(q_{t+\Delta t|t}^{EI}(\boldsymbol{y}'|\boldsymbol{y}) - q_{t+\Delta t|t}^{\leftarrow}(\boldsymbol{y}'|\boldsymbol{y})\right)}{1 - \sum_{\boldsymbol{y}' \neq \boldsymbol{y}} q_{t+\Delta t|t}^{EI}(\boldsymbol{y}'|\boldsymbol{y})}\right]$$

$$= \sum_{i=1}^{\infty} \frac{(-1)^{i+1}}{i} \cdot \left[\frac{\sum_{\boldsymbol{y}' \neq \boldsymbol{y}} \left(q_{t+\Delta t|t}^{EI}(\boldsymbol{y}'|\boldsymbol{y}) - q_{t+\Delta t|t}^{\leftarrow}(\boldsymbol{y}'|\boldsymbol{y})\right)}{1 - \sum_{\boldsymbol{y}' \neq \boldsymbol{y}} q_{t+\Delta t|t}^{EI}(\boldsymbol{y}'|\boldsymbol{y})}\right]^i,$$

which implies (with the dominated convergence theorem)

$$\lim_{\Delta t \to 0}\left[\Delta t^{-1} \cdot \ln \frac{1 - \sum_{\boldsymbol{y}' \neq \boldsymbol{y}} q_{t+\Delta t|t}^{\leftarrow}(\boldsymbol{y}'|\boldsymbol{y})}{1 - \sum_{\boldsymbol{y}' \neq \boldsymbol{y}} q_{t+\Delta t|t}^{EI}(\boldsymbol{y}'|\boldsymbol{y})}\right]$$

$$= \sum_{i=1}^{\infty} \frac{(-1)^{i+1}}{i} \cdot \lim_{\Delta t \to 0} \frac{\sum_{\boldsymbol{y}' \neq \boldsymbol{y}} \left(q_{t+\Delta t|t}^{EI}(\boldsymbol{y}'|\boldsymbol{y}) - q_{t+\Delta t|t}^{\leftarrow}(\boldsymbol{y}'|\boldsymbol{y})\right)}{\Delta t}$$

$$\cdot \lim_{\Delta t \to 0} \frac{\left(\sum_{\boldsymbol{y}' \neq \boldsymbol{y}} \left(q_{t+\Delta t|t}^{EI}(\boldsymbol{y}'|\boldsymbol{y}) - q_{t+\Delta t|t}^{\leftarrow}(\boldsymbol{y}'|\boldsymbol{y})\right)\right)^{i-1}}{\left(1 - \sum_{\boldsymbol{y}' \neq \boldsymbol{y}} q_{t+\Delta t|t}^{EI}(\boldsymbol{y}'|\boldsymbol{y})\right)^i}.$$

Only when $i = 1$, we have

$$\lim_{\Delta t \to 0} \frac{\left(\sum_{\boldsymbol{y}' \neq \boldsymbol{y}} \left(q_{t+\Delta t|t}^{EI}(\boldsymbol{y}'|\boldsymbol{y}) - q_{t+\Delta t|t}^{\leftarrow}(\boldsymbol{y}'|\boldsymbol{y})\right)\right)^{i-1}}{\left(1 - \sum_{\boldsymbol{y}' \neq \boldsymbol{y}} q_{t+\Delta t|t}^{EI}(\boldsymbol{y}'|\boldsymbol{y})\right)^i} = 1,$$

otherwise it will be equivalent to 0. Therefore, we have

$$\lim_{\Delta t \to 0}\left[\Delta t^{-1} \cdot \ln \frac{1 - \sum_{\boldsymbol{y}' \neq \boldsymbol{y}} q_{t+\Delta t|t}^{\leftarrow}(\boldsymbol{y}'|\boldsymbol{y})}{1 - \sum_{\boldsymbol{y}' \neq \boldsymbol{y}} q_{t+\Delta t|t}^{EI}(\boldsymbol{y}'|\boldsymbol{y})}\right] = \lim_{\Delta t \to 0} \frac{\sum_{\boldsymbol{y}' \neq \boldsymbol{y}} \left(q_{t+\Delta t|t}^{EI}(\boldsymbol{y}'|\boldsymbol{y}) - q_{t+\Delta t|t}^{\leftarrow}(\boldsymbol{y}'|\boldsymbol{y})\right)}{\Delta t}$$

$$= \sum_{\boldsymbol{y}' \neq \boldsymbol{y}} \left(\tilde{R}_{kh}(\boldsymbol{y}', \boldsymbol{y}) - R_t^{\leftarrow}(\boldsymbol{y}', \boldsymbol{y})\right) = \tilde{R}_{kh}(\boldsymbol{y}) - R_t^{\leftarrow}(\boldsymbol{y}).$$

Hence, the proof is completed. $\qquad\square$

## D  TRUNCATED UNIFORMIZATION INFERENCE ANALYSIS

### D.1  THE PROOF OF LEMMA 3

*The proof of Lemma 3.*  According to the definition, we have

$$R_t^{\leftarrow}(\boldsymbol{y}) = \sum_{\boldsymbol{y}' \neq \boldsymbol{y}} R_t^{\leftarrow}(\boldsymbol{y}', \boldsymbol{y}) = \sum_{\boldsymbol{y}' \neq \boldsymbol{y}} R^{\rightarrow}(\boldsymbol{y}, \boldsymbol{y}') \cdot \frac{q_t^{\leftarrow}(\boldsymbol{y}')}{q_t^{\leftarrow}(\boldsymbol{y})}$$

Since the definition of the transition rate matrix, i.e., Eq. (7), for any $\boldsymbol{y}'$ with $\mathrm{Ham}(\boldsymbol{y}', \boldsymbol{y}) > 1$, it has $R^{\rightarrow}(\boldsymbol{y}, \boldsymbol{y}') = 0$. Moreover, even when $\mathrm{Ham}(\boldsymbol{y}', \boldsymbol{y}) = 1$, it has

$$R^{\rightarrow}(\boldsymbol{y}, \boldsymbol{y}') = 0 \quad \text{when} \quad \boldsymbol{y}_{\mathrm{DiffIdx}(\boldsymbol{y}, \boldsymbol{y}')} \neq K.$$

Define the function to transfer the $i$–th element of $\boldsymbol{y}$ ($\boldsymbol{y}_i$) from $k'$ to $k$ as

$$\boldsymbol{y}[\boldsymbol{y}_i \colon k' \to k] = [\boldsymbol{y}_1, \boldsymbol{y}_2, \ldots, \boldsymbol{y}_{i-1}, k, \boldsymbol{y}_{i+1}, \ldots, \boldsymbol{y}_d].$$

That means $R_t^{\leftarrow}(\boldsymbol{y})$ can be rewritten as

$$R_t^{\leftarrow}(\boldsymbol{y}) = \sum_{i, \boldsymbol{y}_i = K} \left[ \sum_{k=1}^{K-1} R^{\rightarrow}(\boldsymbol{y}, \boldsymbol{y}[\boldsymbol{y}_i \colon K \to k]) \cdot \frac{q_t^{\leftarrow}(\boldsymbol{y}[\boldsymbol{y}_i \colon K \to k])}{q_t^{\leftarrow}(\boldsymbol{y})} \right]. \tag{52}$$

To upper bound the RHS of the above equation, we consider controlling

$$\frac{q_t^{\leftarrow}(\boldsymbol{y}[\boldsymbol{y}_i \colon K \to k])}{q_t^{\leftarrow}(\boldsymbol{y})} = \frac{q_{T-t}^{\rightarrow}(\boldsymbol{y}[\boldsymbol{y}_i \colon K \to k])}{q_{T-t}^{\rightarrow}(\boldsymbol{y})} = \frac{\sum_{\boldsymbol{y}_0 \in \mathcal{Y}} q_0^{\rightarrow}(\boldsymbol{y}_0) \cdot q_{T-t|0}^{\rightarrow}(\boldsymbol{y}[\boldsymbol{y}_i \colon K \to k]|\boldsymbol{y}_0)}{\sum_{\boldsymbol{y}_0 \in \mathcal{Y}} q_0^{\rightarrow}(\boldsymbol{y}_0) \cdot q_{T-t|0}^{\rightarrow}(\boldsymbol{y}|\boldsymbol{y}_0)}$$

$$= \frac{\sum_{\boldsymbol{y}_0 \in \mathcal{Y}} q_0^{\rightarrow}(\boldsymbol{y}_0) \cdot q_{T-t|0}^{\rightarrow}(\boldsymbol{y}|\boldsymbol{y}_0) \cdot \frac{q_{T-t|0}^{\rightarrow}(\boldsymbol{y}[\boldsymbol{y}_i \colon K \to k]|\boldsymbol{y}_0)}{q_{T-t|0}^{\rightarrow}(\boldsymbol{y}|\boldsymbol{y}_0)}}{\sum_{\boldsymbol{y}_0 \in \mathcal{Y}} q_0^{\rightarrow}(\boldsymbol{y}_0) \cdot q_{T-t|0}^{\rightarrow}(\boldsymbol{y}|\boldsymbol{y}_0)} = \mathbb{E}_{\boldsymbol{y}_0 \sim q_{0|T-t}^{\rightarrow}(\cdot|\boldsymbol{y})} \left[ \frac{q_{T-t|0}^{\rightarrow}(\boldsymbol{y}[\boldsymbol{y}_i \colon K \to k]|\boldsymbol{y}_0)}{q_{T-t|0}^{\rightarrow}(\boldsymbol{y}|\boldsymbol{y}_0)} \right], \tag{53}$$

where the last equation follows from Bayes' Theorem, i.e.,

$$q_{0|T-t}^{\rightarrow}(\boldsymbol{y}_0|\boldsymbol{y}) \cdot q_{T-t}^{\rightarrow}(\boldsymbol{y}) = q_{T-t|0}^{\rightarrow}(\boldsymbol{y}|\boldsymbol{y}_0) \cdot q_0^{\rightarrow}(\boldsymbol{y}_0) \quad \Leftrightarrow \quad q_{0|T-t}^{\rightarrow}(\boldsymbol{y}_0|\boldsymbol{y}) \propto q_{T-t|0}^{\rightarrow}(\boldsymbol{y}|\boldsymbol{y}_0) \cdot q_0^{\rightarrow}(\boldsymbol{y}_0).$$

Then, we only need to control $q_{T-t|0}^{\rightarrow}(\boldsymbol{y}[\boldsymbol{y}_i \to k]|\boldsymbol{y}_0)/q_{T-t|0}^{\rightarrow}(\boldsymbol{y}|\boldsymbol{y}_0)$ where both the denominator and the numerator can be calculated accurately by Lemma 8. Specifically, we have

$$q_{T-t|0}^{\rightarrow}(\boldsymbol{y}|\boldsymbol{y}_0) = \prod_{j \in \{1, \ldots, i-1, i+1, \ldots, d\}} \left[ \mathbf{1}_{(K,K)}(\boldsymbol{y}_j, \boldsymbol{y}_{0,j}) + \left( 1 - \mathbf{1}_{(K,K)}(\boldsymbol{y}_j, \boldsymbol{y}_{0,j}) \right) \cdot \mathbf{1}_0(\boldsymbol{y}_j - \boldsymbol{y}_{0,j}) \cdot e^{-(T-t)} \right.$$

$$\left. + \left( 1 - \mathbf{1}_{(K,K)}(\boldsymbol{y}_j, \boldsymbol{y}_{0,j}) \right) \cdot \mathbf{1}_K(\boldsymbol{y}_j) \cdot (1 - e^{-(T-t)}) \right] \cdot$$

$$\left[ \mathbf{1}_{(K,K)}(K, \boldsymbol{y}_{0,i}) + \left( 1 - \mathbf{1}_{(K,K)}(K, \boldsymbol{y}_{0,i}) \right) \cdot (1 - e^{-(T-t)}) \right]$$

and

$$q_{T-t|0}^{\rightarrow}(\boldsymbol{y}[\boldsymbol{y}_i \colon K \to k]|\boldsymbol{y}_0) = \prod_{j \in \{1, \ldots, i-1, i+1, \ldots, d\}} \left[ \mathbf{1}_{(K,K)}(\boldsymbol{y}_j, \boldsymbol{y}_{0,j}) + \left( 1 - \mathbf{1}_{(K,K)}(\boldsymbol{y}_j, \boldsymbol{y}_{0,j}) \right) \cdot \mathbf{1}_0(\boldsymbol{y}_j - \boldsymbol{y}_{0,j}) \cdot e^{-(T-t)} \right.$$

$$\left. + \left( 1 - \mathbf{1}_{(K,K)}(\boldsymbol{y}_j, \boldsymbol{y}_{0,j}) \right) \cdot \mathbf{1}_K(\boldsymbol{y}_j) \cdot (1 - e^{-(T-t)}) \right] \cdot$$

$$\left[ \left( 1 - \mathbf{1}_{(K,K)}(k, \boldsymbol{y}_{0,i}) \right) \cdot \mathbf{1}_0(k - \boldsymbol{y}_{0,i}) \cdot e^{-(T-t)} \right].$$

Since the factor except for the $i$–th term will be canceled, we have

$$\frac{q_{T-t|0}^{\rightarrow}(\boldsymbol{y}[\boldsymbol{y}_i \to k]|\boldsymbol{y}_0)}{q_{T-t|0}^{\rightarrow}(\boldsymbol{y}|\boldsymbol{y}_0)} = \frac{\left( 1 - \mathbf{1}_{(K,K)}(k, \boldsymbol{y}_{0,i}) \right) \cdot \mathbf{1}_0(k - \boldsymbol{y}_{0,i}) \cdot e^{-(T-t)}}{\mathbf{1}_{(K,K)}(K, \boldsymbol{y}_{0,i}) + \left( 1 - \mathbf{1}_{(K,K)}(K, \boldsymbol{y}_{0,i}) \right) \cdot (1 - e^{-(T-t)})}$$

$$= \frac{\mathbf{1}_0(k - \boldsymbol{y}_{0,i}) \cdot e^{-(T-t)}}{1 - e^{-(T-t)}} \leq \frac{e^{-(T-t)}}{1 - e^{-(T-t)}} = \frac{1}{e^{(T-t)} - 1}. \tag{54}$$

Plugging this result into Eq. (53), the density ratio of the reverse process will have

$$\frac{q_t^{\leftarrow}(\boldsymbol{y}[\boldsymbol{y}_i \rightarrow k])}{q_t^{\leftarrow}(\boldsymbol{y})} = \mathbb{E}_{\boldsymbol{y}_0 \sim q_{0|T-t}^{\rightarrow}(\cdot|\boldsymbol{y})} \left[ \frac{q_{T-t|0}^{\rightarrow}(\boldsymbol{y}[\boldsymbol{y}_i : K \rightarrow k]|\boldsymbol{y}_0)}{q_{T-t|0}^{\rightarrow}(\boldsymbol{y}|\boldsymbol{y}_0)} \right] \leq \frac{1}{e^{(T-t)} - 1}. \qquad (55)$$

Combining with the fact, i.e.,

$$R^{\rightarrow}(\boldsymbol{y}, \boldsymbol{y}[\boldsymbol{y}_i : K \rightarrow k]) = 1$$

from Eq. (7), Eq. (52) can be upper bounded as

$$R_t^{\leftarrow}(\boldsymbol{y}) = \sum_{i, \boldsymbol{y}_i = K} \left[ \sum_{k=1}^{K-1} \frac{q_t^{\leftarrow}(\boldsymbol{y}[\boldsymbol{y}_i : K \rightarrow k])}{q_t^{\leftarrow}(\boldsymbol{y})} \right] \leq \frac{\text{numK}(\boldsymbol{y}) \cdot K}{e^{(T-t)} - 1}.$$

Hence, the proof is completed. $\qquad \square$

**Remark 1.** *Here, an interesting property is that compared with the upper bound of $\beta_t(\boldsymbol{y})$ in the* uniform forward process *Chen & Ying (2024), i.e.,*

$$\sum_{\boldsymbol{y}' \neq \boldsymbol{y}} R_t^{\leftarrow}(\boldsymbol{y}', \boldsymbol{y}) \leq K \cdot d \cdot \frac{1 + e^{-2(T-t)}}{1 - e^{-2(T-t)}} \leq K \cdot d \cdot (1 + (T-t)^{-1}).$$

*the upper bound of $\beta_t(\boldsymbol{y})$ in* absorbing forward process *will only be*

$$\sum_{\boldsymbol{y}' \neq \boldsymbol{y}} R_t^{\leftarrow}(\boldsymbol{y}', \boldsymbol{y}) \leq K \cdot \text{numK}(\boldsymbol{y}) \cdot \frac{e^{-(T-t)}}{1 - e^{-(T-t)}}.$$

*The latter upper bound is strictly better compared with the former one, since the number of mask tokens, i.e., $\text{numK}(\boldsymbol{y}) \leq d$. Besides, with the time growth (from $0$ to $T$), $\text{numK}(\boldsymbol{y})$ will be monotonic decrease for $R_t^{\leftarrow}(\boldsymbol{y})$ (from $d$ to $0$). Since the dominating term in the complexity analysis of truncated uniformization is $\beta_t$, the discrete diffusion models with absorbing forward process are expected to have a better result. The mechanism of the acceleration can be explained in one sentence, i.e.,*

---

At each uniformization step, absorbing the discrete diffusion model knows the token needs (masked token)/ or does not need (unmasked token) to denoise, and an unmasked token will not be denoised twice.

---

*Rigorously, this property can be summarized by Lemma 10.*

**Lemma 10.** *Suppose Assumption [A2] hold, and $0 < t_0 \leq t$, we have $q_{t|t_0}^{\leftarrow}(\boldsymbol{y}|\boldsymbol{y}_0) \neq 0$ if and only if*

$$\boldsymbol{y} \in \mathcal{Y}^{\leftarrow}(\boldsymbol{y}_0) = \{\boldsymbol{y}'|\forall i, \quad \boldsymbol{y}_{0,i} = K \text{ or } \boldsymbol{y}_i' = \boldsymbol{y}_{0,i}\}.$$

*Proof.* According to the Bayes' theorem, for any $t \geq t_0$, it has

$$\begin{aligned} q_{t,t_0}^{\leftarrow}(\boldsymbol{y}, \boldsymbol{y}_0) &= q_{t|t_0}^{\leftarrow}(\boldsymbol{y}|\boldsymbol{y}_0) \cdot q_{t_0}^{\leftarrow}(\boldsymbol{y}_0) = q_{T-t,T-t_0}^{\rightarrow}(\boldsymbol{y}, \boldsymbol{y}_0) \\ &= q_{T-t_0,T-t}^{\rightarrow}(\boldsymbol{y}_0, \boldsymbol{y}) = q_{T-t_0|T-t}^{\rightarrow}(\boldsymbol{y}_0|\boldsymbol{y}) \cdot q_{T-t}^{\rightarrow}(\boldsymbol{y}), \end{aligned} \qquad (56)$$

where the third equation follows from the reversibility of the absorbing forward process shown in Campbell et al. (2022). Following from the forward transition kernel shown in Lemma 8, we know that

$$q_{T-t_0|T-t}^{\rightarrow}(\boldsymbol{y}_0|\boldsymbol{y}) \neq 0 \quad \Leftrightarrow \quad \boldsymbol{y}_0 \in \mathcal{Y}^{\rightarrow}(\boldsymbol{y}) = \{\boldsymbol{y}'|\forall i, \quad \boldsymbol{y}_i' = \boldsymbol{y}_i \text{ or } \boldsymbol{y}_i' = K\}. \qquad (57)$$

Combining Assumption **[A2]** and Lemma 8, we have $q_\tau^{\rightarrow}(\boldsymbol{y}) > 0$ for all $\boldsymbol{y} \in \mathcal{Y}$, which implies

$$q_{t_0}^{\leftarrow}(\boldsymbol{y}_0) = q_{T-t_0}^{\rightarrow}(\boldsymbol{y}_0) > 0 \quad \text{and} \quad q_t^{\rightarrow}(\boldsymbol{y}) > 0. \qquad (58)$$

Then, we can summarize

$$q_{t|t_0}^{\leftarrow}(\boldsymbol{y}|\boldsymbol{y}_0) \neq 0 \quad \Leftrightarrow \quad \boldsymbol{y} \in \mathcal{Y}^{\leftarrow}(\boldsymbol{y}_0) = \{\boldsymbol{y}'|\forall i, \quad \boldsymbol{y}_{0,i} = K \text{ or } \boldsymbol{y}_i' = \boldsymbol{y}_{0,i}\}.$$

Hence, the proof is completed. $\qquad \square$

### D.2 THE CONVERGENCE OF ALG. 1

Suppose, with the infinitesimal reverse transition rate, the particles in Alg. 1 during the reverse process are denotes as random variables $\{\hat{\mathbf{y}}_t\}_{t=0}^{T-\delta}$, whose underlying distributions are $\hat{q}_t$. Then, the implementation will be equivalent to the following Poisson process. For $t \in (t_{w-1}, t_w], \hat{\mathbf{y}}_{t_{w-1}} = \boldsymbol{y}_0$ and $\hat{\mathbf{y}}_t = \boldsymbol{y}$,

1. With probability $\Delta t \cdot \beta_{t_w}(\boldsymbol{y}_0)$, allow a state transition.

2. Conditioning on an allowed transition, move from $\boldsymbol{y}$ to $\boldsymbol{y}'$ with probability

$$\hat{M}_{t|t_{w-1}}(\boldsymbol{y}'|\boldsymbol{y}, \boldsymbol{y}_0) = \begin{cases} \beta_{t_w}^{-1}(\boldsymbol{y}_0) \cdot \hat{R}_{t,\boldsymbol{y}_0}(\boldsymbol{y}', \boldsymbol{y}) & \boldsymbol{y}' \neq \boldsymbol{y} \\ 1 - \beta_{t_w}^{-1}(\boldsymbol{y}_0)\hat{R}_{t,\boldsymbol{y}_0}(\boldsymbol{y}) & \text{otherwise} \end{cases}.$$

Here we should note that

$$\hat{R}_{t,\boldsymbol{y}_0}(\boldsymbol{y}) \leq \beta_t(\boldsymbol{y}) = K \cdot \text{numK}(\boldsymbol{y}) \cdot \frac{1}{e^{T-t}-1} \leq K \cdot \text{numK}(\boldsymbol{y}_0) \cdot \frac{1}{e^{T-t_w}-1} = \beta_{t_w}(\boldsymbol{y}_0),$$

where the second inequality established since $\text{numK}(\hat{\mathbf{y}}_t) \leq \text{numK}(\hat{\mathbf{y}}_{t_{w-1}})$ and $(e^{T-t}-1)^{-1}$ is monotonic increasing. Under these two steps, the practical conditional probability satisfies

$$\hat{q}_{t+\Delta t|t,t_{w-1}}(\boldsymbol{y}'|\boldsymbol{y}, \boldsymbol{y}_0) = \begin{cases} \Delta t \cdot \beta_{t_w}(\boldsymbol{y}_0) \cdot \hat{R}_{t,\boldsymbol{y}_0}(\boldsymbol{y}', \boldsymbol{y}) \cdot \beta_{t_w}^{-1}(\boldsymbol{y}_0) & \boldsymbol{y}' \neq \boldsymbol{y} \\ 1 - \Delta t \cdot \beta_{t_w}(\boldsymbol{y}_0) + \Delta t \cdot \beta_{t_w}(\boldsymbol{y}_0) \cdot (1 - \beta_{t_w}(\boldsymbol{y}_0)^{-1} \cdot \hat{R}_{t,\boldsymbol{y}_0}(\boldsymbol{y})) & \boldsymbol{y}' = \boldsymbol{y} \end{cases},$$

$$= \begin{cases} \Delta t \cdot \hat{R}_{t,\boldsymbol{y}_0}(\boldsymbol{y}', \boldsymbol{y}) & \boldsymbol{y}' \neq \boldsymbol{y} \\ 1 - \Delta t \cdot \hat{R}_{t,\boldsymbol{y}_0}(\boldsymbol{y}) & \boldsymbol{y}' = \boldsymbol{y} \end{cases}.$$

$$(59)$$

**Lemma 11.** *Following the notations shown in Section A, we have*

$$\lim_{\Delta t \to 0} \left[ \Delta t^{-1} \cdot \ln \frac{1 - \sum_{\boldsymbol{y}' \neq \boldsymbol{y}} q_{t+\Delta t|t,t_{w-1}}^{\leftarrow}(\boldsymbol{y}'|\boldsymbol{y}, \boldsymbol{y}_0)}{1 - \sum_{\boldsymbol{y}' \neq \boldsymbol{y}} \hat{q}_{t+\Delta t|t,t_{w-1}}(\boldsymbol{y}'|\boldsymbol{y}, \boldsymbol{y}_0)} \right] = \hat{R}_{t,\boldsymbol{y}_0}(\boldsymbol{y}) - R_t^{\leftarrow}(\boldsymbol{y}).$$

*Proof.* Since we have required $\Delta t \to 0$, for any $\boldsymbol{y}' \neq \boldsymbol{y}$, it has

$$\hat{q}_{t+\Delta t|t,t_{w-1}}(\boldsymbol{y}'|\boldsymbol{y}, \boldsymbol{y}_0) \to \hat{q}_{t|t}(\boldsymbol{y}'|\boldsymbol{y}, \boldsymbol{y}_0) = 0$$
$$\text{and} \quad q_{t+\Delta t|t,t_{w-1}}^{\leftarrow}(\boldsymbol{y}'|\boldsymbol{y}, \boldsymbol{y}_0) = q_{t+\Delta t|t}^{\leftarrow}(\boldsymbol{y}'|\boldsymbol{y}) \to q_{t|t}^{\leftarrow}(\boldsymbol{y}'|\boldsymbol{y}) = 0,$$

where the first row follows from Eq. (59) and the second row follows from Lemma. 1. This automatically makes

$$\left| \frac{\sum_{\boldsymbol{y}' \neq \boldsymbol{y}} \left( \hat{q}_{t+\Delta t|t,t_{w-1}}(\boldsymbol{y}'|\boldsymbol{y}, \boldsymbol{y}_0) - q_{t+\Delta t|t,t_{w-1}}^{\leftarrow}(\boldsymbol{y}'|\boldsymbol{y}, \boldsymbol{y}_0) \right)}{1 - \sum_{\boldsymbol{y}' \neq \boldsymbol{y}} \hat{q}_{t+\Delta t|t,t_{w-1}}(\boldsymbol{y}'|\boldsymbol{y}, \boldsymbol{y}_0)} \right| \leq \frac{1}{2} < 1.$$

Under this condition, we have

$$\ln \frac{1 - \sum_{\boldsymbol{y}' \neq \boldsymbol{y}} q_{t+\Delta t|t,t_{w-1}}^{\leftarrow}(\boldsymbol{y}'|\boldsymbol{y}, \boldsymbol{y}_0)}{1 - \sum_{\boldsymbol{y}' \neq \boldsymbol{y}} \hat{q}_{t+\Delta t|t,t_{w-1}}(\boldsymbol{y}'|\boldsymbol{y}, \boldsymbol{y}_0)} = \ln \left[ 1 + \frac{\sum_{\boldsymbol{y}' \neq \boldsymbol{y}} \left( \hat{q}_{t+\Delta t|t,t_{w-1}}(\boldsymbol{y}'|\boldsymbol{y}, \boldsymbol{y}_0) - q_{t+\Delta t|t}^{\leftarrow}(\boldsymbol{y}'|\boldsymbol{y}) \right)}{1 - \sum_{\boldsymbol{y}' \neq \boldsymbol{y}} \hat{q}_{t+\Delta t|t,t_{w-1}}(\boldsymbol{y}'|\boldsymbol{y}, \boldsymbol{y}_0)} \right]$$

$$= \sum_{i=1}^{\infty} \frac{(-1)^{i+1}}{i} \cdot \left[ \frac{\sum_{\boldsymbol{y}' \neq \boldsymbol{y}} \left( \hat{q}_{t+\Delta t|t,t_{w-1}}(\boldsymbol{y}'|\boldsymbol{y}, \boldsymbol{y}_0) - q_{t+\Delta t|t,t_{w-1}}^{\leftarrow}(\boldsymbol{y}'|\boldsymbol{y}, \boldsymbol{y}_0) \right)}{1 - \sum_{\boldsymbol{y}' \neq \boldsymbol{y}} \hat{q}_{t+\Delta t|t,t_{w-1}}(\boldsymbol{y}'|\boldsymbol{y}, \boldsymbol{y}_0)} \right]^i,$$

which implies (with the dominated convergence theorem)

$$\lim_{\Delta t \to 0} \left[ \Delta t^{-1} \cdot \ln \frac{1 - \sum_{\boldsymbol{y}' \neq \boldsymbol{y}} q^{\leftarrow}_{t+\Delta t|t,t_{w-1}}(\boldsymbol{y}'|\boldsymbol{y}, \boldsymbol{y}_0)}{1 - \sum_{\boldsymbol{y}' \neq \boldsymbol{y}} \hat{q}_{t+\Delta t|t,t_{w-1}}(\boldsymbol{y}'|\boldsymbol{y}, \boldsymbol{y}_0)} \right]$$

$$= \sum_{i=1}^{\infty} \frac{(-1)^{i+1}}{i} \cdot \lim_{\Delta t \to 0} \frac{\sum_{\boldsymbol{y}' \neq \boldsymbol{y}} \left( \hat{q}_{t+\Delta t|t,t_{w-1}}(\boldsymbol{y}'|\boldsymbol{y}, \boldsymbol{y}_0) - q^{\leftarrow}_{t+\Delta t|t,t_{w-1}}(\boldsymbol{y}'|\boldsymbol{y}, \boldsymbol{y}_0) \right)}{\Delta t}$$

$$\cdot \lim_{\Delta t \to 0} \frac{\left( \sum_{\boldsymbol{y}' \neq \boldsymbol{y}} \left( \hat{q}_{t+\Delta t|t,t_{w-1}}(\boldsymbol{y}'|\boldsymbol{y}, \boldsymbol{y}_0) - q^{\leftarrow}_{t+\Delta t|t,t_{w-1}}(\boldsymbol{y}'|\boldsymbol{y}, \boldsymbol{y}_0) \right) \right)^{i-1}}{\left( 1 - \sum_{\boldsymbol{y}' \neq \boldsymbol{y}} \hat{q}_{t+\Delta t|t,t_{w-1}}(\boldsymbol{y}'|\boldsymbol{y}, \boldsymbol{y}_0) \right)^i}.$$

Only when $i = 1$, we have

$$\lim_{\Delta t \to 0} \frac{\left( \sum_{\boldsymbol{y}' \neq \boldsymbol{y}} \left( \hat{q}_{t+\Delta t|t,t_{w-1}}(\boldsymbol{y}'|\boldsymbol{y}, \boldsymbol{y}_0) - q^{\leftarrow}_{t+\Delta t|t,t_{w-1}}(\boldsymbol{y}'|\boldsymbol{y}, \boldsymbol{y}_0) \right) \right)^{i-1}}{\left( 1 - \sum_{\boldsymbol{y}' \neq \boldsymbol{y}} \hat{q}_{t+\Delta t|t}(\boldsymbol{y}'|\boldsymbol{y}) \right)^i} = 1,$$

otherwise it will be equivalent to 0. Therefore, we have

$$\lim_{\Delta t \to 0} \left[ \Delta t^{-1} \cdot \ln \frac{1 - \sum_{\boldsymbol{y}' \neq \boldsymbol{y}} q^{\leftarrow}_{t+\Delta t|t,t_{w-1}}(\boldsymbol{y}'|\boldsymbol{y}, \boldsymbol{y}_0)}{1 - \sum_{\boldsymbol{y}' \neq \boldsymbol{y}} \hat{q}_{t+\Delta t|t,t_{w-1}}(\boldsymbol{y}'|\boldsymbol{y}, \boldsymbol{y}_0)} \right]$$

$$= \lim_{\Delta t \to 0} \frac{\sum_{\boldsymbol{y}' \neq \boldsymbol{y}} \left( \hat{q}_{t+\Delta t|t,t_{w-1}}(\boldsymbol{y}'|\boldsymbol{y}, \boldsymbol{y}_0) - q^{\leftarrow}_{t+\Delta t|t,t_{w-1}}(\boldsymbol{y}'|\boldsymbol{y}, \boldsymbol{y}_0) \right)}{\Delta t}$$

$$= \sum_{\boldsymbol{y}' \neq \boldsymbol{y}} \left( \hat{R}_{t,\boldsymbol{y}_0}(\boldsymbol{y}', \boldsymbol{y}) - R^{\leftarrow}_t(\boldsymbol{y}', \boldsymbol{y}) \right) = \hat{R}_{t,\boldsymbol{y}_0}(\boldsymbol{y}) - R^{\leftarrow}_t(\boldsymbol{y}),$$

where the second equation follows from Eq. (59) and the second row follows from Lemma. 1. Hence, the proof is completed. $\square$

**Theorem 3** (The convergence of Alg. 1). *Suppose Assumption [A1] and [A2] hold, if Alg. 1 has*

$$t_0 = 0, \quad t_W = T - \delta, \quad and \quad \epsilon_{score} \leq T^{-1/2} \cdot \epsilon \quad where \quad T = \ln(4d/\epsilon^2) \quad and \quad \delta \leq d^{-1}\epsilon,$$

*the TV distance between the target discrete distribution $q_*$ and the underlying distribution of the output particle $\hat{q}_{T-\delta}$ will satisfy* $\mathrm{TV}\left(q_*, \hat{q}_{T-\delta}\right) \leq 2\epsilon$.

*Proof.* Here we provide the upper bound of TV distance accumulation in a specific segment, e.g., from $t_{w-1}$ to $t_w$. According to the chain rule of KL divergence, i.e., Lemma 15, we have

$$\mathrm{KL}\left(q^{\leftarrow}_{t_w} \| \hat{q}_{t_w}\right) \leq \mathrm{KL}\left(q^{\leftarrow}_{t_{w-1}} \| \hat{q}_{t_{w-1}}\right) + \mathbb{E}_{\mathbf{y}_0 \sim q^{\leftarrow}_{t_{w-1}}} \left[ \mathrm{KL}\left(q^{\leftarrow}_{t_w|t_{w-1}}(\cdot|\mathbf{y}_0) \| \hat{q}_{t_w|t_{w-1}}(\cdot|\mathbf{y}_0)\right) \right]$$

$$= \mathrm{KL}\left(q^{\leftarrow}_{t_{w-1}} \| \hat{q}_{t_{w-1}}\right) + \int_{t_{w-1}}^{t_w} \mathrm{d}\mathbb{E}_{\mathbf{y}_0 \sim q^{\leftarrow}_{t_{w-1}}} \left[ \mathrm{KL}\left(q^{\leftarrow}_{t|t_{w-1}}(\cdot|\mathbf{y}_0) \| \hat{q}_{t|t_{w-1}}(\cdot|\mathbf{y}_0)\right) \right] \quad (60)$$

Then, it has

$$\mathrm{d}\mathbb{E}_{\mathbf{y}_0 \sim q^{\leftarrow}_{t_{w-1}}} \left[ \mathrm{KL}\left(q^{\leftarrow}_{t|t_{w-1}}(\cdot|\mathbf{y}_0^{\leftarrow}) \| \hat{q}_{t|t_{w-1}}(\cdot|\mathbf{y}_0^{\leftarrow})\right) \right] / \mathrm{d}t$$

$$= \lim_{\Delta \to 0} (\Delta t)^{-1} \cdot \mathbb{E}_{\mathbf{y}_0 \sim q^{\leftarrow}_{t_{w-1}}} \left[ \mathrm{KL}\left(q^{\leftarrow}_{t+\Delta t|t_{w-1}}(\cdot|\mathbf{y}_0) \| \hat{q}_{t+\Delta t|t_{w-1}}(\cdot|\mathbf{y}_0)\right) - \mathrm{KL}\left(q^{\leftarrow}_{t|t_{w-1}}(\cdot|\mathbf{y}_0) \| \hat{q}_{t|t_{w-1}}(\cdot|\mathbf{y}_0)\right) \right]$$

$$\leq \lim_{\Delta \to 0} (\Delta t)^{-1} \cdot \mathbb{E}_{\mathbf{y}_0 \sim q^{\leftarrow}_{t_{w-1}}} \left[ \mathbb{E}_{\mathbf{y} \sim q^{\leftarrow}_{t|t_{w-1}}(\cdot|\mathbf{y}_0)} \left( \mathrm{KL}\left(q^{\leftarrow}_{t+\Delta t|t,t_{w-1}}(\cdot|\mathbf{y}, \mathbf{y}_0) \| \hat{q}_{t+\Delta t|t,t_{w-1}}(\cdot|\mathbf{y}, \mathbf{y}_0)\right) \right) \right]$$

where the inequality follows from the chain rule of the KL divergence, i.e., Lemma 15. Then, it has

$$\mathrm{d}\mathbb{E}_{\mathbf{y}_0 \sim q^{\leftarrow}_{t_{w-1}}} \left[ \mathrm{KL}\left(q^{\leftarrow}_{t|t_{w-1}}(\cdot|\mathbf{y}_0) \| \hat{q}_{t|t_{w-1}}(\cdot|\mathbf{y}_0)\right) \right] / \mathrm{d}t$$

$$\leq \sum_{\boldsymbol{y} \in \mathcal{Y}, \boldsymbol{y}_0 \in \mathcal{Y}^{\to}(\boldsymbol{y})} q^{\leftarrow}_{t,t_{w-1}}(\boldsymbol{y}, \boldsymbol{y}_0) \cdot \underbrace{\lim_{\Delta t \to 0} \left[ \frac{\mathrm{KL}\left(q^{\leftarrow}_{t+\Delta t|t,t_{w-1}}(\cdot|\boldsymbol{y}, \boldsymbol{y}_0) \| \hat{q}_{t+\Delta t|t,t_{w-1}}(\cdot|\boldsymbol{y}, \boldsymbol{y}_0)\right)}{\Delta t} \right]}_{\text{Term 1}}$$

$$(61)$$

where the inequality and the notation $\mathcal{Y}^{\rightarrow}(\cdot)$ follows from Lemma 10. For each $\boldsymbol{y}^{\leftarrow} \in \mathcal{Y}, \boldsymbol{y}_0^{\leftarrow} \in \mathcal{Y}^{\rightarrow}(\boldsymbol{y})$, we focus on Term 1 of Eq. (61), and have

$$
\text{Term 1} = \lim_{\Delta t \to 0} \left[ \Delta t^{-1} \cdot \sum_{\boldsymbol{y}' \in \mathcal{Y}} q_{t+\Delta t|t,t_{w-1}}^{\leftarrow}(\boldsymbol{y}'|\boldsymbol{y}, \boldsymbol{y}_0) \cdot \ln \frac{q_{t+\Delta t|t,t_{w-1}}^{\leftarrow}(\boldsymbol{y}'|\boldsymbol{y}, \boldsymbol{y}_0)}{\hat{q}_{t+\Delta t|t,t_{w-1}}(\boldsymbol{y}'|\boldsymbol{y}, \boldsymbol{y}_0)} \right]
$$

$$
= \underbrace{\lim_{\Delta t \to 0} \left[ \sum_{\boldsymbol{y}' \neq \boldsymbol{y}} \frac{q_{t+\Delta t|t,t_{w-1}}^{\leftarrow}(\boldsymbol{y}'|\boldsymbol{y}, \boldsymbol{y}_0)}{\Delta t} \cdot \ln \frac{q_{t+\Delta t|t,t_{w-1}}^{\leftarrow}(\boldsymbol{y}'|\boldsymbol{y}, \boldsymbol{y}_0)}{\hat{q}_{t+\Delta t|t,t_{w-1}}(\boldsymbol{y}'|\boldsymbol{y}, \boldsymbol{y}_0)} \right]}_{\text{Term 1.1}} +
$$

$$
\underbrace{\lim_{\Delta t \to 0} \left[ \Delta t^{-1} \cdot \left( 1 - \sum_{\boldsymbol{y}' \neq \boldsymbol{y}} q_{t+\Delta t|t,t_{w-1}}^{\leftarrow}(\boldsymbol{y}'|\boldsymbol{y}, \boldsymbol{y}_0) \right) \cdot \ln \frac{1 - \sum_{\boldsymbol{y}' \neq \boldsymbol{y}} q_{t+\Delta t|t,t_{w-1}}^{\leftarrow}(\boldsymbol{y}'|\boldsymbol{y}, \boldsymbol{y}_0)}{1 - \sum_{\boldsymbol{y}' \neq \boldsymbol{y}} \hat{q}_{t+\Delta t|t,t_{w-1}}(\boldsymbol{y}'|\boldsymbol{y}, \boldsymbol{y}_0)} \right]}_{\text{Term 1.2}}.
$$

$$(62)$$

For Term 1.1, we have

$$
\text{Term 1.1} = \sum_{\boldsymbol{y}' \neq \boldsymbol{y}} \lim_{\Delta t \to 0} \left[ \frac{q_{t+\Delta t|t,t_{w-1}}^{\leftarrow}(\boldsymbol{y}'|\boldsymbol{y}, \boldsymbol{y}_0)}{\Delta t} \right] \cdot \lim_{\Delta t \to 0} \left[ \ln \frac{q_{t+\Delta t|t,t_{w-1}}^{\leftarrow}(\boldsymbol{y}'|\boldsymbol{y}, \boldsymbol{y}_0)}{\hat{q}_{t+\Delta t|t,t_{w-1}}(\boldsymbol{y}'|\boldsymbol{y}, \boldsymbol{y}_0)} \right]
$$

$$
= \sum_{\boldsymbol{y}' \neq \boldsymbol{y}} R_t^{\leftarrow}(\boldsymbol{y}', \boldsymbol{y}) \cdot \ln \left[ \lim_{\Delta t \to 0} \left( \frac{q_{t+\Delta t|t,t_{w-1}}^{\leftarrow}(\boldsymbol{y}'|\boldsymbol{y}, \boldsymbol{y}_0)}{\Delta t} \cdot \frac{\Delta t}{\hat{q}_{t+\Delta t|t,t_{w-1}}(\boldsymbol{y}'|\boldsymbol{y}, \boldsymbol{y}_0)} \right) \right]
$$

$$
= \sum_{\boldsymbol{y}' \neq \boldsymbol{y}} R_t^{\leftarrow}(\boldsymbol{y}', \boldsymbol{y}) \cdot \ln \frac{R_t^{\leftarrow}(\boldsymbol{y}', \boldsymbol{y})}{\hat{R}_{t,\boldsymbol{y}_0}(\boldsymbol{y}', \boldsymbol{y})},
$$

$$(63)$$

where the last equation follows from Lemma 1 and Eq. (59). For Term 1.2, we have

$$
\text{Term 1.2} = \lim_{\Delta t \to 0} \left[ 1 - \sum_{\boldsymbol{y}' \neq \boldsymbol{y}} q_{t+\Delta t|t,t_{w-1}}^{\leftarrow}(\boldsymbol{y}'|\boldsymbol{y}, \boldsymbol{y}_0) \right]
$$

$$
\cdot \lim_{\Delta t \to 0} \left[ \Delta t^{-1} \cdot \ln \frac{1 - \sum_{\boldsymbol{y}' \neq \boldsymbol{y}} q_{t+\Delta t|t,t_{w-1}}^{\leftarrow}(\boldsymbol{y}'|\boldsymbol{y}, \boldsymbol{y}_0)}{1 - \sum_{\boldsymbol{y}' \neq \boldsymbol{y}} \hat{q}_{t+\Delta t|t,t_{w-1}}(\boldsymbol{y}'|\boldsymbol{y}, \boldsymbol{y}_0)} \right] \leq 1 \cdot (\hat{R}_{t,\boldsymbol{y}_0}(\boldsymbol{y}) - R_t^{\leftarrow}(\boldsymbol{y}))
$$

$$(64)$$

where the first inequality follows from Lemma 11. Plugging Eq. (63), Eq. (64) and Eq. (62), into Eq. (61) we have

$$
d\mathbb{E}_{\mathbf{y}_0 \sim q_{t_{w-1}}^{\leftarrow}} \left[ \text{KL} \left( q_{t|t_{w-1}}^{\leftarrow}(\cdot|\mathbf{y}_0) \big\| \hat{q}_{t|t_{w-1}}(\cdot|\mathbf{y}_0) \right) \right] / dt
$$

$$
\leq \sum_{\boldsymbol{y} \in \mathcal{Y}, \boldsymbol{y}_0 \in \mathcal{Y}^{\rightarrow}(\boldsymbol{y})} q_{t,t_{w-1}}^{\leftarrow}(\boldsymbol{y}, \boldsymbol{y}_0) \cdot \left( \sum_{\boldsymbol{y}' \neq \boldsymbol{y}} R_t^{\leftarrow}(\boldsymbol{y}', \boldsymbol{y}) \cdot \ln \frac{R_t^{\leftarrow}(\boldsymbol{y}', \boldsymbol{y})}{\hat{R}_{t,\boldsymbol{y}_0}(\boldsymbol{y}', \boldsymbol{y})} + \hat{R}_{t,\boldsymbol{y}_0}(\boldsymbol{y}) - R_t^{\leftarrow}(\boldsymbol{y}) \right).
$$

$$(65)$$

Then, for any $\boldsymbol{y} \in \mathcal{Y}$ and $\boldsymbol{y}_0 \in \mathcal{Y}^{\rightarrow}(\boldsymbol{y})$, we have

$$
\sum_{\boldsymbol{y}' \neq \boldsymbol{y}} R_t^{\leftarrow}(\boldsymbol{y}', \boldsymbol{y}) \cdot \ln \frac{R_t^{\leftarrow}(\boldsymbol{y}', \boldsymbol{y})}{\hat{R}_{t,\boldsymbol{y}_0}(\boldsymbol{y}', \boldsymbol{y})} + \hat{R}_{t,\boldsymbol{y}_0}(\boldsymbol{y}) - R_t^{\leftarrow}(\boldsymbol{y})
$$

$$
= \sum_{\boldsymbol{y}' \neq \boldsymbol{y}} R_t^{\leftarrow}(\boldsymbol{y}', \boldsymbol{y}) \ln \frac{R_t^{\leftarrow}(\boldsymbol{y}', \boldsymbol{y})}{\tilde{R}_t(\boldsymbol{y}', \boldsymbol{y})} + \tilde{R}_t(\boldsymbol{y}) - R_t^{\leftarrow}(\boldsymbol{y})
$$

$$(66)$$

$$
\underbrace{+ \sum_{\boldsymbol{y}' \neq \boldsymbol{y}} R_t^{\leftarrow}(\boldsymbol{y}', \boldsymbol{y}) \ln \frac{\tilde{R}_t(\boldsymbol{y}', \boldsymbol{y})}{\hat{R}_{t,\boldsymbol{y}_0}(\boldsymbol{y}', \boldsymbol{y})} + \hat{R}_{t,\boldsymbol{y}_0}(\boldsymbol{y}) - \tilde{R}_t(\boldsymbol{y})}_{\text{Term 2}}.
$$

When $\tilde{R}_t(\boldsymbol{y}) \leq \beta_{t_w}(\boldsymbol{y}_0)$, due to Eq. (18), we have

$$\hat{R}_{t,\boldsymbol{y}_0}(\boldsymbol{y}',\boldsymbol{y}) = \tilde{R}_t(\boldsymbol{y}',\boldsymbol{y}) \quad \text{and} \quad \hat{R}_{t,\boldsymbol{y}_0}(\boldsymbol{y}) = \sum_{\boldsymbol{y}' \neq \boldsymbol{y}} \hat{R}_{t,\boldsymbol{y}_0}(\boldsymbol{y}',\boldsymbol{y}) = \sum_{\boldsymbol{y}' \neq \boldsymbol{y}} \tilde{R}_t(\boldsymbol{y}',\boldsymbol{y}) = \tilde{R}_t(\boldsymbol{y})$$

which implies Term $2 = 0$ in Eq. (66). Otherwise, we have

$$\frac{\hat{R}_{t,\boldsymbol{y}_0}(\boldsymbol{y}',\boldsymbol{y})}{\tilde{R}_t(\boldsymbol{y}',\boldsymbol{y})} = \frac{\beta_{t_w}(\boldsymbol{y}_0)}{\tilde{R}_t(\boldsymbol{y})} \quad \text{and} \quad \frac{\hat{R}_{t,\boldsymbol{y}_0}(\boldsymbol{y})}{\tilde{R}_t(\boldsymbol{y})} = \frac{\beta_{t_w}(\boldsymbol{y}_0)}{\tilde{R}_t(\boldsymbol{y})},$$

which implies

$$\text{Term } 2 = \sum_{\boldsymbol{y}' \neq \boldsymbol{y}} R_t^\leftarrow(\boldsymbol{y}',\boldsymbol{y}) \cdot \ln \frac{\tilde{R}_t(\boldsymbol{y})}{\beta_{t_w}(\boldsymbol{y}_0)} + \beta_{t_w}(\boldsymbol{y}_0) - \tilde{R}_t(\boldsymbol{y})$$

$$= R_t^\leftarrow(\boldsymbol{y}) \cdot \ln \left[ 1 + \frac{\tilde{R}_t(\boldsymbol{y}) - \beta_{t_w}(\boldsymbol{y}_0)}{\beta_{t_w}(\boldsymbol{y}_0)} \right] + \beta_{t_w}(\boldsymbol{y}_0) - \tilde{R}_t(\boldsymbol{y})$$

$$\leq \beta_{t_w}(\boldsymbol{y}_0) \cdot \left[ \frac{\tilde{R}_t(\boldsymbol{y}) - \beta_{t_w}(\boldsymbol{y}_0)}{\beta_{t_w}(\boldsymbol{y}_0)} \right] + \beta_{t_w}(\boldsymbol{y}_0) - \tilde{R}_t(\boldsymbol{y}) = 0,$$

where the last inequality follows from

$$\boldsymbol{y}_0 \in \mathcal{Y}^\rightarrow(\boldsymbol{y}) \quad \Rightarrow \quad \text{numK}(\boldsymbol{y}) \leq \text{numK}(\boldsymbol{y}_0) \quad \Rightarrow \quad R_t^\leftarrow(\boldsymbol{y}) \leq \beta_{t_w}(\boldsymbol{y}_0).$$

Combining with Eq. (66) and Eq. (65), we have

$$d\mathbb{E}_{\mathbf{y}_0 \sim q_{t_{w-1}}^\leftarrow} \left[ \text{KL}\left( q_{t|t_{w-1}}^\leftarrow(\cdot|\mathbf{y}_0) \big\| \hat{q}_{t|t_{w-1}}(\cdot|\mathbf{y}_0) \right) \right] / dt$$

$$\leq \sum_{\boldsymbol{y} \in \mathcal{Y}} q_t^\leftarrow(\boldsymbol{y}) \cdot \left( \sum_{\boldsymbol{y}' \neq \boldsymbol{y}} R_t^\leftarrow(\boldsymbol{y}',\boldsymbol{y}) \cdot \ln \frac{R_t^\leftarrow(\boldsymbol{y}',\boldsymbol{y})}{\tilde{R}_t(\boldsymbol{y}',\boldsymbol{y})} + \tilde{R}_t(\boldsymbol{y}) - R_t^\leftarrow(\boldsymbol{y}) \right)$$

$$= \sum_{\boldsymbol{y} \in \mathcal{Y}} q_t^\leftarrow(\boldsymbol{y}) \cdot \left( \sum_{\boldsymbol{y}' \neq \boldsymbol{y}} R_t^\leftarrow(\boldsymbol{y}',\boldsymbol{y}) \cdot \ln \frac{R_t^\leftarrow(\boldsymbol{y}',\boldsymbol{y})}{\tilde{R}_t(\boldsymbol{y}',\boldsymbol{y})} + \sum_{\boldsymbol{y}' \neq \boldsymbol{y}} \tilde{R}_t(\boldsymbol{y}',\boldsymbol{y}) - \sum_{\boldsymbol{y}' \neq \boldsymbol{y}} R_t^\leftarrow(\boldsymbol{y}',\boldsymbol{y}) \right) \quad (67)$$

$$= \sum_{\boldsymbol{y} \in \mathcal{Y}} q_t^\leftarrow(\boldsymbol{y}) \cdot \sum_{\boldsymbol{y}' \neq \boldsymbol{y}} R^\rightarrow(\boldsymbol{y},\boldsymbol{y}') \cdot \left[ -\frac{q_t^\leftarrow(\boldsymbol{y}')}{q_t^\leftarrow(\boldsymbol{y})} + \tilde{v}_{t,\boldsymbol{y}}(\boldsymbol{y}') + \frac{q_t^\leftarrow(\boldsymbol{y}')}{q_t^\leftarrow(\boldsymbol{y})} \ln \frac{q_t^\leftarrow(\boldsymbol{y}')}{q_t^\leftarrow(\boldsymbol{y})\tilde{v}_{t,\boldsymbol{y}}(\boldsymbol{y}')} \right]$$

$$= \sum_{\boldsymbol{y} \in \mathcal{Y}} q_t^\leftarrow(\boldsymbol{y}) \cdot \sum_{\boldsymbol{y}' \neq \boldsymbol{y}} R^\rightarrow(\boldsymbol{y},\boldsymbol{y}') D_\phi \left( \frac{q_t^\leftarrow(\boldsymbol{y}')}{q_t^\leftarrow(\boldsymbol{y})} \big\| \tilde{v}_{t,\boldsymbol{y}}(\boldsymbol{y}') \right),$$

where $D_\phi$ is the Bregman divergence with $\phi(c) = c \ln c$ (as Eq. (6)), and the last equation follows from the definition of Bregman divergence:

$$D_\phi(u\|v) = \phi(u) - \phi(v) - \langle \nabla\phi(v), u - v \rangle = u \ln \frac{u}{v} - u + v.$$

Therefore, Eq. (60) can be rewritten as

$$\text{KL}\left( q_{t_w}^\leftarrow \big\| \hat{q}_{t_w} \right) \leq \text{KL}\left( q_{t_{w-1}}^\leftarrow \big\| \hat{q}_{t_{w-1}} \right) + \int_{t_{w-1}}^{t_w} \mathbb{E}_{\mathbf{y} \sim q_{T-t}^\rightarrow} \left[ \sum_{\boldsymbol{y}' \neq \boldsymbol{y}} R^\rightarrow(\mathbf{y},\boldsymbol{y}') \cdot D_\phi\left( v_{t,\mathbf{y}}(\boldsymbol{y}') \big\| \tilde{v}_{t,\mathbf{y}}(\boldsymbol{y}') \right) \right] dt.$$

With a recursive manner, we have

$$\text{KL}\left( q_{T-\delta}^\leftarrow \big\| \hat{q}_{T-\delta} \right) \leq \text{KL}\left( q_0^\leftarrow \big\| \hat{q}_0 \right) + L_{\text{SE}}(\tilde{v}) = \text{KL}\left( q_T^\rightarrow \big\| \hat{q}_0 \right) + L_{\text{SE}}(\tilde{v}) \leq (1 + e^{-T})^d - 1 + T\epsilon_{\text{score}}^2,$$

where the last inequality follows from Lemma 2 and Assumption **[A1]**

$$\hat{q}_0(\boldsymbol{y}) = \tilde{q}_T(\boldsymbol{y}) \propto \exp(-T \cdot (d - \text{numK}(\boldsymbol{y}))).$$

If we set

$$T \geq \ln(4d/\epsilon^2) \quad \text{and} \quad \epsilon_{\text{score}} \leq T^{-1/2} \cdot \epsilon,$$

it has $(1 + e^{-T})^d - 1 \leq \epsilon^2$ and $T\epsilon_{\text{score}}^2 \leq \epsilon^2$, which means $\text{KL}\left( q_{T-\delta}^\leftarrow \big\| \hat{q}_{T-\delta} \right) \leq 2\epsilon^2$.

**Bounding** $\mathrm{TV}\left(q_*, q_\delta^\rightarrow\right)$   We adopt the proof strategy of Theorem 6 in Chen & Ying (2024). Consider the forward process $(X_t)_{t\geq0}$. By the coupling characterization of the total variation distance, we have

$$\mathrm{TV}\left(q_*, q_\delta^\rightarrow\right) \coloneqq \inf_{\gamma\in\Gamma(q_*,q_\delta^\rightarrow)} \mathbb{P}_{(u,v)\sim\gamma}[u \neq v] \leq \mathbb{P}(\mathbf{y} \neq \mathbf{y}'),$$

where $\Gamma(q_*, q_\delta^\rightarrow)$ is the set of all couplings of $(q_*, q_\delta^\rightarrow)$, and the inequality holds because $(\mathbf{y}, \mathbf{y}')$ gives a coupling of $(q_*, q_\delta^\rightarrow)$. Without loss of generality, we suppose $q_0^\rightarrow(\boldsymbol{y}) > 0$ for all $\mathrm{numK}\,(\boldsymbol{y}) = 0$, then, combining the transition kernel given Lemma 8 and Assumption **[A2]**, we have

$$\mathbb{P}(\mathbf{y} = \mathbf{y}') = \sum_{\boldsymbol{y}\in\mathcal{Y},\mathrm{numK}(\boldsymbol{y})=0} q_0^\rightarrow[\boldsymbol{y}] \cdot q_{\delta|0}^\rightarrow(\boldsymbol{y}|\boldsymbol{y}) = \sum_{\boldsymbol{y}\in\mathcal{Y},\mathrm{numK}(\boldsymbol{y})=0} q_0^\rightarrow(\boldsymbol{y}) \cdot e^{-\delta d} = e^{-\delta d}.$$

Thus, by choosing $\delta \leq \epsilon/d$, we have

$$\delta \leq d^{-1}\epsilon \leq d^{-1} \cdot \ln\left(\frac{1}{1-\epsilon}\right) \quad \Rightarrow \quad e^{\delta d} \leq \frac{1}{1-\epsilon} \quad \Rightarrow \quad \mathrm{TV}\left(q_*, q_\delta^\rightarrow\right) \leq 1 - e^{-\delta d} \leq \epsilon. \quad (68)$$

Finally, we have

$$\mathrm{TV}\left(q_0^\rightarrow, \hat{q}_{T-\delta}^\leftarrow\right) \leq \mathrm{TV}\left(q_0^\rightarrow, q_\delta^\rightarrow\right) + \mathrm{TV}\left(q_{T-\delta}^\leftarrow, \hat{q}_{T-\delta}\right) \leq \epsilon + \sqrt{\frac{1}{2}\mathrm{KL}\left(q_{T-\delta}^\leftarrow\big\|\hat{q}_{T-\delta}\right)} \leq 2\epsilon.$$

Hence the proof is completed. $\qquad\square$

### D.3   The complexity of Alg. 1

**Theorem 4** (The complexity of Alg. 1). *Suppose Assumption **[A1]** and **[A2]** hold, following from the settings shown in Theorem 3, if we implement Alg. 1 with*

$$t_w - t_{w-1} = \eta \quad where \quad w \in \{1, 2, \ldots W\}, \quad W = (T - \delta)/\eta, \quad \eta = \epsilon/2d, \quad and \quad \epsilon < 1$$

*the expectation of iteration/score estimation complexity of Alg. 1 will be upper bounded by*

$$2K(d - \epsilon^2/4) + 12Kd\ln d$$

*to achieve $\mathrm{TV}\left(q_*, \hat{q}\right) \leq 2\epsilon$ where $\hat{p}$ denotes the underlying distribution of generated samples.*

*Proof.* We denote $\{\hat{\mathbf{y}}_t\}_{t=0}^{T-\delta}$ to present the reverse process. For a specific trajectory, e.g., $\{\hat{\mathbf{y}}_t\}_{t=0}^{T-\delta} = \{\hat{\boldsymbol{y}}\}_{t=0}^{T-\delta}$, the total expected iteration number will be equivalent to the summation of Poisson expectations of $W$ segments, i.e.,

$$\sum_{i=1}^{W} \beta_{t_w}(\hat{\boldsymbol{y}}_{t_{w-1}}) \cdot (t_w - t_{w-1}) = \frac{K \cdot \mathrm{numK}\left(\hat{\boldsymbol{y}}_{t_{w-1}}\right)}{e^{T-t_w} - 1} \cdot (t_w - t_{w-1}),$$

which means the expected iteration number of the reverse process can be written as

$$\mathbb{E}\left[\sum_{w=1}^{W} \beta_{t_w}(\hat{\mathbf{y}}_{t_{w-1}}) \cdot (t_w - t_{w-1})\right] = \sum_{w=1}^{W} \mathbb{E}[\mathrm{numK}(\hat{\mathbf{y}}_{t_{w-1}})] \cdot \frac{K}{e^{(T-t_w)} - 1} \cdot (t_w - t_{w-1}). \quad (69)$$

Although $\mathbb{E}[\mathrm{numK}(\hat{\mathbf{y}}_{t_{w-1}})]$ is respect to the practical distribution $\hat{\mathbf{y}}_{t_{w-1}} \sim \hat{q}_{t_{w-1}}$, we can approximate it by the forward marginal distribution, i.e.,

$$\mathbb{E}[\mathrm{numK}(\mathbf{y}_{t_{w-1}}^\leftarrow)] = \mathbb{E}[\mathrm{numK}(\mathbf{y}_{T-t_{w-1}}^\rightarrow)] \quad \text{where} \quad \mathbf{y}_{t_{w-1}}^\leftarrow \sim q_{t_{w-1}}^\leftarrow \text{ and } \mathbf{y}_{T-t_{w-1}}^\rightarrow \sim q_{t_{w-1}}^\rightarrow$$

Specifically, with Assumption **[A2]**, we have $\mathbb{E}[\mathrm{numK}\,(\mathbf{y}_0^\rightarrow)] = 0$. Under this condition, the transition kernel becomes

$$q_{t|0}^\rightarrow(\boldsymbol{y}|\boldsymbol{y}') = \prod_{i=1}^{d} \Bigg[ \underbrace{\left(1 - \mathbf{1}_{(K,K)}(\boldsymbol{y}_i, \boldsymbol{y}_i')\right) \cdot \mathbf{1}_0(\boldsymbol{y}_i - \boldsymbol{y}_i') \cdot e^{-t}}_{\text{remain non-mask token}}$$

$$+ \underbrace{\left(1 - \mathbf{1}_{(K,K)}(\boldsymbol{y}_i, \boldsymbol{y}_i')\right) \cdot \mathbf{1}_K(\boldsymbol{y}_i) \cdot (1 - e^{-t})}_{\text{turn into mask token}} \Bigg].$$

due to Lemma 8. Let $\mathbb{P}[\mathrm{numK}\,(\mathbf{y}_t^\rightarrow) = k]$ be the probability that exactly $k$ out of the $d$ coordinates are mask tokens ($K$) at time $t$. Because each of the $d$ coordinates evolves independently (and identically, each with probability $1 - e^{-t}$ of being the mask token at time $t$), we get a standard Binomial random variable:

- Each coordinate is $K$ with probability $1 - e^{-t}$.

- Each coordinate is non-$K$ with probability $e^{-t}$.

Hence, we have

$$\mathbb{P}[\mathrm{numK}\,(\mathbf{y}_t^{\rightarrow}) = k] = C_d^k \cdot (1 - e^{-t})^k \cdot (e^{-t})^{d-k} \quad \text{and} \quad \mathbb{E}[\mathrm{numK}\,(\mathbf{y}_t^{\rightarrow}) = k] = d \cdot (1 - e^{-t}).$$

Then, for any $w$, we have $\mathbb{E}[\mathrm{numK}(\mathbf{y}_{T-t_{w-1}}^{\rightarrow})] = d \cdot (1 - e^{-(T-t_{w-1})})$. Under the settings shown in Theorem 3, we have

$$\mathrm{TV}\left(q_{T-t_{w-1}}^{\rightarrow}, \hat{q}_{t_{w-1}}\right) \leq \mathrm{TV}\left(q_{T-t_W}^{\rightarrow}, \hat{q}_{t_W}\right) \leq 2\epsilon,$$

which implies

$$\left|\mathbb{E}[\mathrm{numK}(\hat{\mathbf{y}}_{t_{w-1}})] - \mathbb{E}[\mathrm{numK}(\mathbf{y}_{t_{w-1}}^{\leftarrow})]\right| \leq d \cdot \mathrm{TV}\left(q_{T-t_{w-1}}^{\rightarrow}, \hat{q}_{t_{w-1}}\right) \leq 2d\epsilon.$$

Then, we have

$$\mathbb{E}[\mathrm{numK}(\hat{\mathbf{y}}_{t_{w-1}})] \leq \mathbb{E}[\mathrm{numK}(\mathbf{y}_{t_{w-1}}^{\leftarrow})] + 2d\epsilon = \mathbb{E}[\mathrm{numK}(\mathbf{y}_{T-t_{w-1}}^{\rightarrow})] + 2d\epsilon$$

$$= d \cdot (1 - e^{-(T-t_{w-1})}) + 2d\epsilon. \tag{70}$$

Plugging Eq. (70) into Eq. (69), we have

$$\mathbb{E}\left[\sum_{w=1}^{W} \beta_{t_w}(\hat{\mathbf{y}}_{t_{w-1}}) \cdot (t_w - t_{w-1})\right]$$

$$\leq \sum_{w=1}^{W} d \cdot (1 - e^{-(T-t_{w-1})}) \frac{K}{e^{(T-t_w)} - 1} \cdot (t_w - t_{w-1})$$

$$+ \sum_{w=1}^{W} 2d\epsilon \cdot \frac{K}{e^{(T-t_w)} - 1} \cdot (t_w - t_{w-1}) \tag{71}$$

$$= \underbrace{Kd \cdot \sum_{w=1}^{W} e^{-(T-t_w)} \cdot (t_w - t_{w-1}) \cdot \frac{1 - e^{-(T-t_{w-1})}}{1 - e^{-(T-t_w)}}}_{\text{Term 1}}$$

$$+ \underbrace{2Kd\epsilon \cdot \sum_{w=1}^{W} \left(e^{T-t_w} - 1\right)^{-1} \cdot (t_w - t_{w-1})}_{\text{Term 2}}$$

Then, we suppose the segments share the same length $\eta$, i.e.,

$$t_w - t_{w-1} = \eta \quad \text{where} \quad w \in \{1, 2, \ldots W\}, \quad W = (T - \delta)/\eta, \quad \text{and} \quad \eta = \epsilon/2d.$$

Under these conditions, we have

$$\eta \leq \frac{\delta}{2} \leq \ln(\frac{1}{2} + \frac{e^{\delta}}{2}) \quad \Rightarrow \quad e^{\eta} \leq \frac{e^{\delta}}{2} + \frac{1}{2} \leq \frac{e^{(T-t_{w-1})}}{2} + \frac{1}{2} \quad \forall\, w \in \{1, \ldots, W\}$$

$$\Rightarrow \quad e^{\eta} \leq \frac{1 + e^{-(T-t_{w-1})}}{2e^{-(T-t_{w-1})}} \quad \Rightarrow \quad 2 \cdot e^{-(T-t_{w-1}-\eta)} \leq 1 + e^{-(T-t_{w-1})} \tag{72}$$

$$\Rightarrow \quad 1 - e^{-(T-t_{w-1})} \leq 2 - 2e^{-(T-t_{w-1}-\eta)} \quad \Rightarrow \quad \frac{1 - e^{-(T-t_{w-1})}}{1 - e^{-(T-t_w)}} \leq 2.$$

Plugging these results into Term 1 of Eq. (71), we have

$$\text{Term 1} = 2Kd \cdot \sum_{w=1}^{W} e^{-(T-t_w)} \cdot \eta = 2Kd \cdot \sum_{w=1}^{W} e^{-(T-w\eta)} \cdot \eta$$

$$= 2Kd \cdot \eta \cdot e^{-T} \cdot \frac{e^{(W+1)\eta} - e^{\eta}}{e^{\eta} - 1} \leq 2Kd \cdot e^{\eta} \cdot \left(e^{-\delta} - e^{-T}\right) \leq 2Kd \cdot (1 - e^{-T}) \tag{73}$$

$$= 2Kd \cdot \left(1 - \frac{\epsilon^2}{4d}\right).$$

Moreover, we have

$$\frac{e^{T-t_{w-1}}-1}{e^{T-t_w}-1} = \frac{e^{T-t_{w-1}}}{e^{T-t_w}} \cdot \frac{1-e^{-(T-t_{w-1})}}{1-e^{-(T-t_w)}} \le e^\eta \cdot 2 \le 2e,$$

where the first inequality follows from Eq. (72) and the last inequality is established when $\eta \le 1$. Then, Term 2 of Eq. (71) can be upper bounded as

$$\text{Term 2} = 2Kd\epsilon \cdot \sum_{w=1}^{W} \frac{\eta}{e^{T-t_w}-1} \le 4e \cdot Kd\epsilon \cdot \sum_{w=1}^{W} \frac{\eta}{e^{T-t_{w-1}}-1} \le 4e \cdot Kd\epsilon \cdot \sum_{w=1}^{W} \frac{\eta}{T-t_{w-1}}$$

$$\le 4e \cdot dK\epsilon \cdot \int_0^{T-\delta} \frac{1}{T-t}\mathrm{d}t = 4e \cdot dK\epsilon \cdot \ln\frac{T}{\delta} \le 4e \cdot dK\epsilon \cdot \ln\frac{4d^2}{\epsilon^3} \le 12e \cdot Kd\ln d \cdot \epsilon \ln\frac{1}{\epsilon} \tag{74}$$

where the last inequality follows from

$$4 \le d \quad \text{and} \quad \ln\frac{d^3}{\epsilon^3} = 3\ln\frac{d}{\epsilon} \le 3\ln d\ln\frac{1}{\epsilon}$$

without loss of generality. Moreover, when $\epsilon < 1$, we have

$$\epsilon \ln\frac{1}{\epsilon} \le e^{-1},$$

which follows from the monotonicity of the function $x\ln x$. Under this condition, the RHS of Eq. (74) has the following bound

$$\text{Term 2} \le 12 \cdot Kd\ln d. \tag{75}$$

Finally, plugging Eq. (73) and Eq. (75) into Eq. (71), the expected calls of discrete scores will be

$$\mathbb{E}\left[\sum_{w=1}^{W} \beta_{t_w}(\hat{\mathbf{y}}_{t_{w-1}}) \cdot (t_w - t_{w-1})\right] \le 2K(d-\epsilon^2) + 12Kd\ln d.$$

Hence, the proof is completed. □

**Corollary 5.** *Suppose Assumption [A1] hold, following from the settings shown in Theorem 3, if we implement Alg. 1 with*

$$t_w - t_{w-1} = \eta \quad \text{where} \quad w \in \{1, 2, \ldots W\}, \quad W = (T-\delta)/\eta, \quad \eta = \epsilon/2d, \quad \text{and} \quad \epsilon < 1$$

*the expectation of iteration/score estimation complexity of Alg. 1 will be upper bounded by*

$$\min\left\{O(Kd\ln(d/\epsilon)), O\left(Kd \cdot \frac{\mathbb{E}[\text{numK}(\mathbf{y}_0^{\to})]}{\epsilon}\right)\right\} + O(Kd\ln d)$$

*to achieve* $\text{TV}(q_*, \hat{q}) \le 2\epsilon$ *where* $\hat{p}$ *denotes the underlying distribution of generated samples.*

*Proof.* Similar to the proof shown in Theorem 4, the expected iteration number of the reverse process can be written as

$$\mathbb{E}\left[\sum_{w=1}^{W} \beta_{t_w}(\hat{\mathbf{y}}_{t_{w-1}}) \cdot (t_w - t_{w-1})\right] = \sum_{w=1}^{W} \mathbb{E}[\text{numK}(\hat{\mathbf{y}}_{t_{w-1}})] \cdot \frac{K}{e^{(T-t_w)}-1} \cdot (t_w - t_{w-1}).$$

Although $\mathbb{E}[\text{numK}(\hat{\mathbf{y}}_{t_{w-1}})]$ is respect to the practical distribution $\hat{\mathbf{y}}_{t_{w-1}} \sim \hat{q}_{t_{w-1}}$, we can approximate it by the forward marginal distribution, i.e.,

$$\mathbb{E}[\text{numK}(\mathbf{y}_{t_{w-1}}^{\leftarrow})] = \mathbb{E}[\text{numK}(\mathbf{y}_{T-t_{w-1}}^{\to})] \quad \text{where} \quad \mathbf{y}_{t_{w-1}}^{\leftarrow} \sim q_{t_{w-1}}^{\leftarrow} \text{ and } \mathbf{y}_{T-t_{w-1}}^{\to} \sim q_{t_{w-1}}^{\to}.$$

Let $\mathbb{P}[\text{numK}(\mathbf{y}_t^{\to}) = k]$ be the probability that exactly $k$ out of the $d$ coordinates are mask tokens $(K)$ at time $t$ presented as

$$\mathbb{P}[\text{numK}(\mathbf{y}_t^{\to}) = k] = \sum_{i=0}^{k} \mathbb{P}[\text{numK}(\mathbf{y}_t^{\to}) = k|\text{numK}(\mathbf{y}_0^{\to}) = i] \cdot \Pr[\text{numK}(\mathbf{y}_0^{\to}) = i].$$

Because each of the $d$ coordinates evolves independently (and identically, each with probability $1 - e^{-t}$ of being the mask token at time $t$), we get a standard Binomial random variable:

- Each coordinate is $K$ with probability $1 - e^{-t}$.

- Each coordinate is non-$K$ with probability $e^{-t}$.

Hence, we have

$$\mathbb{P}[\text{numK}(\mathbf{y}_t^{\rightarrow}) = k | \text{numK}(\mathbf{y}_0^{\rightarrow}) = i] = \underbrace{C_{d-i}^{k-i}}_{\text{unmask}\rightarrow\text{mask count}} \cdot \underbrace{(1 - e^{-t})^{k-i}}_{\text{prob of mask transition}} \cdot \underbrace{(e^{-t})^{d-k}}_{\text{prob of unmask kept}} .$$

Under this condition, the expected number of MASK token at forward time $t$ will become

$$\mathbb{E}[\text{numK}(\mathbf{y}_t^{\rightarrow})] = \sum_{k=0}^{d} k \cdot \mathbb{P}[\text{numK}(\mathbf{y}_t^{\rightarrow}) = k]$$

$$= \sum_{k=0}^{d} \sum_{i=0}^{k} C_{d-i}^{k-i} \cdot (1 - e^{-t})^{k-i} \cdot (e^{-t})^{d-k} \cdot \mathbb{P}[\text{numK}(\mathbf{y}_0^{\rightarrow}) = i] \qquad (76)$$

$$= \sum_{i=0}^{d} \mathbb{P}[\text{numK}(\mathbf{y}_0^{\rightarrow}) = i] \cdot \underbrace{\sum_{k=i}^{d} C_{d-i}^{k-i} \cdot (1 - e^{-t})^{k-i} \cdot (e^{-t})^{d-k}}_{\text{Term 1}} .$$

For Term 1 in Eq. (76), suppose $j = k - i$, we have

$$\text{Term 1} = \sum_{j=0}^{d-i} (j + i) \cdot C_{d-i}^{j} \cdot (1 - e^{-t})^{j} \cdot (e^{-t})^{(d-i-j)}$$

$$= \sum_{j=0}^{d-i} j \cdot C_{d-i}^{j} \cdot (1 - e^{-t})^{j} \cdot (e^{-t})^{(d-i-j)} + i \cdot \sum_{j=0}^{d-i} C_{d-i}^{j} \cdot (1 - e^{-t})^{j} \cdot (e^{-t})^{(d-i-j)}$$

$$= \sum_{j=0}^{d-i} j \cdot C_{d-i}^{j} \cdot (1 - e^{-t})^{j} \cdot (e^{-t})^{(d-i-j)} + i \cdot (1 - e^{-t} + e^{-t})^{d-i} = d - (d-i)e^{-t}$$

where the last equation follows from the expectation of binomial distributions. Then, Eq. (76) can be written as

$$\mathbb{E}[\text{numK}(\mathbf{y}_t^{\rightarrow})] = \sum_{i=0}^{d} \mathbb{P}[\text{numK}(\mathbf{y}_0^{\rightarrow}) = i] \cdot \left( d - d \cdot e^{-t} + i \cdot e^{-t} \right)$$

$$= d \cdot \left( 1 - (1 - \mathbb{E}[\text{numK}(\mathbf{y}_0^{\rightarrow})]/d) \cdot e^{-t} \right).$$

Without loss of generality, we suppose

$$r_0 := 1 - \mathbb{E}[\text{numK}(\mathbf{y}_0^{\rightarrow})]/d > 0.$$

Then, following from Eq. (70), we have

$$\mathbb{E}[\text{numK}(\hat{\mathbf{y}}_{t_{w-1}})] \leq d \cdot (1 - r_0 \cdot e^{-(T-t_{w-1})}) + 2d\epsilon,$$

and

$$\mathbb{E}\left[ \sum_{w=1}^{W} \beta_{t_w}(\hat{\mathbf{y}}_{t_{w-1}}) \cdot (t_w - t_{w-1}) \right] \leq \underbrace{Kd \cdot \sum_{w=1}^{W} e^{-(T-t_w)} \cdot (t_w - t_{w-1}) \cdot \frac{1 - r_0 \cdot e^{-(T-t_{w-1})}}{1 - e^{-(T-t_w)}}}_{\text{Term 1}}$$

$$+ \underbrace{2Kd\epsilon \cdot \sum_{w=1}^{W} \left( e^{T-t_w} - 1 \right)^{-1} \cdot (t_w - t_{w-1})}_{\text{Term 2}}$$

$$(77)$$

Here the second term can be upper bounded as

$$\text{Term 2} \leq 12 \cdot Kd \ln d$$

by choosing the mixing time $T$ and early stopping time $\delta$ as Theorem 3, which follows from Eq. (74).

For Term 1 of Eq. (77), we will discuss it in categories. Suppose the expected number of mask token satisfies

$$\mathbb{E}[\text{numK}\,(\mathbf{y}_0^{\rightarrow})] \leq C_0 \cdot \epsilon \quad \Leftrightarrow \quad r_0 \geq 1 - C_0 \cdot \epsilon/d,$$

and the segments share the same length $\eta$, i.e.,

$$t_w - t_{w-1} = \eta \quad \text{where} \quad w \in \{1, 2, \ldots W\}, \quad W = (T-\delta)/\eta, \quad \text{and} \quad \eta = \epsilon/2d,$$

following from Eq. (72), we have

$$\eta \leq \frac{\delta}{2} \Rightarrow \quad \frac{1 - e^{-(T-t_{w-1})}}{1 - e^{-(T-t_w)}} \leq 2.$$

Combining with the following fact, i.e.,

$$\frac{(1-r_0) \cdot e^{-(T-t_{w-1})}}{1 - e^{-(T-t_w)}} = \frac{1 - r_0}{e^{(T-t_{w-1})} - e^{\eta}} = (1-r_0) \cdot e^{-\eta} \cdot \left(e^{T-t_{w-1}-\eta} - 1\right)^{-1}$$
$$\leq (1-r_0) \cdot \delta^{-1} = C_0,$$

Eq. (73) demonstrates that

$$\text{Term 1} \leq (2 + C_0) \cdot Kd \cdot \left(1 - \frac{\epsilon^2}{4d}\right).$$

On the other hand, we have

$$\text{Term 1} \leq Kd \cdot \sum_{w=1}^{W} \frac{\eta}{e^{T-t_w} - 1} \leq Kd \sum_{w=1}^{W} \frac{\eta}{T - t_w} \leq 1.5Kd \sum_{w=1}^{W} \frac{\eta}{T - t_{w-1}}$$
$$\lesssim 1.5Kd \cdot \int_{\delta}^{1} t^{-1} \mathrm{d}t \leq 1.5Kd \ln(1/\delta) = 1.5Kd \ln(d/\epsilon),$$

where the forth inequality follows from the choice of $\eta$, i.e.,

$$\eta \leq \delta/2 \quad \Rightarrow \quad (T - t_{w-1}) - (T - t_w) = \eta \leq \frac{\delta}{2} \leq \frac{T - t_w}{2}.$$

Hence, the total complexity will be

$$\min\left\{O(Kd\ln(d/\epsilon)), O\left(Kd \cdot \frac{\mathbb{E}[\text{numK}\,(\mathbf{y}_0^{\rightarrow})]}{\epsilon}\right)\right\} + O(Kd\ln d).$$

Hence, the proof is completed. $\qquad\square$

**Corollary 6.** *Suppose Assumption [A1] and [A2] hold, if Alg. 1 has*

$$t_0 = 0, \quad t_W = T - \delta, \quad \text{and} \quad \epsilon_{score} \leq T^{-1/2} \cdot \epsilon \quad \text{where} \quad T = \ln(4d/\epsilon^2) \quad \text{and} \quad \delta \leq d^{-1}\epsilon,$$

*and draw initial $\overline{\mathbf{y}}_0 \sim \delta_{[K, \ldots, K]}(\cdot)$, the TV distance between the target discrete distribution $q_*$ and the underlying distribution of the output particle $\overline{q}_{T-\delta}$ will satisfy* $\text{TV}\left(q_*, \overline{q}_{T-\delta}\right) \leq 2.5\epsilon$.

*Proof.* We consider a stochastic process $\{\overline{\mathbf{y}}_t\}_{t=0}^{T-\delta}$ which satisfies $\overline{\mathbf{y}}_t \sim \overline{q}_t$. The initial distribution is $\overline{\mathbf{y}}_0 \sim \overline{q}_0 = \delta_{[K,K,\ldots,K]}(\boldsymbol{y})$. Suppose the joint and conditional distribution are

$$(\overline{\mathbf{y}}_{t'}, \overline{\mathbf{y}}_t) \sim \overline{q}_{t',t} \quad \text{and} \quad \overline{q}_{t|t'}(\overline{\mathbf{y}}_t \| \overline{\mathbf{y}}_{t'}) = \overline{q}_{t,t'}(\overline{\mathbf{y}}_t, \overline{\mathbf{y}}_{t'})/\overline{q}_{t'}(\overline{\mathbf{y}}_{t'}) \quad \text{where} \quad t > t'.$$

Specifically, we suppose the random variables $\{\overline{\mathbf{y}}_t\}_{t=0}^{T-\delta}$ share the same transition as that in $\{\hat{\mathbf{y}}_t\}_{t=0}^{T-\delta}$ shown in Theorem 3, which means $\overline{q}_{t|t'} = \hat{q}_{t|t'}$ for any $t > t'$, which implies $\{\overline{\mathbf{y}}_t\}_{t=0}^{T-\delta}$ can be implemented by

    1. Initialize the particles as $\overline{\mathbf{y}}_0 \sim \overline{q}_0 = \delta_{[K,K,\ldots,K]}(\boldsymbol{y})$

    2. Update $\{\overline{\mathbf{y}}_t\}_{t>0}^{T-\delta}$ with Alg. 1

Then, due to the chain rule of TV distance, i.e., Lemma 16, we have

$$\text{TV}\left(\hat{q}_{T-\delta}, \overline{q}_{T-\delta}\right) \leq \text{TV}\left(\hat{q}_{T-\delta,0}, \overline{q}_{T-\delta,0}\right)$$
$$\leq \text{TV}\left(\hat{q}_0, \overline{q}_0\right) + \mathbb{E}_{\hat{\mathbf{y}}_0 \sim \hat{q}_0}\left[\text{TV}\left(\hat{q}_{T-\delta|0}, \overline{q}_{T-\delta|0}\right)\right] = \text{TV}\left(\hat{q}_0, \overline{q}_0\right). \tag{78}$$

Since $\overline{q}_0$ is the mask token dirac measure, we have

$$\text{TV}\left(\hat{q}_0, \overline{q}_0\right) = 1 - \hat{q}_0([\text{K}, \text{K}, \ldots, \text{K}]).$$

According to the proof of Lemma 2, we can easily find that

$$\hat{q}_0([\text{K}, \ldots, \text{K}]) = \frac{1}{(1 + e^{-T})^d}.$$

By requiring $T \geq \ln(4d/\epsilon)$ and $\epsilon \leq 1$, we have

$$T \geq \ln(4d/\epsilon) \quad \Rightarrow t \geq \ln(d/\ln(1+\epsilon/2)) \quad \Leftrightarrow \quad d \cdot e^{-T} \leq \ln(1+\epsilon/2)$$
$$\Rightarrow d\ln(1 + e^{-T}) \leq \ln(1+\epsilon/2) \quad \Leftrightarrow \quad (1 + e^{-T})^d - 1 \leq \epsilon/2.$$

That means

$$\text{TV}\left(\hat{q}_0, \overline{q}_0\right) = 1 - \frac{1}{(1 + e^{-T})^d} \leq 1 - \frac{1}{1 + \epsilon/2} \leq \epsilon/2.$$

Plugging this inequality into Eq. 78, we have $\text{TV}\left(\hat{q}_{T-\delta}, \overline{q}_{T-\delta}\right) \leq \epsilon/2$. Then combining it with Theorem 3, i.e., $\text{TV}\left(q_*, \hat{q}_{T-\delta}, \leq\right) 2\epsilon$, we have $\text{TV}\left(q_*, \overline{q}_{T-\delta}, \leq\right) 2.5\epsilon$. Hence, the proof is completed. □

## E    TECHNICAL LEMMAS

**Lemma 12** (Basic Kronecker product). *Supppose the Kronecker product for $n$ matrices defined on $\mathbb{R}^{d \times d}$, i.e.,*

$$\overline{A} := A_1 \otimes A_2 \otimes \ldots \otimes A_n,$$

*then we have*

$$\overline{A}_{[a_{1,i}, a_{2,i}, \ldots, a_{n,i}], [a_{1,j}, a_{2,j}, \ldots, a_{n,j}]} := \overline{A}_{\sum_{k=1}^n a_{k,i} \cdot d^{n-k}, \sum_{k=1}^n a_{k,j} \cdot d^{n-k}} = \prod_{k=1}^n [A_k]_{a_{k,i}, a_{k,j}}.$$

*Proof.* This lemma can easily be proved by the definition of Kronecker product. □

**Lemma 13** (Mixed-product property of Kronecker product). *Suppose the matrices $A, B, C, D \in \mathbb{R}^{d \times d}$, then, the products $AC$ and $BD$ are well-defined. We have*

$$(A \otimes B)(C \otimes D) = (AC) \otimes (BD).$$

*Proof.* We prove this by examining the product on the left-hand side, $(A \otimes B)(C \otimes D)$, and showing it coincides block-by-block with $(AC) \otimes (BD)$.

We starts from the definition of Kronecker products in blocks. By definition, the Kronecker product $A \otimes B$ can be seen as an $(d \times d)$ block matrix in which the $(i, j)$-th block is $a_{ij} B$. Hence,

$$A \otimes B = \begin{pmatrix} a_{11}B & a_{12}B & \cdots & a_{1n}B \\ a_{21}B & a_{22}B & \cdots & a_{2n}B \\ \vdots & \vdots & \ddots & \vdots \\ a_{m1}B & a_{m2}B & \cdots & a_{mn}B \end{pmatrix}.$$

Similarly,

$$C \otimes D = \begin{pmatrix} c_{11}D & c_{12}D & \cdots & c_{1r}D \\ c_{21}D & c_{22}D & \cdots & c_{2r}D \\ \vdots & \vdots & \ddots & \vdots \\ c_{n1}D & c_{n2}D & \cdots & c_{nr}D \end{pmatrix}.$$

Then, we form the Product $(\boldsymbol{A} \otimes \boldsymbol{B})(\boldsymbol{C} \otimes \boldsymbol{D})$. When multiplying two block matrices, we sum over the matching inner block dimensions. Specifically, the $(i,k)$-block of $(\boldsymbol{A} \otimes \boldsymbol{B})(\boldsymbol{C} \otimes \boldsymbol{D})$ is given by

$$\sum_{j=1}^{n} \Big( \big(a_{ij} \boldsymbol{B}\big) \big(c_{jk} \boldsymbol{D}\big) \Big).$$

Inside each term, we treat $a_{ij} \boldsymbol{B}$ and $c_{jk} \boldsymbol{D}$ as scalar-matrix products. We can rewrite the expression as:

$$\sum_{j=1}^{n} a_{ij}\, c_{jk} \left(\boldsymbol{B}\boldsymbol{D}\right) \;=\; \left(\sum_{j=1}^{n} a_{ij}\, c_{jk}\right) \boldsymbol{B}\boldsymbol{D}.$$

Notice that the factor $\sum_{j=1}^{n} a_{ij} c_{jk}$ is precisely $(\boldsymbol{AC})_{ik}$, the $(i,k)$-th entry of the matrix product $\boldsymbol{AC}$. Thus, each $(i,k)$-block of $(\boldsymbol{A} \otimes \boldsymbol{B})(\boldsymbol{C} \otimes \boldsymbol{D})$ simplifies to

$$(\boldsymbol{AC})_{ik} \, (\boldsymbol{BD}).$$

Now observe that the Kronecker product $(\boldsymbol{AC}) \otimes (\boldsymbol{BD})$ can also be viewed as an $(m \times r)$ block matrix whose $(i,k)$-th block is

$$(\boldsymbol{AC})_{ik} \, (\boldsymbol{BD}).$$

Hence, the $(i,k)$-th block of $(\boldsymbol{AC}) \otimes (\boldsymbol{BD})$ matches exactly with the $(i,k)$-th block we computed for $(\boldsymbol{A} \otimes \boldsymbol{B})(\boldsymbol{C} \otimes \boldsymbol{D})$. Since these two matrices agree in every block of a $d^2 \times d^2$ partition, we conclude

$$(\boldsymbol{A} \otimes \boldsymbol{B})(\boldsymbol{C} \otimes \boldsymbol{D}) \;=\; (\boldsymbol{AC}) \otimes (\boldsymbol{BD}),$$

as desired. $\qquad\square$

**Lemma 14** (Kolmogorov backward theorem, adapted from Theorem 5.11 in  Särkkä & Solin (2019))**.** *For a specific SDE, if we denote the transition density from $\mathbf{x}(s)$ to $\mathbf{y}(t)$ as $p(\boldsymbol{y}, t|\boldsymbol{x}, s)$ , then it solves the backward Kolmogorov equation*

$$-\frac{\partial p(\boldsymbol{y}, t|\boldsymbol{x}, s)}{\partial s} = \mathcal{L}p(\boldsymbol{y}, t|\boldsymbol{x}, s)$$

*where $\mathcal{L}$ denotes the infinitesimal operator of the SDE.*

**Lemma 15** (The chain rule of KL divergence)**.** *Consider four random variables, $\mathbf{x}, \mathbf{z}, \tilde{\mathbf{x}}, \tilde{\mathbf{z}}$, whose underlying distributions are denoted as $p_x, p_z, q_x, q_z$. Suppose $p_{x,z}$ and $q_{x,z}$ denotes the densities of joint distributions of $(\mathbf{x}, \mathbf{z})$ and $(\tilde{\mathbf{x}}, \tilde{\mathbf{z}})$, which we write in terms of the conditionals and marginals as*

$$p_{x,z}(\boldsymbol{x}, \boldsymbol{z}) = p_{x|z}(\boldsymbol{x}|\boldsymbol{z}) \cdot p_z(\boldsymbol{z}) = p_{z|x}(\boldsymbol{z}|\boldsymbol{x}) \cdot p_x(\boldsymbol{x})$$
$$q_{x,z}(\boldsymbol{x}, \boldsymbol{z}) = q_{x|z}(\boldsymbol{x}|\boldsymbol{z}) \cdot q_z(\boldsymbol{z}) = q_{z|x}(\boldsymbol{z}|\boldsymbol{x}) \cdot q_x(\boldsymbol{x}).$$

*then we have*

$$\mathrm{KL}\left(p_{x,z}\middle\|q_{x,z}\right) = \mathrm{KL}\left(p_z\middle\|q_z\right) + \mathbb{E}_{\mathbf{z}\sim p_z}\left[\mathrm{KL}\left(p_{x|z}(\cdot|\mathbf{z})\middle\|q_{x|z}(\cdot|\mathbf{z})\right)\right]$$
$$= \mathrm{KL}\left(p_x\middle\|q_x\right) + \mathbb{E}_{\mathbf{x}\sim p_x}\left[\mathrm{KL}\left(p_{z|x}(\cdot|\mathbf{x})\middle\|q_{z|x}(\cdot|\mathbf{x})\right)\right]$$

*where the latter equation implies*

$$\mathrm{KL}\left(p_x\middle\|q_x\right) \le \mathrm{KL}\left(p_{x,z}\middle\|q_{x,z}\right).$$

**Lemma 16** (The chain rule of TV distance)**.** *Consider four random variables, $\mathbf{x}, \mathbf{z}, \tilde{\mathbf{x}}, \tilde{\mathbf{z}}$, whose underlying distributions are denoted as $p_x, p_z, q_x, q_z$. Suppose $p_{x,z}$ and $q_{x,z}$ denotes the densities of joint distributions of $(\mathbf{x}, \mathbf{z})$ and $(\tilde{\mathbf{x}}, \tilde{\mathbf{z}})$, which we write in terms of the conditionals and marginals as*

$$p_{x,z}(\boldsymbol{x}, \boldsymbol{z}) = p_{x|z}(\boldsymbol{x}|\boldsymbol{z}) \cdot p_z(\boldsymbol{z}) = p_{z|x}(\boldsymbol{z}|\boldsymbol{x}) \cdot p_x(\boldsymbol{x})$$
$$q_{x,z}(\boldsymbol{x}, \boldsymbol{z}) = q_{x|z}(\boldsymbol{x}|\boldsymbol{z}) \cdot q_z(\boldsymbol{z}) = q_{z|x}(\boldsymbol{z}|\boldsymbol{x}) \cdot q_x(\boldsymbol{x}).$$

*then we have*

$$\mathrm{TV}\left(p_{x,z}, q_{x,z}\right) \le \min\Big\{\mathrm{TV}\left(p_z, q_z\right) + \mathbb{E}_{\mathbf{z}\sim p_z}\left[\mathrm{TV}\left(p_{x|z}(\cdot|\mathbf{z}), q_{x|z}(\cdot|\mathbf{z})\right)\right],$$
$$\mathrm{TV}\left(p_x, q_x\right) + \mathbb{E}_{\mathbf{x}\sim p_x}\left[\mathrm{TV}\left(p_{z|x}(\cdot|\mathbf{x}), q_{z|x}(\cdot|\mathbf{x})\right)\right]\Big\}.$$

*Besides, we have*

$$\mathrm{TV}\left(p_x, q_x\right) \le \mathrm{TV}\left(p_{x,z}, q_{x,z}\right).$$

---

**Algorithm 2** FIRST HITTING SAMPLING

---

1: **Input:** The sequence length $d$, the vocabulary $\mathcal{V} = \{1, 2, \ldots, K\}$ where K is the mask token, the noise schedule $\alpha_t$ and its inverse function $\alpha^{-1}$, , the pretrained masked diffusion model $p_\theta$
2: $\tilde{\mathbf{y}}_0 = [K, K, \ldots, K]$.
3: $\tilde{\tau}_0 = 1$.
4: **for** $n = 0$ **to** $d - 1$ **do**
5:     Sample $u_n \sim \text{Uniform}(0, 1)$
6:     $\tilde{\tau}_{n+1} = \alpha^{-1}(1 - u_n^{d-n}(1 - \alpha_{\tilde{\tau}_n}))$
7:     Randomly and uniformly select an index $l$ from $\{i : \tilde{\mathbf{y}}_n^{(i)} = K\}$ (i.e., masked positions in $\tilde{\mathbf{y}}_n$)

8:     $\boldsymbol{p}_n = p_{\theta,l}(\cdot | \tilde{\mathbf{y}}_n, \tilde{\tau}_{n+1}) \in \mathbb{R}_+^K$
9:     $\tilde{\mathbf{y}}_{n+1} = \tilde{\mathbf{y}}_n, \tilde{\mathbf{y}}_{n+1}^{(l)} \sim \text{Cat}(\boldsymbol{p}_n(\cdot, l))$
10: **end for**
11: **return** $\tilde{\mathbf{y}}_d$.

---

# F    FHS CONVERGENCE UNDER TIME-INDEPENDENT SCORE PARAMETERIZATION

In the following, we will prove that the distribution generated by first hitting sampling (Zheng et al., 2024) approaches to the target data distribution $p_*$ in TV distance. The core step is to introduce our MATU as the reference probability path.

We starts from some additional notations. Specifically, suppose following two elements $\boldsymbol{y}, \hat{\boldsymbol{y}} \in \mathcal{Y} = \{1, 2, \ldots, K\}^d$ satisfying

$$\boldsymbol{y} = [\boldsymbol{y}_1, \ldots, \boldsymbol{y}_i, \ldots, \boldsymbol{y}_d] \qquad \hat{\boldsymbol{y}} = [\boldsymbol{y}_1, \ldots, \hat{\boldsymbol{y}}_i, \ldots, \boldsymbol{y}_d],$$

which means the Hamming distance between $\boldsymbol{y}$ and $\hat{\boldsymbol{y}}$ is 1 and they are only different at $i$–th coordinate. Suppose $\boldsymbol{y}_i = K$ and $\hat{\boldsymbol{y}}_i \neq K$, then we can define the conditional distribution at specific coordinate, e.g., $i$, given unmask tokens $\boldsymbol{y}_{\mathcal{K}^c(\boldsymbol{y})}$

$$q_{0,i}(\hat{\boldsymbol{y}}_i | \boldsymbol{y}_{\mathcal{K}^c(\boldsymbol{y})}) = \frac{\sum_{\tilde{\boldsymbol{y}} \in \mathcal{Y}^+, \tilde{\boldsymbol{y}}_{\mathcal{K}^c(\hat{\boldsymbol{y}})} = \hat{\boldsymbol{y}}_{\mathcal{K}^c(\hat{\boldsymbol{y}})}} q_0(\tilde{\boldsymbol{y}})}{\sum_{\tilde{\boldsymbol{y}} \in \mathcal{Y}^+, \tilde{\boldsymbol{y}}_{\mathcal{K}^c(\boldsymbol{y})} = \boldsymbol{y}_{\mathcal{K}^c(\boldsymbol{y})}} q_0(\tilde{\boldsymbol{y}})}$$

For the completeness of the analysis, we first show the FHS in Alg. 2.

**Bridge the discrete score estimation error and the pretrained masked diffusion models in FHS.** We need to note that the output of pretrained masked diffusion model satisfies

$$p_{\theta,i}(\cdot | \boldsymbol{y}, \tilde{\tau}_{n+1}) = p_{\theta,i}(\cdot | \boldsymbol{y}) \approx q_{0,i}(\hat{\boldsymbol{y}}_i | \boldsymbol{y}_{\mathcal{K}^c(\boldsymbol{y})}),$$

where first equation comes from the time-independent parameterization, and the second approximation comes from the training objective, i.e.,

$$\mathcal{L}_{\boldsymbol{w}}^d(\boldsymbol{y}_0) = \sum_{i=1}^d \boldsymbol{w}_i \cdot \mathbb{E}_{\mathbb{P}_{\mathbf{y}}[\text{numK}(\mathbf{y})=i | \boldsymbol{y}_0]} \left[ \sum_{\mathbf{y}_i = K} -\log p_{\theta,i}(\boldsymbol{y}_{0,i} | \boldsymbol{y}) \right]. \tag{79}$$

With proper settings on $\boldsymbol{w}$ and the change of summation order, the above training loss of FHS will be equivalent to the $\lambda$–DCE loss shown in Ou et al. (2024), i.e.,

$$\mathcal{L}_{\lambda-\text{DCE}}(\boldsymbol{y}_0) = \mathbb{E}_{\lambda \sim \text{Uniform}(0,1)} \frac{1}{\lambda} \cdot \mathbb{E}_{\mathbf{y}_\lambda \sim q_{\lambda|0}^{\rightarrow}(\cdot | \boldsymbol{y}_0)} \left[ \sum_{\boldsymbol{y}_{\lambda,i} = K} -\log p_\theta(\boldsymbol{y}_{0,i} | \mathbf{y}) \right].$$

Then, following from Appendix C.1 and Appendix C.2 in Ou et al. (2024), by choosing $\lambda(t) = 1 - e^{-t}$, with change of variable, the $\lambda$–DCE loss will be equivalent to the denoising score entropy loss, i.e.,

$$\mathcal{L}_{\text{DSE}}(\boldsymbol{y}_0) = \int_0^T \mathbb{E}_{\mathbf{y}_t \sim q_{t|0}^{\rightarrow}(\cdot | \boldsymbol{y}_0)} \left[ \sum_{\boldsymbol{y}' \neq \mathbf{y}_t} R^{\rightarrow}(\mathbf{y}_t, \boldsymbol{y}') \cdot \left( \frac{e^{-t}}{1 - e^{-t}} \cdot p_\theta(\boldsymbol{y}'_{\text{DiffIdx}(\boldsymbol{y}', \mathbf{y}_t)} | \mathbf{y}_t) \right. \right.$$

$$\left. \left. - \frac{e^{-t}}{1 - e^{-t}} \cdot \delta_{\boldsymbol{y}_{0,\text{DiffIdx}(\mathbf{y}_t, \boldsymbol{y}')}}(\boldsymbol{y}'_{\text{DiffIdx}(\mathbf{y}_t, \boldsymbol{y}')}) \cdot \log \left( \frac{e^{-t}}{1 - e^{-t}} \cdot p_\theta(\boldsymbol{y}'_{\text{DiffIdx}(\boldsymbol{y}', \mathbf{y}_t)} | \mathbf{y}_t) \right) \right) \right] dt.$$

Following from Theorem 3.4 in Lou et al. (2024), we note that DSE and SE share the same minimum, i.e.,

$$\arg\min_\theta \ \mathbb{E}_{\mathbf{y}_0 \sim q_*}[\mathcal{L}_{\mathrm{DSE}}(\mathbf{y}_0)]$$

$$= \arg\min_\theta \ \int_0^T \mathbb{E}_{\mathbf{y}_t \sim q_t^{\rightarrow}} \left[ \sum_{\boldsymbol{y}' \neq \mathbf{y}_t} R^{\rightarrow}(\mathbf{y}_t, \boldsymbol{y}') \cdot \left( \frac{e^{-t}}{1 - e^{-t}} \cdot p_\theta(\boldsymbol{y}'_{\mathrm{DiffIdx}(\boldsymbol{y}', \mathbf{y}_t)} | \mathbf{y}_t) \right. \right. \tag{80}$$

$$\left. \left. \frac{q_t^{\rightarrow}(\boldsymbol{y}')}{q_t^{\rightarrow}(\mathbf{y}_t)} \cdot \log \left( \frac{e^{-t}}{1 - e^{-t}} \cdot p_\theta(\boldsymbol{y}'_{\mathrm{DiffIdx}(\boldsymbol{y}', \mathbf{y}_t)} | \mathbf{y}_t) \right) \right) \right] := \arg\min_\theta \ \mathcal{L}_{\mathrm{SE}}(\theta)$$

By supposing

$$\tilde{v}_{t, \boldsymbol{y}_t}(\boldsymbol{y}') := \frac{e^{-t}}{1 - e^{-t}} \cdot p_\theta(\boldsymbol{y}'_{\mathrm{DiffIdx}(\boldsymbol{y}', \boldsymbol{y}_t)} | \boldsymbol{y}_t) \quad \text{where} \quad \mathrm{Ham}(\boldsymbol{y}', \boldsymbol{y}_t) = 1 \quad \text{and} \quad \boldsymbol{y}'_{\mathrm{DiffIdx}(\boldsymbol{y}', \boldsymbol{y}_t)} \neq \mathrm{K},$$

we know the Eq. 80 exactly matches Eq. 6.

Therefore, optimizing Eq. 79 in FHS is equivalent to parameterize the discrete score as

$$\frac{q_t^{\rightarrow}(\boldsymbol{y}')}{q_t^{\rightarrow}(\boldsymbol{y}_t)} = \frac{e^{-t}}{1 - e^{-t}} \cdot q_{0, \mathrm{DiffIdx}(\boldsymbol{y}', \boldsymbol{y}_t)}(\boldsymbol{y}'_{\mathrm{DiffIdx}(\boldsymbol{y}', \boldsymbol{y}_t)} | \boldsymbol{y}_{t, \mathcal{K}(\boldsymbol{y}_t)}) = v_{t, \boldsymbol{y}_t}(\boldsymbol{y}')$$

$$\approx \tilde{v}_{t, \boldsymbol{y}_t}(\boldsymbol{y}') := \frac{e^{-t}}{1 - e^{-t}} \cdot p_\theta(\boldsymbol{y}'_{\mathrm{DiffIdx}(\boldsymbol{y}', \boldsymbol{y}_t)} | \boldsymbol{y}_t),$$

and optimize Eq. 6. Following the analysis paradigm in this paper, we assume Assumption **[A1]** is also satisfies for this parametrization.

**Bridge the trajectories between FHS and MATU.** We have the following theorem.

**Theorem 7** (The convergence of Alg. 1). *Suppose Assumption [A1] and [A2] hold, if the discrete scores are parameterized by time-independent neural network as Ou et al. (2024), the TV distance between the target discrete distribution $q_*$ and the underlying distribution of the output particle $\overline{q}_0$ of Alg. 2 will satisfy* $\mathrm{TV}\,(q_*, \hat{q}_{T-\delta}) \leq 2\epsilon$.

*Proof.* Under this time-independent parameterization, we suppose the trajectory of MATU as $\{\hat{\boldsymbol{y}}_t\}_{t=0}^T$ whose underlying distribution is denoted as $\hat{\boldsymbol{y}}_t \sim \tilde{q}_t$. For FHS, we consider a sequence of random variables $\{\overline{\mathbf{y}}_k\}_{k \in \{0, 1, \ldots, d\}}$ where $\overline{\mathbf{y}}_k$ denotes the random variables after $(d - k)$-step update of FHS. We have $\mathrm{numK}\,(\overline{\mathbf{y}}_k) = k$. To investigate the TV distance between $\hat{\boldsymbol{y}}_{T-\delta}$ and $\overline{\mathbf{y}}_0$, we have

$$\mathrm{TV}\,(\hat{q}_{T-\delta}, \overline{q}_0) = \frac{1}{2} \cdot \sum_{\boldsymbol{y}, \mathrm{numK}(\boldsymbol{y})=0} |\overline{q}_0(\boldsymbol{y}) - \hat{q}_{T-\delta}(\boldsymbol{y})| + \frac{1}{2} \cdot \sum_{\boldsymbol{y}, \mathrm{numK}(\boldsymbol{y}) \neq 0} \hat{q}_{T-\delta}(\boldsymbol{y})$$

$$= \frac{1}{2} \cdot \sum_{\boldsymbol{y}, \mathrm{numK}(\boldsymbol{y})=0} |\overline{q}_0(\boldsymbol{y}) - \hat{q}_{T-\delta}(\boldsymbol{y})| + \overline{q}_0(\boldsymbol{y}) - \hat{q}_{T-\delta}(\boldsymbol{y}) \leq \sum_{\boldsymbol{y}, \mathrm{numK}(\boldsymbol{y})=0} |\hat{q}_{T-\delta}(\boldsymbol{y}) - \overline{q}_0(\boldsymbol{y})|$$

$$\tag{81}$$

Currently, we define a distribution sequence

$$\{p_k\}_{k \in \{0, 1, \ldots, d\}} \quad \text{where} \quad p_k(t) = \Pr\,[\text{the } k\text{-th transition happens at time } t].$$

Besides, suppose that at the transition time $t$ the particle is $\boldsymbol{y}'$, MATU implies the transition from $\boldsymbol{y}'$ to $\boldsymbol{y}$ follows

$$\Pr[\boldsymbol{y} | \text{transition time} = t \text{ and particle is } \boldsymbol{y}'] = \hat{R}_t(\boldsymbol{y}, \boldsymbol{y}') / \hat{R}_t(\boldsymbol{y}')$$

Under this setting, we have

$$
\begin{aligned}
\hat{q}_{T-\delta}(\boldsymbol{y}) &= \int_0^{T-\delta} p_{\mathrm{numK}(\boldsymbol{y})}(t) \cdot \sum_{\boldsymbol{y}',\mathrm{numK}(\boldsymbol{y}')=\mathrm{numK}(\boldsymbol{y})-1} \hat{q}_t(\boldsymbol{y}') \cdot \Pr[\boldsymbol{y}|\text{transition time} = t \text{ and particle is } \boldsymbol{y}'] \mathrm{d}t \\
&= \int_0^{T-\delta} p_{\mathrm{numK}(\boldsymbol{y})}(t) \cdot \sum_{\boldsymbol{y}',\mathrm{numK}(\boldsymbol{y}')=\mathrm{numK}(\boldsymbol{y})-1} \hat{q}_t(\boldsymbol{y}') \cdot \frac{\hat{R}_t(\boldsymbol{y},\boldsymbol{y}')}{\hat{R}_t(\boldsymbol{y}')} \mathrm{d}t \\
&= \int_0^{T-\delta} p_{\mathrm{numK}(\boldsymbol{y})}(t) \cdot \sum_{\boldsymbol{y}',\mathrm{numK}(\boldsymbol{y}')=\mathrm{numK}(\boldsymbol{y})-1} \hat{q}_t(\boldsymbol{y}') \cdot \frac{\tilde{R}_t(\boldsymbol{y},\boldsymbol{y}')}{\tilde{R}_t(\boldsymbol{y}')} \mathrm{d}t
\end{aligned}
\tag{82}
$$

Due to the time-independent paramterization of the discrete score, we have

$$
\tilde{R}_t(\boldsymbol{y},\boldsymbol{y}') = R^{\rightarrow}(\boldsymbol{y}',\boldsymbol{y}) \cdot \tilde{v}_{t,\boldsymbol{y}'}(\boldsymbol{y}) = R^{\rightarrow}(\boldsymbol{y}',\boldsymbol{y}) \cdot \frac{e^{-t}}{1-e^{-t}} \cdot p_\theta(\boldsymbol{y}_{\mathrm{DiffIdx}(\boldsymbol{y},\boldsymbol{y}')}|\boldsymbol{y}'),
$$

which implies it has

$$
\frac{\tilde{R}_t(\boldsymbol{y},\boldsymbol{y}')}{\tilde{R}_t(\boldsymbol{y}')} = \frac{p_\theta(\boldsymbol{y}_{\mathrm{DiffIdx}(\boldsymbol{y},\boldsymbol{y}')}|\boldsymbol{y}')}{\sum_{\boldsymbol{y}\neq\boldsymbol{y}',\mathrm{Ham}(\boldsymbol{y},\boldsymbol{y}')=1} p_\theta(\boldsymbol{y}_{\mathrm{DiffIdx}(\boldsymbol{y},\boldsymbol{y}')}|\boldsymbol{y}')} = p_\theta(\boldsymbol{y}_{\mathrm{DiffIdx}(\boldsymbol{y},\boldsymbol{y}')}|\boldsymbol{y}').
$$

Plugging this equation into Eq. 82, we have

$$
\begin{aligned}
\hat{q}_{T-\delta}(\boldsymbol{y}) &= \sum_{\boldsymbol{y}',\mathrm{numK}(\boldsymbol{y}')=\mathrm{numK}(\boldsymbol{y})-1} p_\theta(\boldsymbol{y}_{\mathrm{DiffIdx}(\boldsymbol{y},\boldsymbol{y}')}|\boldsymbol{y}') \int_0^{T-\delta} p_{\mathrm{numK}(\boldsymbol{y})}(t) \cdot \hat{q}_t(\boldsymbol{y}') \mathrm{d}t \\
&= \sum_{\boldsymbol{y}',\mathrm{numK}(\boldsymbol{y}')=\mathrm{numK}(\boldsymbol{y})-1} p_\theta(\boldsymbol{y}_{\mathrm{DiffIdx}(\boldsymbol{y},\boldsymbol{y}')}|\boldsymbol{y}') \cdot \sum_{\boldsymbol{y}'',\mathrm{numK}(\boldsymbol{y}'')=\mathrm{numK}(\boldsymbol{y}')-1} p_\theta(\boldsymbol{y}'_{\mathrm{DiffIdx}(\boldsymbol{y}',\boldsymbol{y}'')}|\boldsymbol{y}'') \\
&\quad \cdot \ldots \cdot \int_{t_{\mathrm{numK}(\boldsymbol{y})},t_{\mathrm{numK}(\boldsymbol{y})-1},\ldots,t_1} p_{\mathrm{numK}(\boldsymbol{y}),\mathrm{numK}(\boldsymbol{y}),\ldots,1}(t_{\mathrm{numK}(\boldsymbol{y})-1},\ldots,t_1) \cdot \hat{q}_\tau([\mathrm{K},\ldots,\mathrm{K}]) \mathrm{d} \\
&\leq \sum_{\boldsymbol{y}',\mathrm{numK}(\boldsymbol{y}')=\mathrm{numK}(\boldsymbol{y})-1} p_\theta(\boldsymbol{y}_{\mathrm{DiffIdx}(\boldsymbol{y},\boldsymbol{y}')}|\boldsymbol{y}') \cdot \sum_{\boldsymbol{y}'',\mathrm{numK}(\boldsymbol{y}'')=\mathrm{numK}(\boldsymbol{y}')-1} p_\theta(\boldsymbol{y}'_{\mathrm{DiffIdx}(\boldsymbol{y}',\boldsymbol{y}'')}|\boldsymbol{y}'') \\
&\quad \cdot \ldots \cdot \sum_{\boldsymbol{y}^{(1)},\mathrm{numK}(\boldsymbol{y}^{(1)})=1} p_\theta(\boldsymbol{y}^{(1)}_{\mathrm{DiffIdx}(\boldsymbol{y}^{(1)},[\mathrm{K},\ldots,\mathrm{K}])}|[\mathrm{K},\ldots,\mathrm{K}])\hat{q}_0([\mathrm{K},\ldots,\mathrm{K}]),
\end{aligned}
$$

where the last ineqaulity follows from the fact

$$
\hat{q}_\tau([\mathrm{K},\ldots,\mathrm{K}]) \leq \hat{q}_0([\mathrm{K},\ldots,\mathrm{K}]) \quad \forall \tau > 0.
$$

According to the update of FHS, we can easily find that

$$
\begin{aligned}
\overline{q}_0(\boldsymbol{y}) &= \sum_{\boldsymbol{y}',\mathrm{numK}(\boldsymbol{y}')=\mathrm{numK}(\boldsymbol{y})-1} p_\theta(\boldsymbol{y}_{\mathrm{DiffIdx}(\boldsymbol{y},\boldsymbol{y}')}|\boldsymbol{y}') \cdot \sum_{\boldsymbol{y}'',\mathrm{numK}(\boldsymbol{y}'')=\mathrm{numK}(\boldsymbol{y}')-1} p_\theta(\boldsymbol{y}'_{\mathrm{DiffIdx}(\boldsymbol{y}',\boldsymbol{y}'')}|\boldsymbol{y}'') \\
&\quad \cdot \ldots \cdot \sum_{\boldsymbol{y}^{(1)},\mathrm{numK}(\boldsymbol{y}^{(1)})=1} p_\theta(\boldsymbol{y}^{(1)}_{\mathrm{DiffIdx}(\boldsymbol{y}^{(1)},[\mathrm{K},\ldots,\mathrm{K}])}|[\mathrm{K},\ldots,\mathrm{K}]) \underbrace{\overline{q}_d([\mathrm{K},\ldots,\mathrm{K}])}_{=1}.
\end{aligned}
$$

Suppose the conditional distribution as

$$
\begin{aligned}
\overline{p}_\theta(\boldsymbol{y}|[\mathrm{K},\ldots,\mathrm{K}]) &= \sum_{\boldsymbol{y}',\mathrm{numK}(\boldsymbol{y}')=\mathrm{numK}(\boldsymbol{y})-1} p_\theta(\boldsymbol{y}_{\mathrm{DiffIdx}(\boldsymbol{y},\boldsymbol{y}')}|\boldsymbol{y}') \cdot \sum_{\boldsymbol{y}'',\mathrm{numK}(\boldsymbol{y}'')=\mathrm{numK}(\boldsymbol{y}')-1} p_\theta(\boldsymbol{y}'_{\mathrm{DiffIdx}(\boldsymbol{y}',\boldsymbol{y}'')}|\boldsymbol{y}'') \\
&\quad \cdot \ldots \cdot \sum_{\boldsymbol{y}^{(1)},\mathrm{numK}(\boldsymbol{y}^{(1)})=1} p_\theta(\boldsymbol{y}^{(1)}_{\mathrm{DiffIdx}(\boldsymbol{y}^{(1)},[\mathrm{K},\ldots,\mathrm{K}])}|[\mathrm{K},\ldots,\mathrm{K}]),
\end{aligned}
$$

then we have

$$
\begin{aligned}
\overline{q}_0(\boldsymbol{y}) - \hat{q}_{T-\delta}(\boldsymbol{y}) &\leq \overline{p}_\theta(\boldsymbol{y}|[\mathrm{K},\ldots,\mathrm{K}]) \cdot (\overline{q}_d([\mathrm{K},\ldots,\mathrm{K}]) - \hat{q}_0([\mathrm{K},\ldots,\mathrm{K}])) \\
&= \overline{p}_\theta(\boldsymbol{y}|[\mathrm{K},\ldots,\mathrm{K}]) \cdot (1 - \hat{q}_0([\mathrm{K},\ldots,\mathrm{K}]))
\end{aligned}
\tag{83}
$$

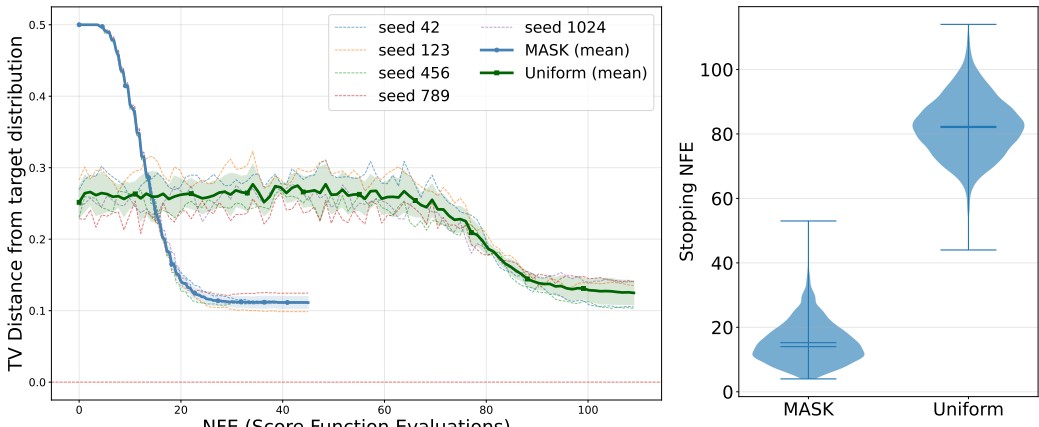

Figure 1: **Synthetic experiment results on sampling efficiency.** We compare our proposed Masked Discrete Diffusion (MASK) against the Uniform baseline with vocabulary size $K = 3$ and sequence length $d = 4$. **Left:** The Total Variation (TV) distance between the empirical and ground truth distributions as a function of the Number of (Score) Function Evaluations (NFE). The solid lines represent the mean over 5 seeds, and shaded regions indicate the standard deviations. Our method achieves faster convergence to the target distribution. **Right:** Violin plots illustrating the distribution of Stopping NFE. The MASK method requires significantly fewer evaluations to terminate compared to the Uniform baseline.

According to the proof of Lemma 2, we know that

$$\hat{q}_0([\mathrm{K}, \ldots, \mathrm{K}]) = (1 + e^{-T})^{-d} \quad \text{and} \quad (1 + e^{-T})^d - 1 \le \epsilon/2$$
$$\Rightarrow 0 \le 1 - (1 + e^{-T})^{-d} = 1 - \hat{q}_0([\mathrm{K}, \ldots, \mathrm{K}]) \le 1 - 1/(1 + \epsilon/2) \le \epsilon/2. \tag{84}$$

Combining Eq. 81, Eq. 83 and Eq. 84, we have

$$\mathrm{TV}\left(\hat{q}_{T-\delta}, \overline{q}_0\right) \le \epsilon/2 \quad \text{and} \quad \mathrm{TV}\left(q_*, \overline{q}_0\right) \le \mathrm{TV}\left(\hat{q}_{T-\delta}, \overline{q}_0\right) + \mathrm{TV}\left(q_*, \hat{q}_{T-\delta}\right) \le \epsilon$$

where last inequality follows from Theorem 3. Hence, the proof is completed. $\qquad\square$

# G EXPERIMENTS

## G.1 SYNTHETIC EXPERIMENTS.

We conduct synthetic experiments to validate our theoretical findings and compare the sampling efficiency of our Masked Discrete Diffusion model against the uniform baseline.

**Experiment Setup.** We utilize a state space defined by vocabulary size $K = 3$ and sequence length $d = 4$. The ground truth distribution, $p^*$, is constructed by assigning a random mass sampled uniformly from $(0, 1)$ to each of the $K^d$ possible sequences and normalizing the distribution. We report results averaged over 5 independent random seeds. For each seed, we generate 1000 trajectories using our method (Algorithm 1, MATU) and the truncated uniformization baseline with a uniform stationary distribution (adapted from Huang et al. (2025)). Performance is evaluated via the Total Variation (TV) distance between the empirical marginal distribution and $p^*$, plotted as a function of the Number of (Score) Function Evaluations (NFE). Quantitative results are shown in Figure 1, and illustrative sampling trajectories are visualized in Figure 2.

## G.2 REAL WORLD EXPERIMENTS

We consider to introduce our Alg. 1 (MATU) into the text generation task.

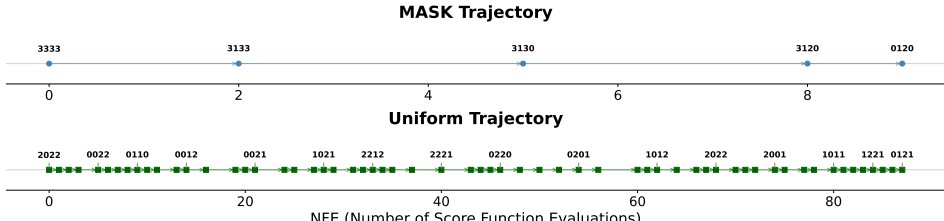

Figure 2: Visualization of individual sampling trajectories. The plots show single sampling paths, with labels indicating the intermediate discrete states. The MASK method (top) navigates the state space efficiently with few steps. In contrast, the Uniform baseline (bottom) exhibits diffusive behavior with many small steps—often reverting previous changes—resulting in a high NFE cost.

**Experimental Settings**   In this paragraph, we follow the problem setting as SEDD shown in Lou et al. (2024), and consider the unconditional text generation task with the small pretrained SEDD Absorbing model. The sequence length of generated sample is constrains as $d = 1024$, and the the vocabulary size will be $K = 50258$, including the mask token. We choose the typical Euler and Tweedie's $\tau$-leaping (analytic samples in Lou et al. (2024)'s implementation) as our baselines. For the step number choice, we only consider $\{1024, 2048\}$. Because MATU does not consider the conditional independent assumption for the reverse process. Under this condition, it requires at least $d$ steps to generate one no-mask sample.

**The inexact adaptation from MATU.**   In SEDD experiments, The exact implementation of Alg. 1 will require the inference complexity to be $K \times d = 50258 \times 1024$, which is far beyond an acceptable inference complexity. Since the choice of $K$ can be used to control the inference complexity, in the following experiment we will choose

$$K = \text{required steps/generated sequence length},$$

which is an inexact implementation of Alg. 1 (MATU), while makes it to be possible to be tuned via the choice of the step number. Moreover, the implementation of Euler and Tweedie's $\tau$-leaping is based on log-linear noise schedule, which means the transition rate matrix of the forward process satisfies

$$R_t^{\rightarrow}(\boldsymbol{y}, \boldsymbol{y}') = \sigma(t) R^{\rightarrow}(\boldsymbol{y}, \boldsymbol{y}') \quad \text{where} \quad \sigma(t) = \frac{1-\epsilon}{1-(1-\epsilon) \cdot t}$$

and $R^{\rightarrow}$ follows from Eq. 7. Under this condition, the reverse transition rate matrix will become

$$R_t^{\leftarrow}(\boldsymbol{y}, \boldsymbol{y}') \;\coloneqq\; \sigma(1-t) \cdot R^{\rightarrow}(\boldsymbol{y}', \boldsymbol{y}) \, \frac{q_t^{\leftarrow}(\boldsymbol{y})}{q_t^{\leftarrow}(\boldsymbol{y}')}$$

$$= \sigma(1-t) \cdot R^{\rightarrow}(\boldsymbol{y}', \boldsymbol{y}) \cdot s_{\theta, 1-t, \boldsymbol{y}'}(\boldsymbol{y}).$$

**Empirical Results.**   We use PPL and entropy as two criteria to measure the generation quality for different samplers. The results are summarized as the following tables. We will release the detailed code and implementation after the acceptance of this paper.

Table 3: Comparison of the inference generation performance, we calculate the average perplexity and entropy for 32 samples generated by Euler, Analytic and MATU. The experiments show even with an inexact implementation, MATU still outperform then other samplers consistently.

| Samplers | Steps | Avg Perplexity | Std Perplexity | Avg Entropy | Std Entropy | Wall-clock time |
|----------|-------|----------------|----------------|-------------|-------------|-----------------|
| Euler | 1024 | 41.42 | 11.68 | 7.588 | 0.301 | 27.35s/sample |
| Analytic | 1024 | 41.81 | 11.57 | 7.597 | 0.286 | 24.15s/sample |
| MATU | 1024 | 40.54 | 11.20 | 7.554 | 0.230 | 32.23s/sample |
| Euler | 2048 | 33.32 | 7.141 | 7.492 | 0.258 | 53.43s/sample |
| Analytic | 2048 | 32.50 | 6.952 | 7.489 | 0.250 | 46.88s/sample |
| MATU | 2048 | 31.82 | 6.717 | 7.394 | 0.332 | 60.05s/sample |

