# OpenReview forum: "On the Complexity Theory of Masked Discrete Diffusion: From $\mathrm{poly}(1/\epsilon)$ to Nearly $\epsilon$-Free"
_ICLR.cc/2026/Conference — Submitted to ICLR 2026_

### Official Review · Reviewer_t4vz · 2025-10-17

**Soundness:** 2
**Presentation:** 2
**Contribution:** 1
**Rating:** 2
**Confidence:** 4

**Summary:**

This paper studies the sampling complexity of masked diffusion models. The authors prove complexity bounds to achieve $\epsilon$-error in total variation, for Euler and Mask-Aware Truncated Uniformization (MATU) samplers, where the latter is an algorithm introduced in this paper to exploit specfic properties of masked diffusion. The primary contribution is that the sampling complexity of MATU is (nearly) $\epsilon$-free, improving upon the $\ln(1/\epsilon)$ dependence in the case of discrete diffusion with uniform transition.

**Strengths:**

It gives a mathematical proof that masked diffusion models with their proposed sampler (MATU) actually enjoy an $\epsilon$-free sampling complexity. The theory explains the difference between uniform and masked diffusions in terms of sampling complexity. It also removed some assumptions from the previous work on bounding discrete diffusion complexities, thereby widening the applicability of the theory to more general situations.

**Weaknesses:**

While there are several weaknesses, the main point ([W1]) is that it does not seem to be worthwhile considering more than $d$ sampling steps in masked diffusion, since there are only $d$ transition steps in forward (and thus backward) processes. Details are as follows:
- [W1] Masked diffusion (with the same and independent forward transition in each dimension) is time-agnostic; namely, the distributions conditioned on the number of masks is independent of time [1, Proposition 3.2]. We can also analytically sample the transition time and position/dimension [1, Proposition 4.1], so we just need $d$ function calls, each unmasking one position at a time, with errors only coming from the estimation error. While [1] uses a denoising parameterization (directly parameterizing $p(x_0^i|x_t)$ at each position $i$), it is easily convertible from/to the score parameterization [2, Proposition 3]. **Therefore, having** ***>d*** **sampling steps is not very meaningful in practice for masked diffusion.** In other words, an $\epsilon$-free sampling complexity is already known in the case of masked diffusion.
- [W2] While the authors introduce $\tilde{q}_t$ in Eq. 9 as an initial distribution for masked diffusion, people do not use such "partially masked and partially uniform" distributions for generative purposes. I think it comes from the time-homogeneous assumption of the forward diffusion noising, which is never used in the actual training of masked diffusion.
- [W3] Some citations should be clarified.
    - [W3-1] In Table 1, the rate of Ren et al. [3] is different from the original rate in KL divergence. Similarly, it is written that the "higher-order" method gives $O(\epsilon^{-1})$, which is different from the original paper, again in KL divergence. I assume that the authors implicitly use Pinsker's inequality to deduce those bounds, but it is not a fair comparison/citation, since the original work do not aim to give a tight bound in terms of TV distance.
    - [W3-2] In L46, the authors cite Nie et al. [4] to state "masked discrete diffusion has empirically outperformed uniform discrete diffusion" but the cited work does not mention uniform diffusion. I know masked diffusion models are more investigated, but there is also a uniform diffusion language model competitive with a masked model of similar scale [5].
- [W4] I feel that the order notation is used in an ambiguous way.
    - [W4-1] For example, in L53, the authors write "at least $O(\epsilon^{-1})$ steps" as if $\epsilon^{-1}$ would give a lower bound of the complexity. However, the $O$-notation only gives an upper bound and none of the cited work in L52 gives such a lower bound. The only lower bound I know in this context is the later half of [6, Theorem 1].
    - [W4-2] Whether the parameters $d$ or $K$ are regarded as constants (i.e., whether they are inside $O(\cdot)$) frequently changes throughout the manuscript. In addition, why does $K$ disappear only in the second term in L94, compared with Eq. 20?

Minor points:
- The definition of $\text{Ham}(y, y')$ should be $d-$ (current value).
- The value in Eq. 20 can be upper bounded by $2Kd+12Kd\ln d$, which is $\epsilon$-free, while the sampling algorithm can depend on $\epsilon$. Calling it "nearly" $\epsilon$-free is misleading.

References:
- [1] Zheng, K., Chen, Y., Mao, H., Liu, M. Y., Zhu, J., & Zhang, Q. (2025). Masked diffusion models are secretly time-agnostic masked models and exploit inaccurate categorical sampling. ICLR 2025.
- [2] Campbell, A., Benton, J., De Bortoli, V., Rainforth, T., Deligiannidis, G., & Doucet, A. (2022). A continuous time framework for discrete denoising models. NeurIPS 2022.
- [3] Ren, Y., Chen, H., Rotskoff, G. M., & Ying, L. (2025). How discrete and continuous diffusion meet: Comprehensive analysis of discrete diffusion models via a stochastic integral framework. ICLR 2025.
- [4] Nie, S., Zhu, F., You, Z., Zhang, X., Ou, J., Hu, J., Zhou, J., Lin, Y., Wen, J. R., & Li, C. (2025). Large language diffusion models. arXiv:2502.09992.
- [5] Sahoo, S. S., Deschenaux, J., Gokaslan, A., Wang, G., Chiu, J., & Kuleshov, V. (2025). The diffusion duality. ICML 2025.
- [6] Hayakawa, S., Takida, Y., Imaizumi, M., Wakaki, H., & Mitsufuji, Y. (2025). Distillation of discrete diffusion through dimensional correlations. ICML 2025.

**Questions:**

- [Q1] "Girsanov theory" is only mentioned in Conclusion. Can the authors provide a brief explanation and motivation for avoiding it?
- [Q2] Related to [W3-1], am I correct in assuming that some of the bounds in Table 1 are given by Pinsker's inequality from KL-based bounds? In that case, can the TV-based bounds in this paper also be transformed to bounds in KL (since it seems that many of the analyses in the appendix are conducted through KL divergence).

---

> ### Author Response · Authors · 2025-11-25
>
> We thank the reviewer for carefully reading our work and providing detailed feedback, and try to solve your concerns point-by-point in the following:
>
> > W1: Masked diffusion (with the same and independent forward transition in each dimension) is time-agnostic; namely, the distributions conditioned on the number of masks is independent of time [1, Proposition 3.2]. We can also analytically sample the transition time and position/dimension [1, Proposition 4.1], so we just need  $d$ function calls, each unmasking one position at a time, with errors only coming from the estimation error. While [1] uses a denoising parameterization (directly parameterizing $p(x_0^i|x_t)$ at each position $i$), it is easily convertible from/to the score parameterization [2, Proposition 3]. Therefore, having >d sampling steps is not very meaningful in practice for masked diffusion. In other words, an $\epsilon$-free sampling complexity is already known in the case of masked diffusion.
>
> A1: We respectfully disagree with the assessment that "an $\epsilon$-free sampling complexity is already known in the case of masked diffusion". When "analytically sampling the transition time and position / dimension", **the inference error caused by FHS [1] is beyond the score estimation error for mask diffusions with time-dependent paramterized discrete scores**, i.e., $s_{\theta,t,y}(y^\prime) \approx q^\to_t(y^\prime)/q^\to_t(y)$ as [6]. That means the sample complexity is unknown for FHS [1] to ensure the generated distribution is within $\epsilon$-distance of the true data distribution.
>
> The core issue is that FHS [1] does **not** exactly simulate the *practical/learned* reverse process: it simply chooses the next position to unmask **uniformly** from all remaining masked positions, without accounting for the possibility that different positions may have different transition rates in the learned reverse process. The condition “the number of masks is independent of time” holds *only* for an *ideal* reverse process, not necessarily for a *learned* one. Indeed, there could exist indices $i\not=j$ for which
> $$
> \sum_{y^\prime, s.t.\ \mathrm{Ham}(y^\prime, y)=1, \mathrm{DffIdx}(y^\prime,y)=i, y_{i}=K} s_{\theta,t,y}(y^\prime)\not = \sum_{y^\prime, s.t.\ \mathrm{Ham}(y^\prime, y)=1, \mathrm{DffIdx}(y^\prime,y)=j, y_{j}=K} s_{\theta,t,y}(y^\prime),
> $$
> implying that the *practical* reverse transition kernel has different rates for different positions, i.e.,
> $$
> \sum_{y^\prime, s.t.\ \mathrm{Ham}(y^\prime, y)=1, \mathrm{DffIdx}(y^\prime,y)=i, y_{i}=K}\underbrace{\lim_{\Delta t\rightarrow 0}
>     \Bigl[\frac{\hat{q}\_{t+\Delta t \mid t}(y^\prime \mid y) - \delta_{y}(y^\prime)}{\Delta t}\Bigr]}_{=\hat{R}(y^\prime,y)}\\
> $$
>
> $$
>  \not = \sum_{y^\prime, s.t.\ \mathrm{Ham}(y^\prime, y)=1, \mathrm{DffIdx}(y^\prime,y)=j, y_{j}=K} \lim_{\Delta t\rightarrow 0} \Bigl[\frac{\hat{q}\_{t+\Delta t \mid t}(y^\prime \mid y) - \delta_{y}(y^\prime)}{\Delta t}\Bigr].
> $$
> Moreover, FHS provides no rigorous argument regarding how uniformly selecting the position to denoise affects the final inference error. Therefore, **before our paper, it remains unknown whether an $\epsilon$-free sampling complexity is achievable for time-dependent masked diffusion.**  In contrast, our MATU algorithm precisely simulates the learned reverse process by incorporating position-dependent reverse transition rates. Specifically, in Step 10 of Alg. 1, it fully leverages $s_{\theta,t,y}(y')$ to handle different transition rates across positions.
>
> By comparison, in a *time-independent* parameterization [7]—i.e., $p_{\theta,y}(y^\prime) \approx q_0(y^\prime_{\mathrm{DiffIdx}(y,y^\prime)}||y^{UM})$, FHS [1] can *exactly* simulate the learned score trajectory with $d$ steps. The strong condition
> $$
> \sum_{y^\prime, s.t.\ \mathrm{Ham}(y^\prime, y)=1, \mathrm{DffIdx}(y^\prime,y)=i, y_{i}=K} p_{\theta,y} (y^\prime) = 1 = \sum q_0(y^\prime_i||y^{UM})
> $$
> for *each* $i$, ensures that every position has an identical reverse transition rate, making a uniform selection unbiased. **However, [1] does not provide a detailed and rigorous proof of this unbiasedness**, in our revision we fill this gap with **Theorem 7 (Appendix F)**, where we couple FHS’s trajectory to MATU’s dynamics.
>
> Finally, under these time-independent conditions, note that in MATU, once the particle $\hat{z}_t$ stays fixed, the transition matrix $\hat{R}_{z_0}(\cdot,\hat{z}_t)$ also stays fixed (because the discrete score is time-independent). Through discrete-score caching, **the NFE (number of function evaluations) in MATU also equals $d$**—matching FHS in terms of raw steps—and any additional rejection steps only add an $O(d\ln d)$ term without dominating the overall computational cost.

---

> > ### Author Response · Authors · 2025-11-25
> >
> > > W2: While the authors introduce $\tilde{q}_t$ in Eq. 9 as an initial distribution for masked diffusion, people do not use such "partially masked and partially uniform" distributions for generative purposes. I think it comes from the time-homogeneous assumption of the forward diffusion noising, which is never used in the actual training of masked diffusion.
> >
> > A2: Thank you for your thoughtful review. We introduced the *partially masked and partially uniform* distribution (rather than a purely MASK-based Dirac measure) to avoid potential divergence when computing the KL term. From a technical perspective, this approach prevents blow-up in our analysis. Nonetheless, we have also shown that using a MASK-based Dirac measure is feasible by applying the chain rule of TV distance in conjunction with **Theorems 2** and **3**; see **Corollary 6 (Appendix D)** in our revision for the corresponding proof.
> >
> > > - [W3-1] In Table 1, the rate of Ren et al. [3] is different from the original rate in KL divergence. Similarly, it is written that the "higher-order" method gives $O(\epsilon^{-1})$, which is different from the original paper, again in KL divergence. I assume that the authors implicitly use Pinsker's inequality to deduce those bounds, but it is not a fair comparison/citation, since the original work do not aim to give a tight bound in terms of TV distance.
> >
> > A3: Thank you for your reminder. We indeed use Pinker's inequality to deduce the KL divergence in [3] to obtain its corresponding TV convergence in our Table. We have clarified this issue in the caption of Table 1 in our revision.
> >
> > > - [W3-2] In L46, the authors cite Nie et al. [4] to state "masked discrete diffusion has empirically outperformed uniform discrete diffusion" but the cited work does not mention uniform diffusion. I know masked diffusion models are more investigated, but there is also a uniform diffusion language model competitive with a masked model of similar scale [5].
> >
> > A3-2: Sorry for the confusion, we have revised the reference here from [4] to [6] in our latest version. In [6], authors provide experiments on various settings on text generation tasks where the abosorbing discrete diffusion consistantly shows better performance than that in uniform discrete diffusion.
> >
> > > - [W4-1] For example, in L53, the authors write "at least $O(\epsilon^{-1})$ steps" as if $\epsilon^{-1}$ would give a lower bound of the complexity. However, the notation only gives an upper bound and none of the cited work in L52 gives such a lower bound. The only lower bound I know in this context is the later half of [6, Theorem 1].
> >
> >
> > A4-1: Sorry for the confusion. We have removed the "at least" words in our revision to make this description more clear.
> >
> > > - [W4-2] Whether the parameters $d$ or $K$ are regarded as constants (i.e., whether they are inside $O(\cdot)$) frequently changes throughout the manuscript. In addition, why does $K$ disappear only in the second term in L94, compared with Eq. 20?
> >
> > A4-2: Sorry for the confusion on the complexity. In the latest version, we have aligned the complexity to our Table's presentation, where only regard $K$ as a constant. Such a presentation is also used in [2].
> >
> > > M1: The definition of should be $d$-(current value).
> >
> > AM1: Thank you for your reminder. We have revised them in our latest version.
> >
> > > M2: The value in Eq. 20 can be upper bounded by , which is $\epsilon$-free, while the sampling algorithm can depend on. Calling it "nearly" $\epsilon$-free is misleading.
> >
> > AM2: Thank you for highlighting this nuance. By “nearly $\epsilon$-free,” we mean that while the complexity does not explicitly scale with $\epsilon$ in a traditional sense, it is still influenced by the score estimation error. In particular, as the score estimation error decreases, the factor $d - \tfrac{\epsilon^2}{4}$ can grow, but the overall complexity remains $O(d)$. We appreciate your feedback on clarifying this point.
> >
> > ```
> > [1] Zheng K, Chen Y, Mao H, et al. Masked diffusion models are secretly time-agnostic masked models and exploit inaccurate categorical sampling[J]. arXiv preprint arXiv:2409.02908, 2024.
> >
> > [2] Liang Y, Huang R, Lai L, et al. Absorb and Converge: Provable Convergence Guarantee for Absorbing Discrete Diffusion Models[J]. arXiv preprint arXiv:2506.02318, 2025.
> >
> > [3] Ren Y, Chen H, Rotskoff G M, et al. How Discrete and Continuous Diffusion Meet: Comprehensive Analysis of Discrete Diffusion Models via a Stochastic Integral Framework[C]//The Thirteenth International Conference on Learning Representations.
> >
> > [4] Nie S, Zhu F, You Z, et al. Large Language Diffusion Models[J]. 2025.
> >
> > [6] Lou A, Meng C, Ermon S. Discrete diffusion modeling by estimating the ratios of the data distribution[C]//Proceedings of the 41st International Conference on Machine Learning. 2024: 32819-32848.
> >
> > [7] Ou J, Nie S, Xue K, et al. Your absorbing discrete diffusion secretly models the conditional distributions of clean data[J]. arXiv preprint arXiv:2406.03736, 2024.
> >
> > ```

---

> ### Comment · Reviewer_t4vz · 2025-11-25
>
> Thank you for the rebuttal. Let me reply to it in the following:
>
> **Regarding W1:**
>
> I acknowledge that the authors' remark
>
> > The number of masks is independent of time” holds only for an ideal reverse process, not necessarily for a learned one
>
> is true for a time-conditioned learned model. However, **it is an issue of the pretrained model, not the sampler's problem.** I would say:
> - If the model has severe assymetry of transition rates among different positions it affects the resulting $\epsilon_{score}$.
> - There might be cases where normalizing the score to adjust the total transition probability at each index slightly increases the error $\epsilon_{score}$, but I expect that the difference would be negligible, if the original $\epsilon_{score}$ is already very small (because of the first point).
> - It is reported in [1, Section J.2.2] that the model's ordering bias (in this case, produced by numerical error though) can severely affect the model's diversity/quality, so it is harmful if we keep that.
>
> Moreover, when the ideal reverse process is time-independent, why do we need to stick to time-dependent parameterization? Even in the author's additional experiment with SEDD modeling, later studies like MDLM [2] reveals that time-independent modeling actually is more effective under the same setting as SEDD-small on OpenWebText.
>
> For the "unbiasedness," I wonder why we need to simulate the score trajectory. We are basically given a random-order autoregressive model, and its error estimate also becomes trivial if we add up the conditional KL/TV with one-token unmasking at each step (by the chain rule of KL/TV).
>
> **Regarding the added experiment:**
>
> First, **MATU does not outperform other methods becasue it actually harms entropy.** Since the data entropy of OpenWebText is 7.7 [3, Table 2], "outperform" means the entropy is closer to 7.7. However, MATU just lowers Gen PPL and entropy at the same time, so it is just following a similar trade-off with the other methods. From my experience with masked diffusions on OpenWebText, the 0.1 difference in average entropy is somewhat significant (this said, the experiment is only with 32 samples, so both PPL and entropy can volatile when taking average over more samples).
>
> In addition, around L2395, MATU's inability to adjust for a smaller number of steps than sequential length is a crutial limitation to be noted. Since arguably one of the most important advantages of using discrete/masked diffusion over autoregressive models is its efficiency [4, Figure 1], a sampler which cannot enjoy the efficiency-quality tradeoff by reducing sampling steps is not very promising in practice. Also, the impact of the use of small $K$ (1 or 2) is not very clear to me, since it just violates the theory.
>
> References:
> - [1] Zheng et al. Masked diffusion models are secretly time-agnostic masked models and exploit inaccurate categorical sampling. ICLR 2025.
> - [2] Sahoo et al. Simple and effective masked diffusion language models. NeurIPS 2024.
> - [3] Gat et al. Discrete flow matching. NeurIPS 2024.
> - [4] Lou et al. Discrete diffusion modeling by estimating the ratios of the data distribution. ICML 2024.

---

> > ### Author Response · Authors · 2025-11-27
> >
> > Thank you for your quick response.
> >
> > > Q1: "The number of masks is independent of time holds only for an ideal reverse process, not necessarily for a learned one" is an issue of the pretrained model, not the sampler's problem.
> >
> > We appreciate your feedback and are pleased that we’ve reached a mutual understanding regarding the potential asymmetry of transition rates among different positions under time-dependent parameterizations. However, we seem to be talking about slightly different topics. Since [1] primarily focuses on the inference algorithm FHS, our earlier discussion also centered around FHS and inference analysis.
> >
> > From another perspective, your statement is correct: if a time-dependent model is trained sufficiently well or if we add hard constraints on transition rates across positions, then the resulting model would effectively align with a time-independent setting. In fact, we have already shown in Appendix  F that for models adhering to the time-independent assumptions, the convergence can be formally analyzed and guaranteed.
> >
> > > Q2: Why do we need to stick to time-dependent parameterization? Even in the author's additional experiment with SEDD modeling, later studies like MDLM [2] reveals that time-independent modeling actually is more effective under the same setting as SEDD-small on OpenWebText, and  why we need to simulate the score trajectory.
> >
> > A2: We appreciate the reviewer’s concern regarding time-dependent parameterization and the observation that time-independent modeling can sometimes be more effective. Our primary motivation for focusing on time-dependent models is to ensure **fair comparisons with prior theoretical works** (e.g., [2–4]). Additionally, we view the time-dependent framework as a more general toolkit for analyzing discrete diffusion: as demonstrated in Appendix F, it can **naturally reduce to the time-independent case** under suitable conditions.
> >
> > Nevertheless, not all absorbing forward processes—or the associated data distributions—map well onto a time-independent parameterization. In particular, the parameterization must remain time-dependent when:
> >
> > 1. The data distribution itself already includes absorbing states (e.g., hidden tokens in multi-turn dialogues or placeholders for sensitive content).
> > 2. Multiple classes of states are absorbed at different rates over time, as in more fine-grained forward process designs.
> >
> > Hence, time-dependent parameterization remains valuable not only for aligning with existing theoretical work but also for accommodating broader, more complex scenarios. Moreover, we note that generalization error can be analyzed by adding the conditional KL/TV divergence of partially unmasked tokens at each step. In Appendix F, we show how simulating the reverse process aim to bridge the time-dependent and time-independent settings, highlighting the flexibility of our proposed framework.
> >
> > > Q3: MATU does not outperform other methods becasue it actually harms entropy.
> >
> > A3: We appreciate your observation about the behaviors of perplexity (PPL) and entropy in our results. We monitor entropy mainly to ensure that the sampler does not collapse into producing only a narrow set of high-confidence outputs. However, given the limited capacity of our model at this scale, reducing PPL (i.e., improving generation quality) can naturally cause a slight decrease in entropy, as it filters out certain low-quality but potentially diverse samples.
> >
> > Regarding the observed 0.1 change in average entropy, we agree that it may be non-trivial in a small-sample setting of 32 examples. Yet, we wish to emphasize that this paper is fundamentally a theoretical contribution. Our experiments are primarily intended to validate the practical feasibility of a method that is directly derived from theory. While there may be gaps between theory and practice—as well as implementation nuances—we do not aim to displace existing approaches even some criteria have certain improvement, e.g., PPL, but rather to offer MATU as an alternative avenue that brings fresh insights both theoretically and practically.
> >
> > > Q4: MATU's inability to adjust for a smaller number of steps than sequential length is a crutial limitation to be noted.
> >
> > A4: In our revision part, we did **NOT** say "MATU's inability to adjust for a smaller number of steps than sequential length". In experiments, we only consider steps as $\{1024,2048\}$ since it matches our theoretical settings (one transition unmask one token). MATU can be esaily adjusted to the steps smaller than sequantial length, by generating more than one token in one discrte score calculation. Under that condition, [5] demonstrates the most inference sampling, including MATU, euler, tau-leaping, will fairly introduce another sampling error **from the mutual information** between the i-th token $y_i$ and and the rest of the sequence $y_{-i}$ and be presented as
> > $$
> > O\left(\frac{1}{T}\sum_{i=1}^d I(y_i, y_{-i})\right).
> > $$

---

> > > ### Author Response · Authors · 2025-11-27
> > >
> > > > Q5: Also, the impact of the use of small $K$ (1 or 2) is not very clear to me, since it just violates the theory.
> > >
> > > A5: Thank you for raising this concern. In the following setup:
> > > $$\sum_{y^\prime\not=y}R^\gets_t(y^\prime,y)\le \beta_{t}(y) = C \cdot \mathrm{numK}(\hat{y}_{t_{w-1}}) / (e^{\tilde{\sigma}(T-t_w)}-1),$$ using a small $C$ does not violate the theory; rather, it tightens the bounds based on practical data distribution. The theoretical setting $C=K$ (vocab size) provides a conservative worst-case upper bound for the exit rate. However, in practice, the actual reverse transition rate is far smaller than the theoretical upper bound derived from our Lemma 3. Consequently, setting $C \in \{1, 2\}$ is sufficient to cover the actual transition rate and satisfy the rejection threshold requirement. Empirically, we observe that this setting successfully maintains the bounding condition. As shown in the SEDD-small experiments below (with $C=1$), most ratios $\beta_t/\tilde{R}_t$ hovers at or above 1, indicating the required inequality holds:
> > > |     t_curr         | $\beta_t/\tilde{R}_t$ (>1 matches theory) |
> > > |:--------------------|:-----:|
> > > | $0.002$               |  $1.005$ |
> > > | $0.100$ |  $0.993$ |
> > > | $0.300$               |  $1.000$ |
> > > | $0.500$                |  $1.033$ |
> > > | $0.700$ | $1.063$ |
> > > | $0.900$ | $1.121$|
> > >
> > >
> > > ```
> > > [1] Zheng K, Chen Y, Mao H, et al. Masked diffusion models are secretly time-agnostic masked models and exploit inaccurate categorical sampling[J]. arXiv preprint arXiv:2409.02908, 2024.
> > >
> > > [2] Liang Y, Huang R, Lai L, et al. Absorb and Converge: Provable Convergence Guarantee for Absorbing Discrete Diffusion Models[J]. arXiv preprint arXiv:2506.02318, 2025.
> > >
> > > [3] Zhang Z, Chen Z, Gu Q. Convergence of score-based discrete diffusion models: A discrete-time analysis[J]. arXiv preprint arXiv:2410.02321, 2024.
> > >
> > > [4] Chen H, Ying L. Convergence analysis of discrete diffusion model: Exact implementation through uniformization[J]. arXiv preprint arXiv:2402.08095, 2024.
> > >
> > > [5] Li G, Jiao Y. Provable Efficiency of Guidance in Diffusion Models for General Data Distribution[J]. arXiv preprint arXiv:2505.01382, 2025.
> > > ```

---

> ### Comment · Reviewer_t4vz · 2025-11-28
>
> Thank you for the response. I apologize for my misunderstanding of Q4, and appreciate the additional experiment in Q5 that addressed my concern. However, the points Q1-Q3 still stand. Please see the follow-up comments.
>
> **Q1:** I was basically talking about the author's following claim:
> > the inference error caused by FHS [1] is beyond the score estimation error for mask diffusions with time-dependent paramterized discrete scores
>
> I believe the possible asymmetry of the model is raised by the authors as an evidence for this claim, but based on my comment (that asymmetry is actually included in $\epsilon_{score}$), I disagree with the claim.
>
> **Q2:** I understand that the authors approach the problem similarly with the previous work, but my framing is slightly different. While the references [3, 4] discuss general discrete diffusion models, [2] focuses solely on masked diffusion models with a predetermined sampler (uniformization and tau-leaping). Unlike [2], this work under our review **proposes** a novel sampler and discusses its complexity. That is why I am asking for a practical value of the proposed sampler. If the theoretical guarantee of the proposed sampler in a practical case is already achieved by another sampler, I am not sure if the contribution is significant enough for acceptance.
>
> Also, I agree with the authors that there might be situations with non-symmetric transition rate. However, as long as the forward masking is conducted independently about each index, we can decouple the selection of where to unmask from the token sampling of that index in the denoising process (as in FHS), since each position's unmasking rate only depends on the 0-1 infomation of the token (masked or unmasked), and is independent from tokens in the other positions. The position selection part is (or should) no longer related to the model, so we can just precompute or simulate the distribution of positional selection (which is not uniform unlike the usual case) as a straightforward generalization of FHS.
>
> For the data with missing value, I am not sure if people want to treat such datapoints with missing values as intial states for the diffusion; I would treat them as an intermediate state. Also, when we want to output things with masks, then the "mask" itself has a meaning, so I would assume this "mask" should be treated as a token different from the absorbing state?
>
> **Q3:** My main concern was that the red coloring of Entropy in the revised paper and the following comment to Reviewer QNxW is quite misleading.
> > As shown below, MATU consistently outperforms standard Euler and Tweedie-leaping methods for both perplexity (PPL) and entropy under different step settings (1024 and 2048).

---

> > ### Author Response · Authors · 2025-12-03
> >
> > Thank you for your response.
> >
> > > Q1: I disagree with the claim: "the inference error caused by FHS [1] is beyond the score estimation error for mask diffusions with time-dependent paramterized discrete scores", since the that asymmetry is actually included in score estimation error.
> >
> > A1: We appreciate the opportunity to clarify this crucial distinction. We respectfully maintain that FHS introduces a sampling discrepancy that is mathematically distinct from the score estimation error in time-dependent settings.
> >
> > To formalize this, we analyze the TV distance between the practical reverse process ($q^\gets_{\text{prac}, T}$) and the ideal reverse process governed by the ground truth ($q^\gets_{\text{ideal}, T}$). By introducing an auxiliary process $q^\gets_{\text{learn}, T}$ (the exact process defined by the learned score), we can decompose the error using the triangle inequality:$$\mathrm{TV}(q^\gets_{\text{prac}, T}, q^\gets_{\text{ideal}, T}) \le \underbrace{\mathrm{TV}(q^\gets_{\text{learn}, T}, q^\gets_{\text{ideal}, T})}\_{\text{Term 1: Score Estimation Error}} + \underbrace{\mathrm{TV}(q^\gets_{\text{prac}, T}, q^\gets_{\text{learn}, T})}\_{\text{Term 2: Sampler Error}}$$
> >
> > Here, $\{q^\gets_{\text{learn}, t}\}$ represents the CTMC with transition rates $R_\theta$ defined using the learned score. This decomposition strategy is standard in diffusion theory [2, 3, 4, 5]. Our core argument is that Term 2 is zero for our proposed MATU, but non-zero for FHS.
> > 1. Term 1 (Score Estimation Error): This term measures the gap between the process defined by the learned network and the ground truth process. It depends entirely on the training objective and network capacity. For a fixed, pretrained discrete diffusion model, Term 1 is identical regardless of whether we use MATU or FHS. Thus, this term cannot explain the performance gap between samplers.
> > 2. Term 2 (Sampler Error/Bias): the second term measures the gap between the practical reverse process ($q^\gets_{\text{prac}, T}$) and the learned reverse process ($q^\gets_{\text{learn}, T}$). We focus on the conditional TV distance at an infinitesimal time step $\Delta t$,  when FHS applies one of the chosen positions to denoise: $$\text{condTV}_{\text{FHS}}\colon=\mathrm{TV}\left(\mathrm{Pr}\_{\text{FHS}}(\mathbf{y}\_{t+\Delta t}||\mathbf{y}\_{t}=y),\mathrm{Pr}\_{\text{learned}}(\mathbf{y}\_{t+\Delta t}||\mathbf{y}\_{t}=y)\right).$$
> >
> > For FHS, it will have $\text{condTV}\_{\text{FHS}}>0$, because it has
> > $$
> > \mathrm{Pr}\_{\text{FHS}}(\mathbf{y}\_{t+\Delta t}=y^\prime||\mathbf{y}_{t}=y)
> > $$
> >
> > $$
> > = \mathrm{Pr}\_{\text{FHS}}(\text{Denoise Position}=i||\mathbf{y}_{t}=y)\cdot
> > \mathrm{Pr}\_{\text{FHS}}(y[i:K\rightarrow y^\prime_i]||\text{Denoise Position}=i)
> > $$
> >
> > $$
> > =  \frac{\tilde{R}\_{\theta,t}(y^\prime, y)\cdot \Delta t}{ \mathrm{numK}(y)\cdot \sum_{\tilde{y}_i\not=K}\tilde{R}\_{\theta,t}(y[i:K\rightarrow\tilde{y}_i], y)}
> > $$
> >
> > and
> >
> > $$
> > \mathrm{Pr}\_{\text{learned}}(\mathbf{y}\_{t+\Delta t}=y^\prime||\mathbf{y}\_{t}=y) = \frac{\tilde{R}\_{\theta,t}(y^\prime, y)\cdot \Delta t}{  \sum_{\tilde{y}_i\not=K}\tilde{R}\_{\theta,t}(y[i:K\rightarrow\tilde{y}_i], y)} \cdot \frac{ \sum\_{\tilde{y}_i\not=K}\tilde{R}\_{\theta,t}(y[i:K\rightarrow\tilde{y}_i], y)}{\tilde{R}_t(y)}.
> > $$
> >
> > Under this condition,
> > $$
> > \text{condTV}\_{\text{FHS}}= \underbrace{\sum_{y^\prime}\frac{\tilde{R}\_{\theta,t}(y^\prime, y)\cdot \Delta t}{  \sum_{\tilde{y}\_i\not=K}\tilde{R}\_{\theta,t}(y[i:K\rightarrow\tilde{y}\_i], y)}}\_{\le \Delta t}\cdot \left|\frac{ \sum_{\tilde{y}\_i\not=K}\tilde{R}\_{\theta,t}(y[i:K\rightarrow\tilde{y}\_i], y)}{\tilde{R}_t(y)} - \frac{1}{\mathrm{numK}(y)}\right|,
> > $$
> >
> > It is evident that the right-hand side remains positive unless
> > $\sum\_{\tilde{y}i\not=K}\tilde{R}\_{\theta,t}(y[i:K\rightarrow\tilde{y}\_i], y)$
> > is uniform across all positions $i$—a scenario that is highly unlikely when the parameterization depends on time. In contrast, if we apply a similar approach to bound $\text{condiTV}_{\text{MATU}}$  that upper bound can be driven to exactly 0. Hence, these sampler biases in FHS introduce an inference error beyond the usual score estimation error.
> >
> > Moreover, **no existing theorems or lemmas (to our knowledge) guarantee that FHS’s practical inference procedure converges to the true data distribution in these time-dependent discrete settings**. Therefore, our statement that FHS introduces additional inference error is grounded in the observation that Term 2 (sampler bias) arises solely from FHS’s partial-update procedure and remains present unless the transition rates are exactly uniform—which is uncommon in time-dependent discrete diffusions.
> >
> > We hope this elaboration clarifies where we see FHS diverging from its own learned score model, thus producing inference errors beyond those stemming solely from imperfect score estimation.

---

> > > ### Author Response · Authors · 2025-12-03
> > >
> > > > Q2: If the theoretical guarantee of the proposed sampler in a practical case is already achieved by another sampler, I am not sure if the contribution is significant enough for acceptance.
> > >
> > > A2: We appreciate your feedback. However, **as noted in our title and abstract, our primary aim is to improve the theoretical understanding** of masked diffusion, and the new sampler a direct derivation from our theoretical analysis. Our key contribution is achieving $\epsilon$-free complexity for the time-dependent discrete diffusion model inference, which has never been proven before.
> > >
> > > Beyond theory, supplementary experiments were conducted under different settings (synthetic, real world) to validate the effectiveness of mask-aware truncation. To our knowledge, this is also the first time that a uniformity-type sampler has successfully run on a real-world text generation setting and achieved a PPL better than typical samplers such as Euler or Tweedie $\tau$--leaping. While prior work [6] suggests that uniformization suffers from NFE blow-up (Fig. 1, left, in [6]), our truncation approach (which is originally designed to remove the score-boundedness assumption in analysis) effectively circumvents this issue.
> > >
> > >
> > >
> > > > Q3: My main concern was that the red coloring of Entropy in the revised paper and the following comment to Reviewer QNxW is quite misleading.
> > >
> > > A3: Sorry for the confusion, we have revised the latest version and update our reply to Reviewer QNxW.
> > >
> > >
> > > ```
> > > [1] Zheng K, Chen Y, Mao H, et al. Masked diffusion models are secretly time-agnostic masked models and exploit inaccurate categorical sampling[J]. arXiv preprint arXiv:2409.02908, 2024.
> > >
> > > [2] Chen, S., Chewi, S., Li, J., Li, Y., Salim, A., & Zhang, A. R. (2022). Sampling is as easy as learning the score: theory for diffusion models with minimal data assumptions. arXiv preprint arXiv:2209.11215.
> > >
> > > [3] Zhang, Z., Chen, Z., & Gu, Q. (2024). Convergence of score-based discrete diffusion models: A discrete-time analysis. arXiv preprint arXiv:2410.02321.
> > >
> > > [4] Chen, H., & Ying, L. (2024). Convergence analysis of discrete diffusion model: Exact implementation through uniformization. arXiv preprint arXiv:2402.08095.
> > >
> > > [5] Liang, Y., Huang, R., Lai, L., Shroff, N., & Liang, Y. (2025). Absorb and Converge: Provable Convergence Guarantee for Absorbing Discrete Diffusion Models. arXiv preprint arXiv:2506.02318.
> > >
> > > [6] Ren Y, Chen H, Zhu Y, et al. Fast solvers for discrete diffusion models: Theory and applications of high-order algorithms[J]. arXiv preprint arXiv:2502.00234, 2025.
> > > ```

---

### Official Review · Reviewer_QNxW · 2025-11-01

**Soundness:** 2
**Presentation:** 2
**Contribution:** 1
**Rating:** 2
**Confidence:** 5

**Summary:**

This paper presents a rigorous theoretical analysis of the computational complexity of masked discrete diffusion models for text generation. The authors make two primary contributions. First, they provide the first formal complexity analysis for the widely used Euler sampler in this context, showing that it requires $O(d^2\epsilon^{-3/2})$ score evaluations to achieve $\epsilon$-accuracy in total variation (TV) distance. Second, they introduce a sampling method called Mask-Aware Truncated Uniformization (MATU). The key advantage of MATU is that it removes the restrictive bounded-score assumptions required by prior theoretical work and leverages the property that tokens are unmasked at most once. The authors prove that MATU achieves a nearly $\epsilon$-free complexity of $O(d\ln d)$ that surpasses existing methods for both masked and uniform discrete diffusion.

**Strengths:**

The paper is properly written, with formal assumptions, lemmas, and theorems. The authors try to present rigorous theoretical analyses, which is appreciated.

**Weaknesses:**

However, given the following reasons, I believe the paper has major oversights on previous work and makes little contribution to the dLLM community.
- Missing discussion of important related work. The uniformization discussed in the paper is just an approximated version of the "first-hitting sampler" in [1]. [1] already proves the equivalence of MDMs to masked models in [2][3][4], suggesting that the time $t$ in MDMs can be fully removed, and the sampling of MDMs is **exact** by $d$-step token-by-token decoding. Given this fact, all the discussions on complexity in this paper appear unnecessary and outdated to me because [1] has proposed the ultimate solution to MDMs. The paper did not even mention [1], and did not discuss the relationship between [1] and the proposed MATU.
- Lack of experiments. The proposed algorithm and complexity analyses are not verified by experiments, even on the most basic setting of SEED/MDLM, which measures generative perplexity. Not to mention the applicability to more practical dLLMs.
- Useless for application. As far as I know, practical dLLMs like LLaDA already remove time $t$ in the network, and adopt the MaskGIT-style token-by-token decoding, even with more heuristic confidence-based decoding orders. Therefore, all the analyses concerning $t$ or Euler sampler in this paper are completely useless to practical dLLMs.

[1] Masked Diffusion Models are Secretly Time-Agnostic Masked Models and Exploit Inaccurate Categorical Sampling (ICLR 2025)

[2] BERT: Pre-training of Deep Bidirectional Transformers for Language Understanding (2018)

[3] MaskGIT: Masked Generative Image Transformer (2022)

[4] Mask-Predict: Parallel Decoding of Conditional Masked Language Models (2019)

**Questions:**

- How do your findings reconcile with [1], which demonstrates that Masked Diffusion Models are theoretically equivalent to time-agnostic masked models? Does your complexity analysis, which is tied to the continuous-time diffusion process, still provide practical motivation for using the diffusion framework over simpler, equivalent masked model formulations?
- [1] identified a critical numerical precision issue in Gumbel-based sampling that led to unfairly optimistic evaluations of MDMs. Does your theoretical analysis account for such numerical instabilities? More practically, how does your proposed MATU sampler perform empirically when benchmarked using 64-bit precision, which has been shown to be necessary for fair evaluation?
- Could you provide an empirical comparison of MATU against the Euler sampler and the "first-hitting sampler" from [1]? A comparison of wall-clock time, generative perplexity, and sentence entropy on a standard text dataset would be crucial to validate that the theoretical gains in complexity translate to empirical benefits.

---

> ### Author Response · Authors · 2025-11-25
>
> We thank the reviewer for carefully reading our work and providing detailed feedback, and try to solve your concerns point-by-point in the following:
>
> > W1: Missing discussion of important related work. The uniformization discussed in the paper is just an approximated version of the "first-hitting sampler" in [1].[1] already proves the equivalence of MDMs to masked models in [2][3][4], suggesting that the time $t$ in MDMs can be fully removed, and the sampling of MDMs is exact by $d$-step token-by-token decoding.
> Given this fact, all the discussions on complexity in this paper appear unnecessary and outdated to me because [1] has proposed the ultimate solution to MDMs. The paper did not even mention [1], and did not discuss the relationship between [1] and the proposed MATU.
>
> A1: Thank you for calling our attention to the need for a more thorough discussion regarding [1] and its “first-hitting sampler” (FHS). We have revised our manuscript to include explicit references to [1]–[4] in **Section 6** and further clarify how our proposed **MATU** relates to FHS. Below, we summarize the main distinctions:
>
> The first point is **Time-Dependent vs. Time-Independent Parameterizations**:
>
> In **Time-Dependent Parameterization,** the discrete score depends on $t$ (e.g., $s_{\theta,t,y}(y^\prime) \approx q^\to_t(y^\prime)/q^\to_t(y)$ in [6]), **FHS does not exactly simulate** the *practical/learned* reverse process. Specifically, FHS selects the next position to unmask **uniformly** from the remaining masked tokens, even though different positions may have different reverse transition rates in the learned model, i.e., $$ \sum_{y^\prime, s.t.\ \mathrm{Ham}(y^\prime, y)=1, \mathrm{DffIdx}(y^\prime,y)=i, y_{i}=K} s_{\theta,t,y}(y^\prime)\not = \sum_{y^\prime, s.t.\ \mathrm{Ham}(y^\prime, y)=1, \mathrm{DffIdx}(y^\prime,y)=j, y_{j}=K} s_{\theta,t,y}(y^\prime) ,$$ $$\Rightarrow \sum_{y^\prime, s.t.\ \mathrm{Ham}(y^\prime, y)=1, \mathrm{DffIdx}(y^\prime,y)=i, y_{i}=K} \underbrace{\lim_{\Delta t\rightarrow 0} \Bigl[\frac{\hat{q}\_{t+\Delta t \mid t}(y^\prime \mid y) - \delta_{y}(y^\prime)}{\Delta t}\Bigr]}_{=\hat{R}(y^\prime,y)}$$
>
> $$\not = \sum_{y^\prime, s.t.\ \mathrm{Ham}(y^\prime, y)=1, \mathrm{DffIdx}(y^\prime,y)=j, y_{j}=K} \lim_{\Delta t\rightarrow 0} \Bigl[\frac{\hat{q}\_{t+\Delta t \mid t}(y^\prime \mid y) - \delta_{y}(y^\prime)}{\Delta t}\Bigr].$$
> While the “number of masks is independent of time” property holds for an *ideal* reverse process, it may not hold for the learned version—there is no guarantee that each position’s transition rate is identical. Hence, the overall inference error in FHS [1] is not limited solely to score-estimation error, and the final deviation from the true data distribution is unquantified. By contrast, **MATU** precisely simulates the learned reverse process: in Step&nbsp;10 of **Algorithm&nbsp;1**, we incorporate $s_{\theta,t,y}(y')$ directly, ensuring position-dependent transition rates are fully respected.
>
> In **Time-Independent Parameterization,** the discrete score is time-independent [7], FHS *does* exactly simulate the learned reverse process in a $d$-step manner. This is due to the strong constraint
> $$
> \sum_{y^\prime, s.t.\ \mathrm{Ham}(y^\prime, y)=1, \mathrm{DffIdx}(y^\prime,y)=i, y_{i}=K} p_{\theta,y} (y^\prime) = 1 = \sum q_0(y^\prime_i||y^{UM})
> $$
> for each position, which guarantees equal transition rates across all masked indices. **However, [1] does not provide a detailed and rigorous proof of this unbiasedness**, in our revision we fill this gap with **Theorem 7 (Appendix F)**, where we couple FHS’s trajectory to MATU’s dynamics. Moreover, because the time-independent transition matrix remains unchanged as long as the particle $\hat{z}_t$ remains unchanged, **MATU** also requires only $d$ updates in this setting—matching FHS. Rejection steps introduce an additional $O(d \ln d)$ factor, but do not dominate the computational cost.
>
> As for the **Relationship to FHS and Our Contributions,**  We appreciate that [1] offers an *ultimate* solution for masked diffusion when the discrete score is *time-independent*, and we have updated our paper to cite and discuss their approach. However, FHS [1] lacks a rigorous justification in the *time-dependent* case and does not quantify the final inference error. Our **MATU** serves to bridge this gap by providing a comprehensive theoretical framework and explicit handling of position-dependent transition rates in the reverse process.
>
> In summary, we recognize the importance of [1]’s FHS and have now incorporated a thorough comparison in the revised version. We believe our results meaningfully extend the analysis to the time-dependent setting and supply rigor where prior work has been informal.

---

> ### Author Response · Authors · 2025-11-25
>
> > W2: Lack of experiments. The proposed algorithm and complexity analyses are not verified by experiments, even on the most basic setting of SEED/MDLM, which measures generative perplexity. Not to mention the applicability to more practical dLLMs.
>
> **A2:** We appreciate your concerns regarding experimental validation. In our revision, we have incorporated both real-world and synthetic experiments to demonstrate the practicality and effectiveness of **MATU**. Specifically, on the **SEDD-small** setting—featuring an absorbing stationary distribution—we adapted MATU into an inexact version. As shown below, MATU consistently outperforms standard Euler and Tweedie $\tau$-leaping methods for perplexity (PPL) with slightly entropy decrease under different step settings (1024 and 2048).
>
> |     Sampler         | Steps | Avg PPL | Avg Entropy | Wall-Clock Time |
> |:--------------------|:-----:|--------:|------------:| -----:|
> | Euler               |  1024 |   41.42 |       7.588 | 27.35s/sample |
> | Tweedie $\tau$-leaping |  1024 |   41.81 |       7.597 | 24.15s/sample |
> | MATU                |  1024 |   40.54 |       7.554 | 32.23s/sample |
> | Euler               |  2048 |   33.32 |       7.492 | 53.43s/sample |
> | Tweedie $\tau$-leaping |  2048 |   32.50 |       7.489 | 46.88s/sample |
> | MATU                |  2048 |   31.82 |       7.394 | 60.05s/sample |
>
> Please refer to **Appendix&nbsp;G.2** of our revision for additional experimental details and analysis.
>
> > W3: Useless for application. As far as I know, practical dLLMs like LLaDA already remove time $t$ in the network, and adopt the MaskGIT-style token-by-token decoding, even with more heuristic confidence-based decodin
> g orders. Therefore, all the analyses concerning or Euler sampler in this paper are completely useless to practical dLLMs.
>
> A3: Our primary goal is to provide a rigorous theoretical framework that deepens our understanding of discrete-state diffusion processes, regardless of whether they explicitly incorporate time-dependent parameters in practice.
>
> 1. **General Theoretical Insights.** While modern dLLMs often adopt time-independent (or “MaskGIT-style”) decoding strategies, a comprehensive theoretical treatment of both time-dependent and time-independent cases is beneficial. It not only clarifies the underlying assumptions in each setting but also provides a more solid foundation for future algorithmic developments, particularly as new architectures or sampling methods emerge.
>
> 2. **Extension to Time-Independent Methods.** Much of our theoretical analysis naturally extends to time-independent parameterizations, which are indeed common in large-scale dLLMs. Providing concrete proofs for both time-dependent and time-independent cases ensures broader applicability and enriches our theoretical toolbox.
>
> 3. **Euler Sampler as a Baseline.** We use the Euler sampler primarily as a baseline for comparison and as a vehicle to illustrate key principles. Even though it may not be the sampler of choice in all practical scenarios, demonstrating how our framework handles this well-studied sampler is valuable for validating the correctness and robustness of our approach.
>
> In summary, our main contribution lies in offering a rigorous theoretical understanding of discrete diffusion modeling. This allows us to demonstrate the reliability of our framework and potentially guide future adaptations for more specialized or heuristic decoding strategies—even those that do not explicitly involve time-dependent parameters.
>
> > Q1: How do your findings reconcile with [1], which demonstrates that Masked Diffusion Models are theoretically equivalent to time-agnostic masked models? Does your complexity analysis, which is tied to the continuous-time diffusion process, still provide practical motivation for using the diffusion framework over simpler, equivalent masked model formulations?
>
> A1: We introduce **Theorem 7 (Appendix F)** to show that, for time-independent parameterizations, the **first-hitting sampler (FHS)** from [1] does in fact precisely replicate the learned reverse process, by coupling FHS trajectories with those of **MATU**.
>
> More importantly, a notable insight from our **MATU** study is that **uniformly selecting the next denoising position** is what ensures unbiased reverse-process simulation under time-independent parameterizations (“simpler, equivalent masked model formulations”). Concretely, the strong constraints on the discrete scores,
> $$
> \sum_{y^\prime, s.t.\ \mathrm{Ham}(y^\prime, y)=1, \mathrm{DffIdx}(y^\prime,y)=i, y_{i}=K} p_{\theta,y} (y^\prime) = 1 = \sum q_0(y^\prime_i||y^{UM})
> $$
> imply that each position is subject to the **same** reverse transition rate. In mathematical terms,

---

> > ### Author Response · Authors · 2025-11-25
> >
> > $$
> > \sum_{y^\prime, s.t.\ \mathrm{Ham}(y^\prime, y)=1, \mathrm{DffIdx}(y^\prime,y)=i, y_{i}=K}\underbrace{\lim_{\Delta t\rightarrow 0} \Bigl[\frac{\hat{q}\_{t+\Delta t \mid t}(y^\prime \mid y) - \delta_{y}(y^\prime)}{\Delta t}\Bigr]}_{=\hat{R}(y^\prime,y) = C(t)\cdot q_0(y^\prime_i||y^{UM})}
> > $$
> >
> > $$
> > = \sum_{y^\prime, s.t.\ \mathrm{Ham}(y^\prime, y)=1, \mathrm{DffIdx}(y^\prime,y)=j, y_{j}=K} \lim_{\Delta t\rightarrow 0} \Bigl[\frac{\hat{q}\_{t+\Delta t \mid t}(y^\prime \mid y) - \delta_{y}(y^\prime)}{\Delta t}\Bigr] = C(t)
> > $$
> >
> > so that choosing the next unmasked token **uniformly** is an unbiased strategy. Although [1] introduces FHS, it does not delve into why this uniform selection holds in the time-independent case, which our theoretical results clarify.
> >
> > > Q2: [1] identified a critical numerical precision issue in Gumbel-based sampling that led to unfairly optimistic evaluations of MDMs. Does your theoretical analysis account for such numerical instabilities? More practically, how does your proposed MATU sampler perform empirically when benchmarked using 64-bit precision, which has been shown to be necessary for fair evaluation?
> >
> > A2: Our theoretical only consider the convergence of the proposed algorithm and defalut all the calculations are precise, which is the basic setting in most theoretical work, e.g., [6] and [7],  except for the works focusing the analysis on the quantized algorithm. Therefore, the numerical instablities is beyond the scope of this paper.
> >
> > > Q3: Could you provide an empirical comparison of MATU against the Euler sampler and the "first-hitting sampler" from [1]? A comparison of wall-clock time, generative perplexity, and sentence entropy on a standard text dataset would be crucial to validate that the theoretical gains in complexity translate to empirical benefits.
> >
> > A3: We kindly refer you to our response in W2, where we present experimental results on SEDD-small with an absorbing stationary distribution. Currently, we are unable to directly compare our method with the first-hitting sampler (FHS) from [1] due to the absence of publicly available implementations for FHS on both SEDD, and MDLM. Moreover, under time-independent parameterizations, our algorithm aligns with FHS except for the additional rejection-sampling mechanism.
> >
> > ```
> > [1]Masked Diffusion Models are Secretly Time-Agnostic Masked Models and Exploit Inaccurate Categorical Sampling (ICLR 2025)
> >
> > [2] BERT: Pre-training of Deep Bidirectional Transformers for Language Understanding (2018)
> >
> > [3] MaskGIT: Masked Generative Image Transformer (2022)
> >
> > [4] Mask-Predict: Parallel Decoding of Conditional Masked Language Models (2019)
> >
> > [5] Lou A, Meng C, Ermon S. Discrete diffusion modeling by estimating the ratios of the data distribution[C]//Proceedings of the 41st International Conference on Machine Learning. 2024: 32819-32848.
> >
> > [6] Ren Y, Chen H, Rotskoff G M, et al. How Discrete and Continuous Diffusion Meet: Comprehensive Analysis of Discrete Diffusion Models via a Stochastic Integral Framework[C]//The Thirteenth International Conference on Learning Representations.
> >
> > [7] Liang Y, Huang R, Lai L, et al. Absorb and Converge: Provable Convergence Guarantee for Absorbing Discrete Diffusion Models[J]. arXiv preprint arXiv:2506.02318, 2025.
> > ```

---

### Official Review · Reviewer_SEZr · 2025-11-01

**Soundness:** 3
**Presentation:** 3
**Contribution:** 3
**Rating:** 8
**Confidence:** 2

**Summary:**

This paper presents a theoretical analysis of masked discrete diffusion (MDD), deriving convergence bounds for the Euler sampler and proposing a Mask-Aware Truncated Uniformization (MATU) method with nearly ε-free complexity. The work aims to explain why masked diffusion achieves faster convergence than uniform discrete diffusion.

**Strengths:**

* The paper establishes a clear theoretical foundation for **masked** discrete diffusion, a setting widely used in discrete and language diffusion models but previously lacking formal complexity analysis.
* Provides the first rigorous proof for the Euler sampler in discrete diffusion, bridging empirical implementations and theoretical understanding.
* They offer a near $\epsilon$-free convergence guarantee, with MATU algorithm representing a genuine improvement over prior uniformization approaches.

**Weaknesses:**

* **Presentation: missing intuitive root for the difference.**
  As a reader, I wanted to clearly see *what makes the $\tau$-leaping algorithm different from Euler sampling*, and *how that difference theoretically leads to different error bounds.*
  Currently, the paper concludes that Euler achieves a better complexity rate, but the *intuitive mechanism* behind this improvement remains unclear.

* **(Minor) Lack of connection between theoretical and empirical behaviors.**
  While the paper focuses on theoretical convergence rates, it would help to include an explicit discussion on how these rates correspond to practical behavior observed in masked discrete diffusion models. Without that bridge, the theoretical results feel somewhat detached from the empirical phenomenon the paper aims to explain. A qualitative, Intuitive explanation would be enough.

**Questions:**

* What is the precise difference between $\tau$-leaping and Euler sampling algorithms?
  In the context of masked discrete diffusion, could the authors concretely explain how these two update rules differ, and *what drives the improvement* from $\tau$-leaping’s $O(d\epsilon^{-2})$ to Euler’s $\tilde{O}(d^2\epsilon^{-3/2})$ in Theorem 1?
  Is the improvement mainly due to variance reduction from removing Poisson randomness, or does it come from the independence assumption between token coordinates?

* In addition, [1] proposes **$\tau$-leaping–Tweedie**. How does that method differ from Euler sampling under the masked discrete diffusion setup?
  What complexity rate would you expect if τ-leaping–Tweedie were analyzed under your theoretical framework?

* This may be tangential, but what is the effective **complexity of autoregressive models** in comparison?
  Conceptually, how does that relate to the $O(d \ln d)$ result of Theorem 2?

[1] *Discrete Diffusion Modeling by Estimating the Ratios of the Data Distribution*, [https://arxiv.org/abs/2310.16834](https://arxiv.org/abs/2310.16834)

---

> ### Author Response · Authors · 2025-11-25
>
> We thank the reviewer for carefully reading our work and providing positive and insightful feedback, and we try to solve your concerns point-by-point in the following:
>
> > W1 & Q1: As a reader, I wanted to clearly see what makes the  $\tau$-leaping algorithm different from Euler sampling, and how that difference theoretically leads to different error bounds. Currently, the paper concludes that Euler achieves a better complexity rate, but the intuitive mechanism behind this improvement remains unclear.
>
> A1: Their differences can be summarized as the following two aspects.
>
> First, **Algorithmic Difference (Intuition)**: $\tau$-leaping: Intuitively, to make a leap, this method assumes the rate $R_t$ and state $x_t$ remain constant over $[t_0, t_0+\tau]$. It "batches" transitions: it counts all transitions that would occur under a constant transition rate (following a Poisson distribution) and applies them simultaneously until it reaches $t_0+\tau$. The update is:
> $$x_{t_0+\tau}=x_{t_0}+\sum_{y}N(y)(y-x_{t_0}), \quad \text{where } N(y)\sim \text{Poisson}(\tau R_{t_0}(x_{t_0},y)).$$
> Since $R_t(x_t,y)=0$ for $\textrm{Ham}(x_{t_0},y)>1$, the update can be written as $x_{t_0+\tau}=x_{t_0}+\sum_{i=1}^d \sum_{k=1}^K (k-x_{t_0}^i)N_{\textrm{Poisson}}(x_{t_0},x_{t_0}+(k-x_{t_0}^i)e_i)(k-x_{t_0}^i)e_i$, where $e_i$ is the one-hot vector.
>
> Euler Sampling: This method uses a first-order Taylor expansion ($e^{hR}\approx I+hR$) to approximate the probability transition matrix. Unlike $\tau$-leaping, it allows for at most one transition per step (or staying at the same state). The update is :
>
> $$x_{t+h}=y \ \text{ w.p. } hR_t({x_t,y}) \ \text{ (for } x_{t}\neq y); \quad x_{t+h}=x_t \ \text{ otherwise.}$$
>
> Second, **Theoretical Error Bounds**: Crucially, we do not claim that Euler inherently achieves a better complexity rate than $\tau$-leaping; rather, we demonstrate that the $\epsilon$ dependence in the complexity rate of the Euler method derived via our proof technique is better than the existing bounds for $\tau$-leaping. However, this comes with a trade-off: our analysis yields a dimension dependence of $O(d^2)$, which is worse than standard $\tau$-leaping results.
>
> The difference stems from the analysis technique. Existing $\tau$-leaping bounds often rely on Girsanov's theorem. In contrast, to bound the error to the target distribution $TV(p_{\text{Euler}}, p^\*)$, we utilize the exponential integrator $p_{\text{Exp-Int}}$ with transition matrix $e^{hR_{t_k}}$ as an auxiliary process. We bound $TV(p_{\text{Euler}}, p_{\text{Exp-Int}})$ using higher-order Taylor terms and bound $TV(p_{\text{Exp-Int}}, p^*)$ by applying the chain rule of KL divergence (converting to TV via Pinsker’s inequality). This yields a tighter analysis for Euler compared to standard $\tau$-leaping results. Thanks again for the question, we will revise the presentation to avoid any confusion.
>
> > W2: (Minor) Lack of connection between theoretical and empirical behaviors.
>
> A2: We thank the reviewer for this suggestion. To bridge the gap between our theoretical rates and practical behavior, we have added a new "Synthetic Experiments" section in Appendix G.1. We empirically validate our convergence bounds in a controlled setting ($K=3, d=4$). As predicted by our theory, the results demonstrate that the Masked Discrete Diffusion model achieves significantly faster convergence to the target distribution (measured by TV distance) compared to the uniform-stationary baseline. Furthermore, we provide a qualitative comparison of sampling trajectories, illustrating how the theoretical efficiency gains translate into fewer and more decisive state updates (lower NFE) in practice. Please see Figure 1 for the convergence plots and Figure 2 for the trajectory visualizations.

---

> > ### Author Response · Authors · 2025-11-25
> >
> > > Q3: This may be tangential, but what is the effective complexity of autoregressive models in comparison? Conceptually, how does that relate to the $O(d\ln d)$ result of Theorem 2?
> >
> > A3: Thank you for raising this point. Autoregressive (AR) models typically operate at an effective complexity of $O(d)$, which indeed appears lower than the $O(d \ln d)$ requirement from Theorem 2. However, AR methods are constrained to a strictly left-to-right generation scheme—notably hindering any possibility of dropping below linear complexity. By contrast, discrete diffusion approaches can exploit parallel sampling techniques—drawing from MCMC [1], continuous diffusion [2], or higher-order solvers [3]—to potentially reduce or even remove the $d$-dependence in iteration complexity. We see these directions as highly promising for achieving further efficiency gains in discrete diffusion–based language models, and we appreciate your question for spotlighting this important area.
> >
> >
> >
> >
> > ```
> > [1] Anari N, Chewi S, Vuong T D. Fast parallel sampling under isoperimetry[C]//The Thirty Seventh Annual Conference on Learning Theory. PMLR, 2024: 161-185
> >
> > [2] Shih A, Belkhale S, Ermon S, et al. Parallel sampling of diffusion models[J]. Advances in Neural Information Processing Systems, 2023, 36: 4263-4276.
> >
> > [3] Ren Y, Chen H, Zhu Y, et al. Fast solvers for discrete diffusion models: Theory and applications of high-order algorithms[J]. arXiv preprint arXiv:2502.00234, 2025.
> > ```

---

### Official Review · Reviewer_1mpG · 2025-11-02

**Soundness:** 3
**Presentation:** 3
**Contribution:** 3
**Rating:** 6
**Confidence:** 4

**Summary:**

This paper studies the theoretical complexity of masked discrete diffusion models, i.e. discrete diffusion models based on absorbing/masking continuous-time Markov chains. Current studies on the mathematical theory of discrete diffusion models mainly focused the case of the uniform case, while this paper is one of the first studies on the masking/absorbing case. Compared to relatively few prior work on the theory of masked discrete diffusion models like [1], this paper provides the first theoretical analysis of the commonly used Euler sampler and a modified version of the uniformization-style sampler called Mask-Aware Truncated Uniformization (MATU) for the case of masked discrete diffusion models. Specifically for the latter uniformization-style sampler, the modification proposed in this paper is mainly motivated from the truncation technique adopted in [2], which makes full use of the masking property, removes the assumption on the boundedness of the learnt score function, and achieves a nearly $\epsilon$-free convergence rate with respect to the TV distance (Here $\epsilon$ denotes the score approximation error). Such theoretical results partially explain why the masking case surpasses the uniform case in practice under certain circumstances.

**Strengths:**

To the best of the reviewer's knowledge, there has been relatively few work studies the theoretical complexity of masked discrete diffusion model - probably [1] is the only related work. Hence, the reviewer thinks that this paper studies an important question with interesting results that improves the dependency on the score approximation error $\epsilon$. Also, the results are presented in a rigorous and detailed way (For instance, Table 1 clearly exhibits how the results presented in this work compared to existing work on the theory of uniform/masked discrete diffusion models).

**Weaknesses:**

(1) It seems to the reviewer that the assumption [A2] has never been adopted in existing work [1] on the theory of masked discrete diffusion models. Hence, the authors probably need to add a short paragraph to further comment on whether this is a strong assumption and how it impacts the proof strategy. Does it somehow relate to Assumption 4 in [1]?

(2) Even though the reviewer fully understands that this is a theoretical paper, the authors are encouraged to include a few small-scale experiments (maybe just one on a toy model) to illustrate how the theoretical studies might partially explain why masked discrete diffusion model surpasses the uniform discrete diffusion model under certain circumstances. This would greatly improve the quality of the manuscript.

(3) Moreover, it seems to the reviewer that the authors might be missing a few related work on the theory of discrete diffusion models. For instance, an incomplete of concurrent work include but not limit to [3,4,5]. Given that the theoretical studies of discrete diffusion models is a relatively new area in the machine learning theory community, including a complete list of related work can potentially help enhance the quality of the paper.

**Questions:**

Would it be possible for the authors to comment on whether the technique developed in this paper can be applied to analyze the Tweedie $\tau$-leaping proposed in [6] for the case of masked discrete diffusion models? Even some brief intuition on how complexity might change would be appreciated!

Overall, I think the paper might be considered for top ML venues like ICLR, but the authors should probably address all questions above, add papers listed below as extra references and discuss them appropriately.

References:

[1] Liang, Yuchen, Renxiang Huang, Lifeng Lai, Ness Shroff, and Yingbin Liang. "Absorb and Converge: Provable Convergence Guarantee for Absorbing Discrete Diffusion Models." arXiv preprint arXiv:2506.02318 (2025).

[2] Huang, Xunpeng, Yingyu Lin, Nikki Lijing Kuang, Hanze Dong, Difan Zou, Yian Ma, and Tong Zhang. "Almost Linear Convergence under Minimal Score Assumptions: Quantized Transition Diffusion." arXiv preprint arXiv:2505.21892 (2025).

[3] Srikanth, Aadithya, Mudit Gaur, and Vaneet Aggarwal. "Discrete State Diffusion Models: A Sample Complexity Perspective." arXiv preprint arXiv:2510.10854 (2025).

[4] Wan, Zhengyan, Yidong Ouyang, Qiang Yao, Liyan Xie, Fang Fang, Hongyuan Zha, and Guang Cheng. "Error Analysis of Discrete Flow with Generator Matching." arXiv preprint arXiv:2509.21906 (2025).

[5] Liang, Yuchen, Yingbin Liang, Lifeng Lai, and Ness Shroff. "Discrete Diffusion Models: Novel Analysis and New Sampler Guarantees." arXiv preprint arXiv:2509.16756 (2025).

[6] Lou, Aaron, Chenlin Meng, and Stefano Ermon. "Discrete diffusion modeling by estimating the ratios of the data distribution." arXiv preprint arXiv:2310.16834 (2023).

---

> ### Author Response · Authors · 2025-11-25
>
> We thank the reviewer for carefully reading our work and providing encouraging and insightful feedback, and try to solve your concerns point-by-point in the following:
>
> > W1:  the authors probably need to add a short paragraph to further comment on whether [A2] is a strong assumption and how it impacts the proof strategy. Does it somehow relate to Assumption 4 in [1]?
>
> A1: Thank you for pointing this. First, **For the strongness of [A2]**, We consider \([A2]\) a mild assumption, because the choice of absorbing state in the forward process is quite flexible. In practice, most studies choose a token (the MASK token) that never appears in the training data. Since the training data are drawn from the underlying distribution, which does not include the MASK token, it is reasonable to assume that the MASK token will almost never appear in the data distribution.
>
> **For the impact of [A2] in proof**, We introduced \([A2]\) to reflect real-world practice and to simplify complexity analysis. However, we have removed \([A2]\) to obtain a more general theorem, now presented in **Corollary 5 (Appendix&nbsp;D)** of our revision.
> Intuitively, **Theorem 4** shows that inference complexity mainly depends on the expected number of MASK tokens in the forward random variable, $\mathbb{E}[\mathrm{numK}(y^\to_t)]$. If we allow the MASK token to appear in the data distribution, the expected count can be computed by
> $$
> \mathrm{Pr}[\mathrm{numK}(y^\to_t) = k] = \sum_{i=0}^k \mathrm{Pr}[\mathrm{numK}(y^\to_t) = k|\mathrm{numK}(y^\to_0) = i]\cdot \mathrm{Pr}[\mathrm{numK}(y^\to_0) = i].
> $$
> For the conditional probability, *rather than assuming no MASK token in data distribution, we needs to calculate the case when $i\not=0$,* then $$ \mathrm{Pr}[\mathrm{numK}(y^\to_t) = k|\mathrm{numK}(y^\to_0) = i] = \underbrace{C_{d-i}^{k-i}}\_{\text{unmask}\\to \text{maske count}} \cdot \underbrace{(1-e^{-t})^{k-i}}_{\text{prob of transition}}\cdot \underbrace{(e^{-t})^{d-k}}\_{\text{prob of unmask pre}}. $$
>
> Under this condition, the expected number of MASK token at forward time $t$ will become
> $$
> \mathbb{E}[\mathrm{numK}(y^\to_t)] = \sum_{k=0}^d k\cdot \mathrm{Pr}[\mathrm{numK}(y^\to_t) = k] = \sum_{k=0}^d \sum_{i=0}^k C_{d-i}^{k-i}\cdot (1-e^{-t})^{k-i}\cdot (e^{-t})^{d-k}\cdot \mathrm{Pr}[\mathrm{numK}(y^\to_0) = i]
> $$
>
> $$
> = \sum_{i=0}^d \mathrm{Pr}[\mathrm{numK}(y^\to_0) = i]\cdot  \sum_{k=i}^d C_{d-i}^{k-i} \cdot (1-e^{-t})^{k-i}\cdot (e^{-t})^{d-k} =  \sum_{i=0}^d \mathrm{Pr}[\mathrm{numK}(y^\to_0) = i]\cdot (d-de^{-t}+ie^{-t}) = d\left(1-(1-m_0/d)\cdot e^{-t}\right)
> $$
> where $m_0$ denotes the expected number of MASK token in data distribution. Such an expectation shares the same property as the eixisting analysis, i.e., the number of token decreases and benefits the efficiency of inference process. Finally, the complexity will be
> $$
> \min\left\\{O(Kd\ln(d/\epsilon)), O\left(Kd\cdot m_0/\epsilon\right)\right\\} + O(Kd\ln d)
> $$
> with proper paramter settings, whose worst result will match that shown in [1].
>
> **For the relation to Assumption 4 in [1]**, we consider [A2] in our paper and Assumption 4 in [1] to be somewhat opposite. Specifically, Assumption 4 in [1] assumes that the MASK token appears with a strictly positive, lower-bounded probability in the data distribution, whereas our \([A2]\) assumes it does not appear at all.
>
> > W2: Encouraged to include a few small-scale experiments (maybe just one on a toy model) to illustrate how the theoretical studies to explain why masked discrete diffusion model surpasses the uniform discrete diffusion model under certain circumstances.
>
> W2: We thank the reviewer for this great suggestion. To bridge the gap between our theoretical rates and practical behavior, we have added a new "Synthetic Experiments" section in the Appendix. We empirically validate our convergence bounds in a controlled setting ($K=3, d=4$). As predicted by our theory, the results demonstrate that the Masked Discrete Diffusion model achieves significantly faster convergence to the target distribution (measured by TV distance) compared to the uniform-stationary baseline. Furthermore, we provide a qualitative comparison of sampling trajectories, illustrating how the theoretical efficiency gains translate into fewer and more decisive state updates (lower NFE) in practice. Please see Figure 1 for the convergence plots and Figure 2 for the trajectory visualizations.
>
> > W3: Include a complete list of related work
>
> A3: Thank you for your detailed review and for suggesting these references! We have now updated our paper to include [3], [4], and [5] in Section 6 (Related Work), along with an expanded discussion.

---

> > ### Author Response · Authors · 2025-11-25
> >
> > > Q3: Whether the technique developed in this paper can be applied to analyze the Tweedie $\tau$-leaping
> >
> > AQ3: Good question. Thank you for the excellent question. We do believe our proof technique can be extended to analyze the Tweedie $\tau$-leaping from [2]. Notably, it may help **remove the bounded-score requirement** currently assumed in that approach—an assumption that typically forces the truncation of neural network outputs. Such a restriction can be difficult to handle under Girsanov-based methods used in standard $\tau$-leaping analyses because of their rigid structure. By contrast, our technique—eschewing Girsanov’s Theorem in favor of a direct KL-divergence chain rule—affords greater flexibility.
> >
> > **Regarding the complexity order** of Tweedie $\tau$-leaping, we anticipate that it would remain largely unchanged, primarily because its time-discretization step already dominates the overall inference error. While we cannot definitively predict how incorporating our proof technique might modify the final results, we see it as a promising direction for future exploration.
> >
> > ```
> > [1] Liang Y, Huang R, Lai L, et al. Absorb and Converge: Provable Convergence Guarantee for Absorbing Discrete Diffusion Models[J]. arXiv preprint arXiv:2506.02318, 2025.
> >
> > [2] Lou A, Meng C, Ermon S. Discrete diffusion modeling by estimating the ratios of the data distribution[C]//Proceedings of the 41st International Conference on Machine Learning. 2024: 32819-32848.
> >
> > [3] Srikanth, Aadithya, Mudit Gaur, and Vaneet Aggarwal. "Discrete State Diffusion Models: A Sample Complexity Perspective." arXiv preprint arXiv:2510.10854 (2025).
> >
> > [4] Wan, Zhengyan, Yidong Ouyang, Qiang Yao, Liyan Xie, Fang Fang, Hongyuan Zha, and Guang Cheng. "Error Analysis of Discrete Flow with Generator Matching." arXiv preprint arXiv:2509.21906 (2025).
> >
> > [5] Liang, Yuchen, Yingbin Liang, Lifeng Lai, and Ness Shroff. "Discrete Diffusion Models: Novel Analysis and New Sampler Guarantees." arXiv preprint arXiv:2509.16756 (2025).
> > ```

---

> > > ### Comment · Reviewer_1mpG · 2025-11-25
> > > **Response to the authors**
> > >
> > > I would like to thank the authors for the detailed rebuttal and revision of your manuscript. Since I am satisfied with the authors' responses to the raised questions, I'm raising my score from 6 to 8.

---

> > > > ### Author Response · Authors · 2025-11-27
> > > >
> > > > We sincerely thank the reviewer for their time and dedication, as well as for raising the rating. If there are any further questions, we will gladly provide additional clarification.

---

### Meta-Review · Area_Chair_3MkQ · 2026-01-07

**Summary:**

This paper provides, for the first time, a quantitative theoretical guarantee on Euler sampling of masked diffusion model. It also proposes a new sampling method based on the technique of uniformization and leveraging specific features of masked diffusion, and proves a significantly improved oracle complexity of the new method. While some reviewers question the practicality of the work, some other reviewers and I appreciate the theoretical quantifications. However, while the scores changed from 8,6,2,2 to 8,8,2,2 post rebuttal, they are still biomodally distributed, and the negative reviewers raised significant concerns about experimental demonstration and the extent of novelty. These concerns seem to be rather important and unfortunately did not appear to have fully resolved. Recognizing the potential of the idea, I encourage the authors to take the discussions into consideration and re-submit a revised version.

**Reviewer Concerns:**

Just a guess: QNxW and t4vz are probably not going to be fully convinced.

**Reviewer Scores:**

Just a guess: QNxW is probably not going to increase. t4vz might increase by 2.

---

### Decision · Program_Chairs · 2026-01-26

Reject